# Bridging Unsupervised and Semi-Supervised Anomaly Detection: A Provable and Practical Framework with Synthetic Anomalies

## Abstract

Anomaly detection (AD) is a critical task across domains such as cybersecurity and healthcare. In the unsupervised setting, an effective and theoretically-grounded principle is to train classifiers to distinguish normal data from (synthetic) anomalies. We extend this principle to semi-supervised AD, where training data also include a limited labeled subset of anomalies possibly present in test time. We propose a **theoretically-grounded and empirically effective framework** for semi-supervised AD that combines known and synthetic anomalies during training. To analyze semi-supervised AD, we introduce the first mathematical formulation of semi-supervised AD, which generalizes unsupervised AD. Here, we show that synthetic anomalies enable (i) better anomaly modeling in low-density regions and (ii) optimal convergence guarantees for neural network classifiers — the first theoretical result for semi-supervised AD. We empirically validate our framework on five diverse benchmarks, observing consistent performance gains. These improvements also extend beyond our theoretical framework to other classification-based AD methods, validating the generalizability of the synthetic anomaly principle in AD.

## 1 Introduction

Anomaly detection (AD) — the task of identifying data that deviate from expected behavior — is central in many domains, from daily usage in manufacturing (Bergmann et al., 2019) and content moderation (Chen et al., 2022) to high stakes domains like cybersecurity (Tavallaee et al., 2009; Lee et al., 1999) and healthcare (Quinlan, 1987; Guvenir et al., 1998). Despite its broad applicability, most AD research focuses on unsupervised AD, where only normal data are available during training. When limited anomalies are also available during training, many unsupervised methods do not handle this additional information and remove these "known" training anomalies (e.g., Kim et al. (2020); Chen et al. (2017); Zhang et al. (2022); Qiu et al. (2021); Shenkar & Wolf (2022); Xiao & Fan (2024)). Ideally, models should incorporate these known anomalies during training while still detecting "unknown anomalies" (i.e., anomaly types absent during training) during test time. Can unsupervised AD principles generalize to semi-supervised AD?

We address this question by focusing on a key principle from unsupervised AD: training classifiers to distinguish normal data from (randomly generated synthetic) anomalies. This principle has both theoretical justification and empirical success in unsupervised settings (Steinwart et al., 2005; Zhou et al., 2024; Sipple, 2020), yet its effectiveness and validity in the semi-supervised regime remain unexplored.

At first glance, mixing synthetic with known anomalies might dilute the known anomaly signal — the anomaly class during training contains both known and synthetic anomalies. Synthetic anomalies may also contaminate regions with normal data. However, we claim that synthetic anomalies are key in semi-supervised AD. In this work, **we propose that adding synthetic anomalies during training is a theoretically-grounded and empirically effective framework for semi-supervised AD**.

Theoretically, we provide the first mathematical formulation of semi-supervised AD (Figure 1). This formulation reveals the benefits of synthetic anomalies: they (i) label low density regions of normal data as anomalous and (ii) improve model learning. The former suggests that our formulation *models* AD well, while the latter allows us to prove the first theoretical *learning* guarantees for

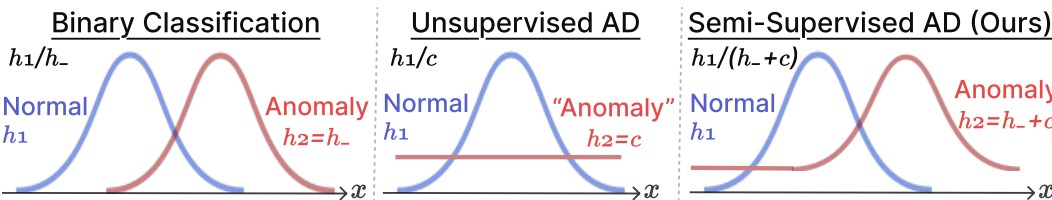

Figure 1: **Mathematical Formulations.** Different tasks based on the common test $h_1/h_2$. Our semi-supervised anomaly detection formulation combines known anomaly information $h_-$ with the density level set estimation formulation from unsupervised anomaly detection.

semi-supervised AD with neural networks. Our theoretical model also recommends the number of synthetic anomalies to add, mitigating issues of dilution and contamination of real training data.

We also demonstrate that our theoretical framework of adding synthetic anomalies translates into a practical and effective implementation, evaluating our framework on five real-world datasets. We observe that synthetic anomalies can improve performance on both known and unknown anomalies. This improvement is not only seen for our theoretical model, but also for other state-of-the-art classification-based AD methods. These analyses on theoretical guarantees and empirical evaluations on diverse datasets and AD methods demonstrate the feasibility of adding synthetic anomalies in semi-supervised AD. We summarize our contributions below:

- We propose a theoretically-governed and empirically effective framework for semi-supervised AD, adding synthetic anomalies to the anomaly class for binary classification during training.

- We provide the first mathematical formulation for semi-supervised AD which generalizes unsupervised AD to allow for known anomalies.

- We show that adding synthetic anomalies to the anomaly class during training sidesteps two potential problems of anomaly modeling and ineffective learning.

- To show effective learning, we prove the optimal convergence of the excess risk of our neural network binary classifiers, the first theoretical result in semi-supervised AD.

- Our experiments demonstrate that adding synthetic anomalies improves performance. This improvement extends beyond our concrete example of vanilla binary classifiers to other classification-based AD methods, highlighting our method's generalizability.

## 2 RELATED WORKS

**Semi-Supervised AD** Unlike unsupervised AD methods which assume all training data are normal, other methods have been able to leverage on the known anomaly sample during training with some empirical success (Han et al., 2022; Ruff et al., 2020; Pang et al., 2019; Zhou et al., 2022; Lau et al., 2024b;a; Goyal et al., 2020; Yamanaka et al., 2019; Pang et al., 2019; Dong et al., 2024; Zhao et al., 2023). For instance, Han et al. (2022) shows that even with 1% labeled anomalies, methods incorporating supervision empirically outperform unsupervised AD methods. However, there is currently no mathematical formulation of the goal of semi-supervised AD, let alone a theoretically-grounded approach towards it. Without a mathematical formulation, unsupervised and semi-supervised AD remain as research areas with disjoint scopes.

**Auxiliary Data** Using auxiliary data for (unsupervised) AD is popular in applied domains, such as generating anomalies from normal data (Fan et al., 2001; Cao et al., 2024; Dong et al., 2024). In our work, we wish to understand the general theoretical underpinnings of AD, so we avoid using domain-specific knowledge. The first general theory for unsupervised AD with synthetic anomalies used uniformly random data as synthetic anomalies for support vector machine (SVM) classification (Scott & Nowak, 2006). Sipple (2020) experimented with neural networks instead, while Cai & Fan (2022) used another neural network for anomaly generation and Hendrycks et al. (2019) used open-source data as anomalies for image AD. Correspondingly, Zhou et al. (2024) provided the

theoretical analysis for neural networks using synthetic anomalies. However, these works are for unsupervised AD, not semi-supervised AD.

# 3 FORMULATING ANOMALY DETECTION

In this section, we provide a general AD formulation assuming full knowledge of anomalies. Then, we explore a potential formulation of semi-supervised AD that relaxes this assumption.

## 3.1 BACKGROUND: MODELING ANOMALY DETECTION AS BINARY CLASSIFICATION

First, consider a binary classification problem between $Y = 1$ (normal class) and $Y = -1$ (anomaly class). Let $\mu$ be a known probability measure on our domain $\mathcal{X} \subseteq \mathbb{R}^d$. Without loss of generality, let $\mathcal{X} = [0,1]^d$. Assume data from the normal and anomaly classes are drawn respectively from unknown distributions $Q$ and $W$ on $\mathcal{X}$, where $Q$ has density $h_1$ with respect to $\mu$, and $W$ has density $h_2$ with respect to $\mu$.

Let $s \in (0,1)$ denote the proportion of normal data on $\mathcal{X}$ such that $\mathrm{P}(Y = 1) = s$ and $\mathrm{P}(Y = -1) = 1 - s$. Let $P$ be a probability measure on $\mathcal{X} \times \mathcal{Y}$ such that the marginal distribution on $\mathcal{X}$ is $P_{\mathcal{X}} = sQ + (1-s)W$. For any classifier $\mathrm{sign}(f)$ induced by a function $f : \mathcal{X} \to \mathbb{R}$, its misclassification error is given as $R(f) = \mathrm{P}(\mathrm{sign}(f(X)) \neq Y)$. The best we can do is obtain the *Bayes classifier, denoted by $f_c$*, which minimizes the misclassification error, i.e., $R(f_c) = R^* := \inf_{f:\mathcal{X} \to \mathbb{R} \text{ measurable}} R(f)$. Like other settings (Zhang, 2004), the Bayes classifier $f_c$ is *explicitly given as $f_c(X) = sign(f_P(X))$* (discussed in Appendix B), where $f_P$ is the regression function

$$f_P(X) := \mathrm{E}[Y|X] = \frac{s \cdot h_1(X) - (1-s) \cdot h_2(X)}{s \cdot h_1(X) + (1-s) \cdot h_2(X)}, \qquad \forall X \in \mathcal{X}. \tag{1}$$

The Bayes classifier can also be defined with the likelihood ratio test (Bartlett et al., 2006; Hastie, 2009)

$$\mathbb{1}\left(\frac{h_1(X)}{h_2(X)} \geq \rho\right) \tag{2}$$

for threshold $\rho = \frac{\mathrm{P}(Y=-1)}{\mathrm{P}(Y=1)}$. However, in AD, we set threshold $\rho$ based on desired type I and II errors.

We proceed to define the AD error of function $f : \mathcal{X} \to \mathbb{R}$. Define the set of data classified as normal as $\{f > 0\} := \{X : f(X) > 0\}$. Let $s = \frac{1}{1+\rho}$ and the classical Tsybakov noise condition (Tsybakov, 1997; 2004)

$$\mathrm{P}_X(\{X \in \mathcal{X} : |f_P(X)| \leq t\}) \leq c_0 t^q, \qquad \forall t > 0, \tag{3}$$

hold with some $c_0 > 0$ and noise exponent $q \in [0, \infty)$. Then, for any measurable function $f : \mathcal{X} \to \mathbb{R}$, we extend Steinwart et al. (2005) (proven in Appendix E.1) to derive a bound on the AD error

$$S_{\mu,h_1,h_2,\rho}(f) := \mu\big(\{f > 0\} \Delta \{h_1/h_2 > \rho\}\big) \geq C_q(R(f) - R^*)^{\frac{q}{q+1}}. \tag{4}$$

Here, $\Delta$ denotes the symmetric difference, $S_{\mu,h_1,h_2,\rho}(f)$ measures how well $\{f > 0\}$ matches the ground-truth set $\{h_1/h_2 \geq \rho\} := \{X : h_1(X)/h_2(X) \geq \rho\}$ (as in equation 2). $C_q$ is a positive constant depending on $c_0$ and $q$. From (4), we see $S_{\mu,h_1,h_2,\rho}(f) \to 0$ if $R(f) - R^* \to 0$. This implies that the excess risk $R(\cdot) - R^*$, a standard error metric for binary classification, serves as a surrogate for $S_{\mu,h_1,h_2,\rho}(\cdot)$ and, thus, provides a viable error metric for AD (similar to Steinwart et al. (2005)). In other words, **to solve AD, we can solve a standard binary classification problem**.

However, the test-time anomaly density $h_2$ is not known in AD. Unsupervised AD (i.e., only normal data during training) gets around this challenge with a density level set estimation formulation (Ruff et al., 2021)

$$\{h_1 \geq \rho\} := \{X : h_1(X) \geq \rho\}. \tag{5}$$

This formulation (5) can be interpreted as a likelihood ratio test between $h_1$ and a constant, because it is a special case of (2) with $h_2 \equiv 1$. In contrast, for semi-supervised AD, we would like to set $h_2$ to reflect our partial knowledge through our known anomaly sample.

The question we seek to answer is — **is it possible to apply this generalization to semi-supervised AD?** If so, what should $h_2$ be to model semi-supervised AD? Straightforwardly, we can set $h_2$ to be the known anomaly density. However, we proceed to show two potential issues with this approach.

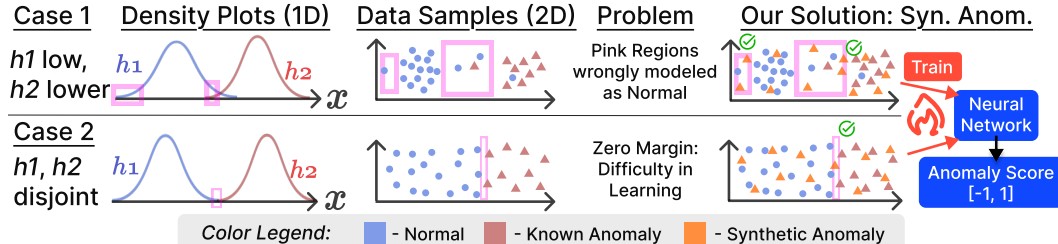

Figure 2: **Problem and Proposed Method.** We present two cases of potential issues (pink regions) from using the known anomaly density as $h_2$ (i.e., set $h_2 = h_-$). Sample 1D density plots and 2D data plots are provided. Circles are normal data while triangles are anomalies. The first row demonstrates the issue of wrongly modeling anomalies as normal, while the second row demonstrates the issue of insufficient regularity of learning, with the example of zero margin between normal data and known anomalies. Our solution of adding synthetic anomalies addresses these issues. In the first row, the pink region is **populated with anomalies**, so we can correctly model it as anomalous. In the second row, **synthetic anomalies smoothen the regression function**, which improve learning.

### 3.2 TWO POTENTIAL ISSUES WITHOUT SYNTHETIC DATA

For concreteness, let our training data contain normal samples $T = \{(X_i, 1)\}_{i=1}^{n} \overset{\text{i.i.d.}}{\sim} Q$ and anomalies $T^- = \{(X_i^-, -1)\}_{i=1}^{n^-} \overset{\text{i.i.d.}}{\sim} V$, where $V \neq W$ is an unknown distribution with density $h_-$. **The straightforward approach is to use $T^-$ during training (i.e., without synthetic anomalies), implicitly setting $h_2 = h_-$.** However, we proceed to show two potential issues with this approach.

**The first potential issue is the "false negative modeling" problem**, where anomalies are modeled as normal data. This may happen in regions where normal density $h_1$ is low, but known anomaly density $h_-$ is even lower, leading to $h_1(X)/h_2(X)$ exploding. In other words, **low-density regions of $h_1$ can still be classified as normal**. This is undesirable. Take a medical application. Refer to the density plot in the first row of Figure 2 and let $x$ refer to blood pressure. Let $h_1$ refer to normal patients and $h_2$ (known anomalies) refer to sick patients with high blood pressure. Consider a "test-time" patient with low blood pressure $X$ (see pink region on the left of $h_1$ in Figure 2). Here, $h_1(X) \gg h_2(X)$, so this patient will be modeled as normal. However, we wish to model low blood pressure as anomalous because the probability of a normal patient with low blood pressure $h_1(X)$ is low.

**The second potential issue is the "insufficient regularity of learning" problem**, where the trained neural network classifier can produce high error. This can arise from the discontinuity of the regression function $f_P$, making it challenging to learn the optimal classifier. **Our novel observation is that, without synthetic anomalies in training data, the regression function is prone to discontinuity**, which impacts effective learning. Proposition 3.1 (proven in Appendix E.2) illustrates a general scenario (see Figure 2 and Appendix D for examples), where $f_P$ is discontinuous despite both $h_1$ and $h_-$ being continuous.

**Proposition 3.1** (Separable Data with Zero Margin). *Let $r > 0$ and $\mathcal{X}$ be the union of two intersecting, closed subdomains $\mathcal{X}_1$ and $\mathcal{X}_-$ with $\mathrm{interior}(\mathcal{X}_1 \cap \mathcal{X}_-) = \emptyset$. Suppose $h_1 \in C^r(\mathcal{X})$ has support $\mathcal{X}_1$ and $h_- \in C^r(\mathcal{X})$ has support $\mathcal{X}_-$. For $h_2 = h_-$, the regression function reduces to*

$$f_P(X) = \begin{cases} \frac{h_1(X)}{h_1(X)} = 1, & \text{if } X \in \mathrm{interior}(\mathcal{X}_1), \\ -\frac{h_-(X)}{h_-(X)} = -1, & \text{if } X \in \mathrm{interior}(\mathcal{X}_-), \end{cases}$$

*which is discontinuous on $\mathcal{X}$. Moreover, for any continuous function $f : \mathcal{X} \to \mathbb{R}$, the approximation error is at least $\|f - f_P\|_{L^\infty[0,1]^d} \geq 1$.*

Next, we show that the discontinuity of $f_P$ poses a difficulty for classification by neural networks. We consider feedforward rectified linear unit (ReLU) neural networks. We outline notation below.

**Definition 3.2.** Let $\sigma(x) = \max\{0, x\}$ be the ReLU activation function. A ReLU network $f : \mathcal{X} \to \mathbb{R}$ with $L \in \mathbb{N}$ hidden layers and width vector $\boldsymbol{p} = (p_1, \dots, p_L) \in \mathbb{N}^L$, which indicates the width in each hidden layer, is defined in the following compositional form:

$$f(X) = a \cdot \sigma\big(W^{(L)} \dots \sigma\big(W^{(1)} X + b^{(1)}\big) \dots\big) + b^{(L)}, \tag{6}$$

where $X \in \mathcal{X} = [0,1]^d$ is the input, $a \in \mathbb{R}^{p_L}$ is the outer weight, $W^{(i)}$ is a $p_i \times p_{i-1}$ weight matrix with $p_0 = d$, and $b^{(i)} \in \mathbb{R}^{p_i}$ is a bias vector, for $i = 1, \ldots, L$. Let $\sigma^k$ be the ReLU$^k$ function, a generalization of ReLU for $k \in \mathbb{N}$, defined by $\sigma^k(x) = (\max\{0, x\})^k$. Define the "approx-sign function" $\sigma_\tau : \mathbb{R} \to [0, 1]$, with a bandwidth parameter $\tau > 0$, as

$$\sigma_\tau(x) = \frac{1}{\tau}\sigma(x) - \frac{1}{\tau}\sigma(x - \tau) - \frac{1}{\tau}\sigma(-x) + \frac{1}{\tau}\sigma(-x - \tau). \tag{7}$$

We also define the generalized approx-sign$^k$ function as $\sigma_\tau^k(x) := \frac{1}{k!\tau^k}\sum_{\ell=0}^{k}(-1)^\ell\binom{k}{\ell}\sigma^k(x - \ell\tau) - \frac{1}{k!\tau^k}\sum_{\ell=0}^{k}(-1)^\ell\binom{k}{\ell}\sigma^k(-x - \ell\tau)$ for $k \in \mathbb{N}$. Here, the approx-sign function is designed to approximate the sign function (as $\tau \to 0$). Meanwhile, $k \in \mathbb{N}$ is a parameter controlling the smoothness of ReLU (and the approx-sign activation function), generalizing our analysis beyond the original non-smooth ReLU function. We defer discussions and visualizations to Appendix C.2.

The following **novel theorem** presents an upper bound for the excess risk of a function $\sigma_\tau^k(f)$ induced by the output activation $\sigma_\tau^k$ in terms of the bandwidth $\tau$ and the approximation error.

**Theorem 3.3.** *Assume the Tsybakov noise condition (3) holds for some exponent $q \in [0, \infty)$ and constant $c_0 > 0$. For any measurable function $f : \mathcal{X} \to \mathbb{R}$, there holds*

$$\underbrace{R\left(\sigma_\tau^k(f)\right) - R(f_c)}_{\text{excess risk}} \le 4c_0\big(k\tau + \underbrace{\|f - f_P\|_{L^\infty[0,1]^d}}_{\text{approximation error}}\big)^{q+1}.$$

Theorem 3.3 shows that the smaller the approximation error, the smaller the excess risk in classification. We discuss the significance of this theorem in Remark C.3 and prove it in Appendix F.

From Proposition 3.1 and Theorem 3.3, we can see that **if the regression function is discontinuous, the approximation error is high (at least 1), which may lead to vacuous excess risk bounds**[1] (i.e., excess risk can be high and is not guaranteed to converge). Lacking theoretical guarantees, **the Bayes classifier cannot be effectively learned**. Due to (i) an undesirable formulation and (ii) lack of theoretical guarantees, we see that $h_2 = h_-$ is not ideal. In the next section, we propose a semi-supervised AD method to mitigate these two issues.

# 4 OUR PROPOSED METHOD: SEMI-SUPERVISED AD WITH SYNTHETIC ANOMALIES

## 4.1 OVERVIEW OF OUR METHOD

Building on the previous classification framework and inspired by the connection between density level set estimation and synthetic anomalies, we propose to add synthetic anomalies to mitigate the two aforementioned issues (Figure 2). In addition to samples $T$ and $T^-$, we generate a set of synthetic anomalies $T' = \{(X_i', -1)\}_{i=1}^{n'}$, where each $X_i'$ is sampled i.i.d. from $\mu = \text{Uniform}(\mathcal{X})$. Our full training dataset becomes $T \cup T^- \cup T'$, which we use to train a ReLU network classifier.

## 4.2 MITIGATING ISSUE 1: FALSE NEGATIVE MODELING PROBLEM

Let $\tilde{s} \in (0, 1)$ denote a mixture parameter. By introducing synthetic anomalies, we are implicitly changing the density function representing the anomaly class to

$$h_2 = \tilde{s}h_- + (1 - \tilde{s}), \tag{8}$$

which corresponds to a mixture. Here, a proportion $\tilde{s}$ of anomalies are drawn from known anomaly density $h_-$, and the remaining proportion $(1 - \tilde{s})$ of (synthetic) anomalies are drawn from the distribution $\mu$. We see that (in (8)) $h_2$ **is bounded away from 0** due to the constant term $1 - \tilde{s} > 0$, **preventing** $h_1/h_2$ **from exploding even when** $h_1$ **is small**. Hence, **low probability density normal data will not be modeled as anomalous** even in regions where $h_2$ is small.

---

[1]The bound could be vacuous because excess risk is always at least 1.

*Remark* 4.1. When $h_-$ is constant, known anomalies are drawn from the uniform distribution, providing no additional prior on how anomalies can arise. This uninformative case also arises when mixture parameter $\tilde{s} = 0$. In both cases, $h_2$ will be constant, and (2) reduces to the density level set estimation problem $\{h_1 > \rho\}$ of unsupervised AD. In other words, **our semi-supervised AD framework is a generalization of unsupervised AD that allows for known anomaly supervision**.

### 4.3 MITIGATING ISSUE 2: INSUFFICIENT REGULARITY OF LEARNING PROBLEM

Adding synthetic anomalies from the uniform distribution can also improve the smoothness of the regression function. Later in Section 4.4, we use this fact to show how we can effectively learn the Bayes classifier. While Proposition 3.1 illustrated that the regression function $f_P$ can be discontinuous despite $h_1$ and $h_2$ being continuous, **our next novel result shows that adding synthetic anomalies ensures continuity of regression function** $f_P$ under the same conditions.

**Proposition 4.2.** *Suppose the condition stated in Proposition 3.1 holds. If we add synthetic anomalies from $\mu = Uniform(\mathcal{X})$ (i.e., $h_2 = \tilde{s}h_- + (1 - \tilde{s})$ with $\tilde{s} \in (0, 1)$), the regression function is*

$$f_P(X) = \frac{s \cdot h_1(X) - (1-s)\tilde{s} \cdot h_-(X) - (1-s)(1-\tilde{s})}{s \cdot h_1(X) + (1-s)\tilde{s} \cdot h_-(X) + (1-s)(1-\tilde{s})},$$

*which is $C^r$ continuous.*

For concreteness, we present 2 examples in Appendix D to illustrate how synthetic anomalies enhance the smoothness of regression function $f_P$.

Previously, from Proposition 3.1, we know that if $f_P$ is discontinuous, no ReLU neural network can approximate it well. Conversely, if $f_P$ **is continuous**, a well-established body of research has proved that ReLU neural networks can approximate it to any desired accuracy (e.g., Theorems 1 and 2 in Yarotsky (2017), Theorem 5 in Schmidt-Hieber (2020), Theorem 1.1 in Shen et al. (2022)). However, we **cannot directly use existing results because the i.i.d. assumption is violated** — anomalies are not drawn i.i.d. from $h_2$, but they are drawn from $h_-$ (known anomalies) and $\mu$ (synthetic anomalies) separately. We proceed to derive a novel theoretical result that accommodates this non-i.i.d. setting.

### 4.4 PROPOSED NEURAL NETWORK WITH SYNTHETIC ANOMALIES AND THEORETICAL GUARANTEES

We proceed to show that our method achieves minimax optimal convergence of the excess risk (and consequently, the AD error metric), **the first theoretical guarantee in semi-supervised AD**.

We adopt ReLU neural networks. We construct a specific class of ReLU neural networks (i.e., our hypothesis space) to learn the Bayes classifier $f_c$ well. We introduce some notation to formally define this hypothesis space.

**Definition 4.3.** Let $\|W^{(i)}\|_0$ and $|b^{(i)}|_0$ denote the number of nonzero entries of $W^{(i)}$ and $b^{(i)}$ in the $i$-th hidden layer, $\|\boldsymbol{p}\|_\infty$ denote the maximum number of nodes among all hidden layers, and $\|\boldsymbol{\theta}\|_\infty$ denote the largest absolute value of entries of $\{W^{(i)}, b^{(i)}\}_{i=1}^L$. For $L, w, v, K > 0$, we denote the form of neural network we consider in this work by

$$\mathcal{F}(L, w, v, K) := \left\{ f \text{ of the form of (6)} : \|\boldsymbol{p}\|_\infty \le w, \sum_{i=1}^L \left( \|W^{(i)}\|_0 + |b^{(i)}|_0 \right) \le v, \|\boldsymbol{\theta}\|_\infty \le K \right\}.$$

With $\sigma_\tau$ given in (7), we **define our hypothesis space $\mathcal{H}_\tau$ with $\tau \in (0, 1]$ to be functions generated by $\mathcal{H}_\tau := \text{span}\left\{ \sigma_\tau \circ f : f \in \mathcal{F}(L^*, w^*, v^*, K^*) \right\}$** for specific $L^*, w^*, v^*, K^* > 0$.

**Definition 4.4.** To make computation feasible, it is common to adopt some convex, continuous loss to replace the 0-1 classification loss function. Among all functions in $\mathcal{H}_\tau$, we specifically consider the empirical risk minimizer (ERM) w.r.t. Hinge loss $\phi(x) := \max\{0, 1 - x\}$ defined as

$$f_{\text{ERM}} := \arg \min_{f \in \mathcal{H}_\tau} \varepsilon_{T, T^-, T'}(f), \tag{9}$$

where the empirical risk w.r.t. $\phi$ is

$$\varepsilon_{T, T^-, T'}(f) := \frac{s}{n} \sum_{i=1}^n \phi\left(f(X_i)\right) + \frac{(1-s)\tilde{s}}{n^-} \sum_{i=1}^{n^-} \phi(-f(X_i^-)) + \frac{(1-s)(1-\tilde{s})}{n'} \sum_{i=1}^{n'} \phi(-f(X_i')),$$

$$\tag{10}$$

which uses normal data, known anomalies and synthetic anomalies from a uniform distribution. Note that $n$ and $n^-$ denote the number of normal and (real) anomalous training samples respectively, and $n'$ denotes the number of synthetic anomalies we generate.

**The following theorem shows that the excess risk of the ERM, $f_{\text{ERM}}$ (9), trained on normal data, known anomalies and synthetic anomalies, converges to 0 at an optimal rate** (up to a logarithmic factor) **as the number of training data increases.**

**Theorem 4.5.** *Let* $n, n^-, n' \geq 3$, $n_{min} = \min\{n, n^-, n'\}$, $d \in \mathbb{N}$, $\alpha > 0$. *Assume the Tsybakov noise condition (3) holds for noise exponent* $q \in [0, \infty)$ *and constant* $c_0 > 0$, *and the regression function* $f_P$ *is* $\alpha$-*Hölder continuous. Consider the hypothesis space* $\mathcal{H}_\tau$ *with* $N = \left\lceil \left( \frac{n_{min}}{(\log(n_{min}))^4} \right)^{\frac{d}{d+\alpha(q+2)}} \right\rceil$, $\tau = N^{-\frac{\alpha}{d}}$, $K^* = 1$, *and* $L^*, w^*, v^*$ *depending on* $N, \alpha, d$ *given explicitly in Appendix G. For any* $0 < \delta < 1$, *with probability* $1 - \delta$, *there holds,*

$$R(sign(f_{ERM})) - R(f_c) = \mathcal{O}\left( \left( \frac{(\log n_{min})^4}{n_{min}} \right)^{\frac{\alpha(q+1)}{d+\alpha(q+2)}} \right).$$

The full proof and explicit excess risk bound are given in Appendix G. Theorem 4.5 tells us that when $n_{\min} = \min\{n, n^-, n'\}$ increases, the excess risk converges to 0 at a rate $\mathcal{O}\left( (n_{\min})^{-\frac{\alpha(q+1)}{d+\alpha(q+2)}} \right)$ (dropping the logarithmic factor). This rate matches the minimax rates in the literature (Audibert & Tsybakov, 2007) because $n_{\min}$ captures the minimum sample size across normal, anomalous and synthetic training data. Applying Theorem 4.5 to (4), we obtain with probability $1 - \delta$, the AD error $S_{\mu,h_1,h_2,\rho}(sign(f_{\text{ERM}})) = \mathcal{O}\left( (n_{\min})^{-\frac{\alpha q}{d+\alpha(q+2)}} \right)$. As the number of training data grows, the AD error converges to 0, **suggesting that ReLU network can solve semi-supervised AD effectively.** Next, we conduct experiments with real-world data to evaluate the practical efficacy of synthetic anomalies.

## 5 EXPERIMENTS

### 5.1 SET-UP

In this section, we evaluate the area under the precision-recall curve (AUPR) of neural networks with vanilla classification (VC) (Han et al., 2022). We also test other AD methods (mentioned in order of their proximity in modeling VC): ES (VC with modified activation function) (Lau et al., 2024b;a), DROCC (VC but with adversarial synthetic anomalies) (Goyal et al., 2020), ABC (VC with autoencoder structure) (Yamanaka et al., 2019) and DeepSAD (Ruff et al., 2020) (autoencoder with latent hypersphere classification). All methods are evaluated with and without our proposed method of including randomly sampled synthetic anomalies (SA). VC-SA models our theoretical framework. Here, we are interested in 2 research questions (RQs). **RQ1. Do synthetic anomalies improve performance of VC** (i.e., VC-SA versus VC)? **RQ2. [Generalizability] Do synthetic anomalies improve performance of other state-of-the-art methods?** Results for RQ1 and RQ2 are reported in Tables 1a and 1b respectively. To avoid diluting the known anomaly supervision signal and contaminating the normal data during training, we *avoid adding too many synthetic anomalies*. Based on Theorem 4.5, we add $n' = n + n^-$ number of synthetic anomalies. We explained our choice of $n'$ and other hyperparameters in Appendix H.5.2.

As a note, we also evaluate 9 methods, each composing a binary classifier on an unsupervised AD method. These methods first do unsupervised AD to identify data that belong to the training classes, and then binary classifiers differentiate normal from known anomalous data given that the data are known. However, they consistently produce random (i.e., poor) performance and are unsuitable. We defer further details and discussions to Appendix H.4.

**Dataset** We summarize our five diverse real-world evaluation datasets spanning across **tabular**, **image** and **language** benchmarks. More details are in Appendix H.2. Our tabular datasets comprise NSL-KDD (cybersecurity) (Tavallaee et al., 2009), Thyroid (medical) (Quinlan, 1987) and Arrhythmia (medical) (Guvenir et al., 1998). MVTec (Bergmann et al., 2019) and AdvBench (Chen et al., 2022) are our image and language AD datasets respectively. Here, anomalies arise naturally from cyber-attacks, medical sickness, manufacturing defects and harmful text. For all datasets, we train with

Table 1: **AUPR↑ results with and without synthetic anomalies for (a) our theoretical model (vanilla classification, VC) and (b) other AD models**. Table 1b is a continuation of Table 1a, but separated as a different subtable to highlight the different RQs they are answering. Other models are arranged from left to right in order of how close they are to VC. More often than not, **synthetic anomalies** (-SA suffix) **generally improve results for VC and other AD models closer to the left (ES and DROCC)**. Performance gains are seen for both unknown (unk.) and known anomalies. Meanwhile, autoencoder-based methods ABC and DeepSAD have more mixed results.

(a) **RQ1.** AUPR↑ for our VC model. (b) **RQ2.** AUPR↑ for other methods.

| Dataset | Type | Anom. | Random | VC | VC-SA (Ours) | ES | ES-SA | DROCC | DROCC-SA | ABC | ABC-SA | DeepSAD | DeepSAD-SA |
|---|---|---|---|---|---|---|---|---|---|---|---|---|---|
| NSL-KDD | Unk. | DoS | 0.431 | 0.345±0.036 | **0.793±0.055** | 0.716±0.083 | **0.794±0.005** | **0.930±0.015** | 0.856±0.067 | 0.935±0.002 | **0.940±0.013** | **0.899±0.001** | 0.842±0.001 |
| | | Probe | 0.197 | 0.180±0.010 | **0.649±0.078** | 0.423±0.030 | **0.690±0.053** | 0.577±0.037 | **0.634±0.205** | **0.900±0.018** | 0.860±0.006 | **0.871±0.001** | 0.823±0.003 |
| | | RA | 0.218 | 0.543±0.019 | **0.609±0.050** | 0.565±0.036 | **0.583±0.035** | **0.663±0.081** | 0.582±0.062 | **0.546±0.034** | 0.507±0.010 | 0.391±0.001 | **0.520±0.003** |
| | Known | PE | 0.007 | **0.609±0.001** | 0.486±0.146 | 0.606±0.021 | **0.618±0.027** | **0.201±0.004** | 0.168±0.036 | **0.226±0.065** | 0.054±0.013 | **0.058±0.003** | 0.018±0.000 |
| Thyroid | Unk. | Hyper. | 0.023 | 0.565±0.465 | **0.817±0.039** | 0.501±0.006 | **0.588±0.032** | 0.064±0.056 | **0.115±0.018** | 0.092±0.001 | **0.154±0.003** | **0.221±0.004** | 0.161±0.005 |
| | Known | Sub. | 0.053 | 0.512±0.380 | **0.751±0.020** | 0.823±0.027 | **0.826±0.027** | **0.051±0.004** | 0.045±0.002 | 0.034±0.000 | 0.039±0.000 | **0.053±0.000** | 0.049±0.000 |
| Arrhyth. | Unk. | All | 0.751 | **0.854±0.030** | 0.846±0.003 | 0.826±0.017 | **0.853±0.003** | 0.815±0.016 | **0.842±0.005** | **0.896±0.002** | 0.890±0.004 | 0.863±0.000 | **0.873±0.001** |
| MVTec (Image) | Unk. | Bottle | 0.683 | 0.996±0.001 | **0.997±0.000** | **0.999±0.000** | **0.999±0.000** | **0.999±0.003** | **0.999±0.000** | 0.987±0.009 | **0.994±0.005** | **0.973±0.001** | 0.954±0.002 |
| | | Cable | 0.577 | 0.795±0.013 | **0.868±0.005** | 0.816±0.005 | **0.852±0.008** | 0.785±0.006 | **0.856±0.009** | 0.843±0.002 | **0.917±0.002** | 0.799±0.003 | **0.802±0.006** |
| | | Capsule | 0.789 | 0.908±0.005 | **0.947±0.002** | 0.911±0.002 | **0.949±0.002** | 0.907±0.005 | **0.956±0.002** | 0.964±0.009 | **0.976±0.001** | **0.933±0.000** | 0.925±0.001 |
| | | Carpet | 0.714 | 0.998±0.000 | **0.999±0.000** | 0.998±0.000 | **0.999±0.000** | 0.999±0.001 | **1.000±0.000** | 0.994±0.001 | **0.995±0.000** | **0.990±0.000** | 0.983±0.000 |
| | | Grid | 0.682 | 0.999±0.000 | **1.000±0.000** | 0.995±0.004 | **0.998±0.001** | 0.959±0.014 | **0.997±0.000** | 0.992±0.011 | **1.000±0.000** | 0.965±0.001 | **0.968±0.003** |
| | | Hazelnut | 0.565 | **0.954±0.004** | 0.943±0.009 | 0.935±0.036 | **0.958±0.005** | 0.911±0.059 | **0.941±0.003** | 0.934±0.063 | **0.952±0.002** | **0.810±0.002** | 0.792±0.005 |
| | | Leather | 0.695 | **1.000±0.000** | **1.000±0.000** | **1.000±0.000** | **1.000±0.000** | **1.000±0.000** | **1.000±0.000** | 0.977±0.006 | **0.998±0.000** | **0.998±0.000** | 0.998±0.001 |
| | | Metal Nut | 0.756 | **0.982±0.001** | 0.975±0.006 | **0.988±0.002** | 0.974±0.010 | **0.983±0.001** | 0.973±0.006 | 0.969±0.002 | **0.974±0.001** | **0.943±0.001** | 0.941±0.001 |
| | | Pill | 0.817 | 0.925±0.002 | **0.950±0.009** | **0.918±0.007** | 0.915±0.043 | 0.898±0.035 | **0.948±0.003** | 0.968±0.007 | **0.971±0.002** | 0.944±0.001 | **0.947±0.000** |
| | | Screw | 0.699 | 0.839±0.044 | **0.844±0.033** | **0.844±0.014** | 0.797±0.102 | 0.830±0.005 | **0.867±0.008** | 0.754±0.027 | **0.933±0.005** | 0.709±0.002 | **0.725±0.003** |
| | | Tile | 0.670 | 0.980±0.007 | **0.997±0.000** | 0.997±0.000 | **0.998±0.000** | 0.997±0.000 | **0.998±0.000** | 0.996±0.000 | **0.998±0.000** | **0.991±0.000** | 0.989±0.000 |
| | | Transistor | 0.333 | 0.786±0.026 | **0.872±0.010** | 0.780±0.019 | **0.815±0.066** | 0.795±0.014 | **0.884±0.012** | 0.851±0.017 | **0.900±0.008** | 0.836±0.000 | **0.840±0.003** |
| | | Wood | 0.732 | 0.984±0.008 | **0.991±0.008** | 0.986±0.001 | **0.995±0.002** | 0.992±0.002 | **0.996±0.002** | 0.980±0.001 | **0.997±0.001** | **0.979±0.001** | 0.977±0.001 |
| | | Zipper | 0.758 | 0.995±0.002 | **0.998±0.001** | 0.995±0.002 | **0.995±0.003** | **0.998±0.001** | **0.998±0.001** | 0.994±0.001 | **0.999±0.000** | **0.990±0.000** | 0.988±0.000 |
| Adv-Bench (Text) | Unk. | satnews | 0.082 | **0.798±0.028** | 0.232±0.030 | **0.585±0.037** | 0.540±0.023 | 0.283±0.010 | **0.367±0.031** | **0.151±0.031** | 0.082±0.013 | **0.080±0.006** | 0.074±0.000 |
| | | CGFake | 0.130 | **0.097±0.004** | 0.080±0.002 | **0.275±0.031** | 0.229±0.016 | 0.150±0.012 | **0.163±0.011** | **0.104±0.007** | 0.091±0.004 | **0.090±0.000** | 0.084±0.001 |
| | | jigsaw | 0.130 | 0.185±0.040 | **0.340±0.011** | **0.845±0.017** | 0.825±0.030 | **0.754±0.028** | 0.715±0.055 | **0.668±0.086** | 0.532±0.060 | 0.107±0.001 | **0.169±0.002** |
| | | EDENCE | 0.113 | 0.102±0.008 | **0.721±0.082** | **0.080±0.001** | 0.079±0.001 | 0.101±0.004 | **0.109±0.005** | 0.149±0.018 | **0.156±0.010** | **0.159±0.001** | 0.137±0.001 |
| | | FAS | 0.140 | 0.087±0.002 | **0.126±0.007** | 0.737±0.010 | **0.756±0.021** | 0.619±0.044 | **0.712±0.014** | **0.555±0.127** | 0.090±0.006 | **0.032±0.000** | 0.025±0.000 |
| | Known | LUN | 0.074 | **0.762±0.032** | 0.532±0.027 | 0.356±0.009 | **0.375±0.014** | 0.335±0.022 | **0.406±0.019** | **0.410±0.056** | 0.319±0.011 | **0.144±0.002** | 0.114±0.000 |
| | | amazon_lb | 0.107 | 0.123±0.017 | **0.824±0.036** | 0.427±0.054 | **0.444±0.070** | 0.154±0.008 | **0.311±0.085** | **0.078±0.032** | 0.026±0.001 | **0.023±0.001** | 0.021±0.000 |
| | | HSOL | 0.030 | 0.042±0.009 | **0.731±0.015** | 0.360±0.027 | **0.387±0.039** | 0.195±0.012 | **0.308±0.078** | **0.163±0.031** | 0.121±0.009 | 0.076±0.000 | **0.079±0.000** |
| | | assassin | 0.022 | 0.048±0.003 | **0.533±0.062** | 0.714±0.029 | **0.760±0.026** | 0.594±0.011 | **0.645±0.038** | **0.319±0.176** | 0.303±0.044 | **0.108±0.000** | 0.102±0.000 |
| | | enron | 0.080 | 0.211±0.037 | **0.361±0.015** | 0.122±0.003 | **0.138±0.005** | 0.119±0.002 | **0.125±0.006** | 0.097±0.004 | **0.101±0.002** | 0.131±0.000 | **0.137±0.001** |

normal data and one type of "known" anomaly, and evaluate on normal data and the remaining anomalies in the dataset (mostly unknown anomalies). For instance, NSL-KDD has benign (normal) network traffic and 4 types of attacks (anomalies) during training and testing: Denial of Service (DoS), probe, remote access (RA), and privilege escalation (PE). To simulate semi-supervised AD, we use RA as known anomalies and the other 3 as unknown anomalies. To convert image and text data to tabular form, we use 1024-dimensional DINOv2 embeddings (Oquab et al., 2023) and 384-dimensional BERT sentence embeddings (Reimers & Gurevych, 2019) respectively.

In total, we have 24 unknown categories and 7 known categories. Due to the small dataset size of Arrhythmia and MVTec, known anomalies are used only in training and not testing, and all unknown anomaly types are grouped together as a large unknown anomaly class for evaluation. We emphasize more on unknown anomaly evaluation because unknowns characterize the AD problem more than knowns (see the common density level set estimation formulation in Section 3). Nevertheless, we also include known categories to gauge if synthetic anomalies will dilute known anomaly training signal, or if they can improve known anomaly performance.

In addition to the AUPR results presented here, **we provide the corresponding AUROC results in Table 3 in Appendix H.1**, comparing performance of neural networks across all datasets with and without synthetic anomalies. The AUROC results generally demonstrate the superior performance of the VC model when synthetic anomalies are included.

## 5.2 Discussion

**RQ1. Is VC-SA better than VC?** Across all 5 datasets, VC-SA generally outperforms VC. VC-SA expectedly has better performance on unknown anomalies (better on 19/24 unknown anomaly categories), with synthetic anomalies providing supervision signal to classify unknown regions as anomalous (Figure 3). Interestingly, VC-SA outperforms VC on known anomalies on 5/7 of known anomaly categories (PE from NSL-KDD, subnormal from Thyroid and 5 categories from AdvBench). Adding synthetic anomalies improves our modeling of density level set estimation (Case 1 in Figure 2), so improving (unsupervised) AD is not necessarily negatively correlated with improving known anomaly performance, as seen here. Overall, **VC-SA performs better than VC**.

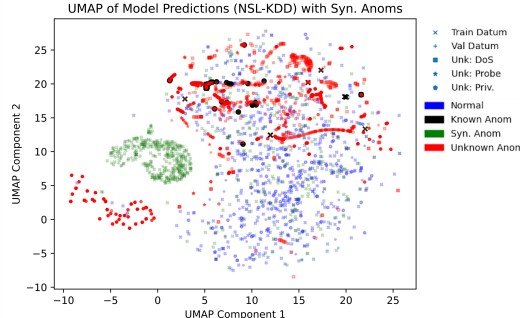

Figure 3: **Visualization**. Synthetic anomalies occupy regions with unknown anomalies (top right), training the model to classify unknown anomalies as anomalous.

**RQ2. How beneficial is adding random synthetic anomalies?** Across datasets, synthetic anomalies had the most number of performance gains in MVTec image AD regardless of method. This dataset has the highest dimension and fewest training samples. Here, known anomalies are the least dense, suggesting that the added synthetic anomalies increased the anomaly signal for improved performance.

Of other methods, ES and DROCC are the closest to VC and, likewise, benefit from adding synthetic anomalies. ES-SA outperforms ES in 16/24 (and tied 3) unknown and 7/7 known anomaly categories, while DROCC-SA outperforms DROCC in 19/24 (and tied 3) unknown and 5/7 known anomaly categories. Consistent performance gains demonstrate that adding **synthetic anomalies generalize well to other classifier-based AD methods**.

Meanwhile, ABC and DeepSAD enforce autoencoder (i.e., encoder-decoder) structures. ABC is the next closest to VC, using a binary classification objective with an autoencoder structure. ABC-SA outperforms ABC in 18/24 unknown anomaly categories, but most (14) improvements come from one dataset (MVTec); performance on other datasets are mixed. DeepSAD is the least similar to VC with a two-stage training procedure: first training an autoencoder, then using the encoder for binary classification. DeepSAD-SA outperforms DeepSAD in only 9/24 (and tied 1) unknown anomaly categories and is the only model where adding synthetic anomalies is not better. Notably, DeepSAD has good performance on DoS and probe anomalies in NSL-KDD, which are easy anomalies (Lee & Stolfo, 2000), but struggles on other anomalies (e.g., Thyroid and AdvBench). Here, DeepSAD already underperforms, and adding synthetic anomalies may not be the solution for that. Moreover, only 2/7 known anomaly categories are better for both ABC-SA vs. ABC and DeepSAD-SA vs. DeepSAD, suggesting that synthetic anomalies dilute known anomaly training signal for these autoencoder models.

**Ablations** Due to space constraints, we leave details in Appendix H.5, summarizing ablations across 3 key hyperparameters: width, depth and number of synthetic anomalies (Table 2). Wider and deeper networks provide higher expressivity, but we observe vanishing gradients in the latter. Performance is not as sensitive to width. Meanwhile, more synthetic anomalies (even a small amount amount) improves unknown anomaly performance, but contaminates the supervision signal from known anomalies, hence affecting known anomaly performance. Therefore, in our experiments, we choose depth and width to balance expressivity (determined by data dimension and number of samples), and $n' = n + n^-$.

Table 2: **Ablations for NSL-KDD** for the width $w$, depth $L$ and proportion of synthetic anomalies $n'$ to real training data $r := n + n^-$. AUPR↑ of our vanilla classifier reported across attacks (anomalies).

| Model\Attack | Unknown Anomalies | | | Known |
| | DoS | Probe | RA | PE |
|---|---|---|---|---|
| Random | 0.431 | 0.197 | 0.218 | 0.007 |
| $w = 300$ | 0.756±0.048 | 0.658±0.106 | 0.584±0.025 | 0.309±0.026 |
| $w = 678$ (Ours) | 0.793±0.055 | 0.649±0.078 | 0.609±0.050 | 0.486±0.146 |
| $w = 1500$ | 0.772±0.034 | 0.547±0.068 | 0.602±0.025 | 0.486±0.077 |
| $L = 2$ (Ours) | 0.793±0.055 | 0.649±0.078 | 0.609±0.050 | 0.486±0.146 |
| $L = 3$ | 0.720±0.023 | 0.337±0.063 | 0.581±0.035 | 0.560±0.116 |
| $L = 8$ | 0.431±0.000 | 0.197±0.000 | 0.218±0.000 | 0.007±0.000 |
| $L = 17$ | 0.431±0.000 | 0.197±0.000 | 0.218±0.000 | 0.007±0.000 |
| $n' = 0r$ | 0.345±0.036 | 0.180±0.010 | 0.543±0.019 | 0.609±0.001 |
| $n' = 0.001r$ | 0.744±0.009 | 0.651±0.049 | 0.598±0.030 | 0.513±0.082 |
| $n' = r$ (Ours) | 0.793±0.055 | 0.649±0.078 | 0.609±0.050 | 0.486±0.146 |
| $n' = 5r$ | 0.801±0.026 | 0.649±0.070 | 0.633±0.063 | 0.431±0.158 |
| $n' = 20r$ | 0.763±0.051 | 0.503±0.152 | 0.553±0.087 | 0.298±0.070 |

## 6 Conclusion

Semi-supervised AD paves a way to incorporate known anomalies; we establish the first mathematical formulation of semi-supervised AD, which generalizes the unsupervised setting. Here, synthetic anomalies enable (i) anomaly modeling in low-density regions and (ii) optimal convergence guarantees for neural network classifiers — the first theoretical guarantee for semi-supervised AD. Experiments on five diverse benchmarks reveal consistent performance gains for our theoretical model *and* other classification-based AD methods. These theoretical and empirical results highlight that using synthetic anomalies in AD is generalizable.

ETHICS STATEMENT

All authors have read and confirm that the research conducted in this work adhere to the ICLR Code of Ethics.

REPRODUCIBILITY STATEMENT

We have provided the complete code used for all experiments, along with detailed instructions for running them, in the supplementary material. All datasets used in our experiments are open-sourced, and we have included their sources, descriptions, and preprocessing steps in **Appendix H.1**. For the theoretical results, all assumptions for the theorems and propositions are explicitly stated within their respective statements, and the full proofs are provided in the corresponding sections of the Appendix.

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

## APPENDIX

## A  LIMITATIONS AND EXTENSIONS

We discuss two potential extensions to further our current work.

First, we acknowledge our mathematical formulation is not the only approach to formulate semi-supervised AD. Another intuitive formulation is a constrained optimization problem, such as minimizing the error of estimating the density level set subject to an upper bound on the misclassification on known anomalies. We opted for our mathematical framework because it generalizes unsupervised AD in an elegant manner and admits statistical guarantees. A future research direction is to understand if different approaches of semi-supervised AD are the same or fundamentally different.

Second, our empirical results show improvements on mainly on unsupervised AD methods. Although we do see classification-based methods (VC, ES and DROCC) perform well in general with synthetic anomalies, we do not observe that one AD method is definitively better than the rest. We suspect that different datasets present anomalies that are anomalous in different ways. Future work should understand how synthetic anomalies can complement existing methods, such as understanding how to generate synthetic anomalies more effectively (e.g., DROCC uses synthetic anomalies that are adversarial), along with theoretical guarantees.

## B DISCUSSIONS ON THE BAYES CLASSIFIER, AND THE BAYES RULE MINIMIZING GENERALIZATION ERROR

Recall that $s \in (0,1)$ represents the proportion of normal data on $\mathcal{X}$. For a pair of random variable $(X,Y) \in \mathcal{X} \times \mathcal{Y}$, we define $\mathrm{P}(Y=1) = s$ and $\mathrm{P}(Y=-1) = 1-s$. For any function $f$ on $\mathcal{X} \times \mathcal{Y}$, we have

$$\int_{\mathcal{X} \times \mathcal{Y}} f(X,Y)dP = s \int_{\mathcal{X}} f(X,1)h_1 d\mu + (1-s) \int_{\mathcal{X}} f(X,-1)h_2 d\mu. \tag{11}$$

From (11), we derive the conditional class probability function as

$$\eta(X) := \mathrm{P}(Y=1|X) = \frac{s \cdot h_1(X)}{s \cdot h_1(X) + (1-s) \cdot h_2(X)}. \tag{12}$$

We can also derive the regression function, denoted by $f_P$, given as

$$\begin{aligned}
f_P(X) &:= \mathrm{E}[Y|X] \\
&= 1 \cdot \mathrm{P}(Y=1|X) + (-1) \cdot \mathrm{P}(Y=-1|X) \\
&= \frac{s \cdot h_1(X) - (1-s) \cdot h_2(X)}{s \cdot h_1(X) + (1-s) \cdot h_2(X)}, \qquad \forall X \in \mathcal{X}.
\end{aligned}$$

Both $\eta$ and $f_P$ are fundamental in characterizing the classification problem and assessing the performance of neural network classifiers.

Throughout this work, we adopt the **hinge loss**, defined as

$$\phi(u) := \max\{1-u, 0\},$$

which is one of the most widely used convex loss functions in binary classification. While the $0-1$ loss is the most natural choice for binary classification, minimizing the empirical risk w.r.t. it is computationally intractable (NP-hard) due to its non-convex and discontinuous nature (Bartlett et al., 2006). To overcome this challenge, convex surrogate loss functions $V : \mathcal{X} \times \mathcal{Y} \to [0,\infty)$ are often employed to make computation feasible. Among these, learning a neural network classifier with hinge loss is relatively straightforward owing to the gradient descent algorithm (Molitor et al., 2021; George et al., 2024).

The generalization error for $f : \mathcal{X} \to \mathbb{R}$ w.r.t. the hinge loss $\phi$ and the probability measure $P$ is defined as

$$\begin{aligned}
\varepsilon(f) &= \int_{\mathcal{X} \times \mathcal{Y}} \phi(Yf(X))dP \\
&= \int_{\mathcal{X}} \int_{\mathcal{Y}} \phi(Yf(X))dP(Y|X)dP_{\mathcal{X}} \\
&= \int_{\mathcal{X}} [\phi(f(X))\mathrm{P}(Y=1|X) + \phi(-f(X))\mathrm{P}(Y=-1|X)] \, dP_{\mathcal{X}} \\
&= \int_{\mathcal{X}} [s\phi(f(X))h_1(X) + (1-s)\phi(-f(X))h_2(X)]d\mu.
\end{aligned}$$

The following proposition establishes that the Bayes classifier $f_c$ also minimizes the generalization error. In fact, $f_c$ can be defined explicitly to be

$$f_c(X) = \mathrm{sign}(f_P(X)).$$

Its proof is given below.

**Proposition B.1.** *The generalization error $\varepsilon(f)$ is minimized by the Bayes rule; that is, for all $X \in \mathcal{X}$,*

$$
\begin{aligned}
f_c(X) &= sign(f_P(X)) \\
&= sign(s \cdot h_1(X) - (1-s) \cdot h_2(X)) \\
&= \begin{cases} 1, & \text{if } s \cdot h_1(X) - (1-s) \cdot h_2(X)) \geq 0, \\ -1, & \text{if } s \cdot h_1(X) - (1-s) \cdot h_2(X)) < 0. \end{cases}
\end{aligned}
$$

*Proof of Proposition B.1.* A minimizer $f^*$ of $\varepsilon(f)$ can be found by taking its value $f^*(X)$ at every $X \in \mathcal{X}$ to be a minimum of the convex function $\Phi = \Phi_X$ defined by

$$
\Phi(u) = s \cdot h_1(X)\phi(u) + (1-s) \cdot h_2(X)\phi(-u), \qquad u \in \mathbb{R}. \tag{13}
$$

Notice that the hinge loss $\phi(u) = \max\{1 - u, 0\}$ is not differentiable at $u = 1$. Thus, to find a minimizer of the function $\Phi$, we need to look at its one-sided derivatives.

Let $\phi'_-(u)$ and $\phi'_+(u)$ denote the left derivative and right derivative of $\phi(u)$, respectively. The one-sided derivatives of the hinge loss $\phi(u)$ are

$$
\phi'_-(u) = \begin{cases} -1, & \text{if } u \leq 1, \\ 0, & \text{if } u > 1, \end{cases} \qquad \text{and} \qquad \phi'_+(u) = \begin{cases} -1, & \text{if } u < 1, \\ 0, & \text{if } u \geq 1. \end{cases} \tag{14}
$$

Moreover, the left derivative of $\phi(-u)$ is

$$
\begin{aligned}
\phi'_-(-u) &= \lim_{x \to 0^-} \frac{\phi(-(u+x)) - \phi(-u)}{x} \\
&= \lim_{x \to 0^-} -\frac{\phi(-u-x) - \phi(-u)}{-x} \\
&= -\phi'_+(-u).
\end{aligned}
$$

It follows that the left derivative of the function $\Phi$ is given by

$$
\begin{aligned}
\Phi'_-(u) &= s \cdot h_1(X)\phi'_-(u) + (1-s) \cdot h_2(X)\phi'_-(-u) \\
&= s \cdot h_1(X)\phi'_-(u) - (1-s) \cdot h_2(X)\phi'_+(-u) \\
&= \begin{cases} (1-s) \cdot h_2(X), & \text{if } u > 1, \\ (1-s) \cdot h_2(X) - s \cdot h_1(X), & \text{if } -1 < u \leq 1, \\ -s \cdot h_1(X), & \text{if } u \leq -1. \end{cases}
\end{aligned}
$$

Similarly, the right derivative of the function $\Phi$ is given by

$$
\begin{aligned}
\Phi'_+(u) &= s \cdot h_1(X)\phi'_+(u) + (1-s) \cdot h_2(X)\phi'_+(-u) \\
&= s \cdot h_1(X)\phi'_+(u) - (1-s) \cdot h_2(X)\phi'_-(-u) \\
&= \begin{cases} (1-s) \cdot h_2(X), & \text{if } u \geq 1, \\ (1-s) \cdot h_2(X) - s \cdot h_1(X), & \text{if } -1 \leq u < 1, \\ -s \cdot h_1(X), & \text{if } u < -1. \end{cases}
\end{aligned}
$$

It follows that a minimizer $u^*$ of $\Phi$ must be on $[-1, 1]$. Also, the convexity of $\Phi$ implies that a minimizer $u^*$ must satisfies

$$
\Phi'_-(u^*) = \begin{cases} (1-s) \cdot h_2(X) - s \cdot h_1(X) \leq 0, & \text{if } -1 < u^* \leq 1, \\ -s \cdot h_1(X) \leq 0, & \text{if } u^* = -1. \end{cases} \tag{15}
$$

and

$$
\Phi'_+(u^*) = \begin{cases} (1-s) \cdot h_2(X) \geq 0, & \text{if } u^* = 1, \\ (1-s) \cdot h_2(X) - s \cdot h_1(X) \geq 0, & \text{if } -1 \leq u^* < 1. \end{cases} \tag{16}
$$

We consider all the possible cases:

- If $(1-s) \cdot h_2(X) - s \cdot h_1(X) < 0$, the minimizer $u^*$ must be 1; this is because $(1-s) \cdot h_2(X) - s \cdot h_1(X) \geq 0$ for all $u^* \in [-1, 1)$.

- Similarly, if $(1-s) \cdot h_2(X) - s \cdot h_1(X) > 0$, the minimizer $u^*$ must be $-1$; this is because $(1-s) \cdot h_2(X) - s \cdot h_1(X) \leq 0$ for all $u^* \in (-1, 1]$.

- If $(1-s) \cdot h_2(X) - s \cdot h_1(X) = 0$, the function $\Phi$ is just constant on $[-1, 1]$ so a minimizer $u^*$ can be any number on $[-1, 1]$. We take $u^* = 1$.

This completes the proof. $\qquad\square$

## C  REMARKS ABOUT THEORETICAL ANALYSIS

### C.1  REMARKS ABOUT THE TSYBAKOV'S NOISE CONDITION

Here, we provide a detailed discussion and elaboration on the well-known Tsybakov noise condition, which is utilized in this work to model the classification problem.

*Remark* C.1. Throughout this work, we assume the Tsybakov noise condition (see, e.g., (Mammen & Tsybakov, 1999; Tsybakov, 2004)), which is a common assumption imposed in the literature in classification. This condition describes the behavior of the regression function $f_P(X) = \mathrm{E}[Y|X]$ (with parameter $q$) around the decision boundary $\{x : f_P(x) = 0\}$. Specifically, it assumes that for some constant $c_0 > 0$ and $q \in [0, \infty)$,

$$\mathrm{P}_X(\{X \in \mathcal{X} : |f_P(X)| \leq t\}) \leq c_0 t^q, \qquad \forall t > 0, \tag{17}$$

where $q$ is commonly referred to as the noise exponent. Intuitively, bigger $q$ means there is a higher chance that $f_P(X)$ is bounded away from 0, which is favorable for classification. If $q$ approaches $\infty$, the regression function $f_P$ is bounded away from 0 for $P_X$-almost all $x \in \mathcal{X}$. Conversely, smaller $q$ implies that there is a plateau behavior near the boundary 0, which is considered a difficult situation for classification. Note that the assumption becomes trivial for $q = 0$ because every probability measure satisfies (17) by taking the constant $c_0 = 1$.

Equivalently, the noise condition can be stated in terms of $\eta(X) = \mathrm{P}(Y = 1|X)$ as

$$\mathrm{P}_X(\{X \in \mathcal{X} : |2\eta(X) - 1| \leq t\}) \leq c_0 t^q. \tag{18}$$

The statements (17) and (18) are equivalent because

$$f_P(X) = \eta(X) - (1 - \eta(X)) = 2\eta(X) - 1.$$

See (Tsybakov, 2004; Audibert & Tsybakov, 2007; Bartlett et al., 2006) for further discussion on this assumption.

### C.2  REMARKS ABOUT THE "APPROX-SIGN" FUNCTION

*Remark* C.2. Recall that $\sigma^k$ is the ReLU$^k$ activation function with $k \in \mathbb{N}$, which is the generalization of ReLU defined as

$$\sigma^k(x) = (\max\{0, x\})^k.$$

These higher-order variants of the ReLU functions preserve the non-negativity and sparsity of ReLU while introducing smoother transitions near the origin. See Figure 4 for a plot comparing $\sigma(x)$, $\sigma^2(x)$, and $\sigma^3(x)$.

Recall the "approx-sign" function $\sigma_\tau : \mathbb{R} \to [0, 1]$, defined previously in equation (7) with a bandwidth parameter $\tau > 0$, as

$$\sigma_\tau(x) = \frac{1}{\tau}\sigma(x) - \frac{1}{\tau}\sigma(x - \tau) - \frac{1}{\tau}\sigma(-x) + \frac{1}{\tau}\sigma(-x - \tau),$$

which simplifies to the piecewise form:

$$\sigma_\tau(x) = \begin{cases} 1, & \text{if } x \geq \tau, \\ \frac{x}{\tau}, & \text{if } x \in [-\tau, \tau), \\ -1, & \text{if } x < -\tau. \end{cases}$$

We see that this function is a linear combination of four scaled ReLU units. Moreover, it is a smoothed sign function that transitions linearly between $-1$ and $1$ within the interval $[-\tau, \tau)$. Outside this

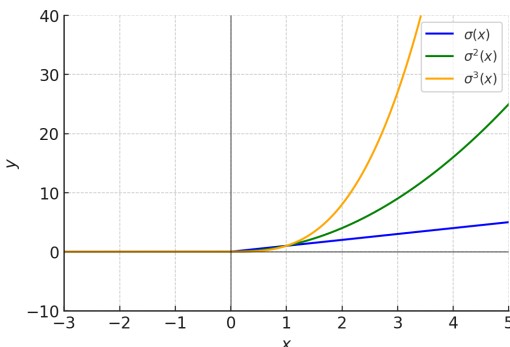

Figure 4: Plot of the ReLU function $\sigma(x) = \max(0, x)$ and its squared and cubed variants, $\sigma^2(x)$ and $\sigma^3(x)$, illustrating their increasing smoothness and growth rates for $x > 0$.

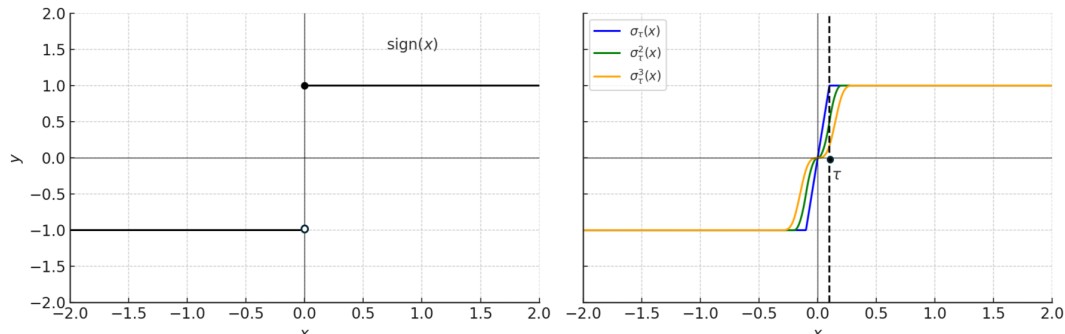

Figure 5: Comparison of the sign function and "approx-sign" functions. smoothed localized approximations. The left-hand plot shows the discontinuous sign function, defined as $\text{sign}(x) = -1$ for $x < 0$ and $1$ for $x \geq 0$. The right-hand plot displays the functions "approx-sign" functions: $\sigma_\tau(x)$, $\sigma_\tau^2(x)$, and $\sigma_\tau^3(x)$. These functions are considered the smoothed localized approximations of the sign function. They are constructed using combinations of scaled ReLU $\sigma(x)$ and their powers $\sigma^2(x), \sigma^3(x)$ to create increasingly smooth and localized bump-like functions. In this example, we set $\tau = 0.1$.

interval, it behaves like the standard sign function. As $\tau \to 0$, $\sigma_\tau(x)$ converges to the sign function. We use the function $\sigma_\tau$ to approximate the sign function.

We also define higher-order approximations $\sigma_\tau^k(x)$, which are smoother versions of $\sigma_\tau(x)$, constructed using $\sigma^k$ functions for $k \in \mathbb{N}$. Specifically, the function $\sigma_\tau^k(x)$ is defined as

$$\sigma_\tau^k(x) := \frac{1}{k!\tau^k} \sum_{\ell=0}^{k} (-1)^\ell \binom{k}{\ell} \sigma^k(x - \ell\tau) - \frac{1}{k!\tau^k} \sum_{\ell=0}^{k} (-1)^\ell \binom{k}{\ell} \sigma^k(-x - \ell\tau).$$

See Figure 5 for visual comparison using $\tau = 0.1$.

## C.3 SIGNIFICANCE AND IMPLICATIONS OF THEOREM 3.3

*Remark* C.3. Here, we discuss and highlight the novelty of Theorem 3.3. In binary classification, the generalization error $\varepsilon(f)$ is minimized by the Bayes rule (as shown in Proposition B.1). The Bayes classifier $f_c = \text{sign}(f_P)$ is typically discontinuous.

Existing literature often estimates the excess generalization error $\varepsilon(f) - \varepsilon(f_c)$ by the $L_1$ norm (w.r.t. marginal distribution on $\mathcal{X}$) of the difference between $f$ and $f_c$ (Chen et al., 2004; Lin et al., 2017). However, due to the discontinuous nature of $f_c$, the $L_1$ error usually decays slowly to zero.

In contrast, Theorem 3.3 shows that as long as the error in approximating the regression function $f_P$ (instead of $f_c$) converges fast to zero, the excess generalization error (and consequently the excess risk $R(f) - R(f_c)$ will also converge fast to zero.

Our results leverage Tsybakov's noise condition to establish a relationship between the excess generalization error of $f$ and the approximation error $\|f - f_P\|_{L^\infty}$. Since the regression function $f_P$ is smooth (particularly when synthetic anomalies are incorporated during training), errors of approximating this smooth regression function can be small and decay fast, as supported by many existing results in approximation theory (e.g., (Yarotsky, 2017; Shen et al., 2022)). This novel approach offers an alternative estimate of the excess generalization error in classification, leveraging on both the noise condition and the smoothness of the regression function.

## D  EXAMPLES

In this section, we present two examples, **each based on specific choices of the density functions $h_1$ and $h_-$, to illustrate two key points: (i) how the resulting regression function $f_P$ can be discontinuous, and (ii) how the inclusion of synthetic anomalies can improve the smoothness of $f_P$.**

*Example* D.1.  Here, let us consider a simple example of $h_1, h_-$ and the corresponding regression function.

Let $H$ be the univariate hat function on $[-1, 1]$ given by

$$H(x) = \max\{1 - |x|, 0\},$$

where it peaks at $x = 0$ and linearly decreases to 0 at $x = -1$ and $x = 1$. We let

$$h_1(X) = 4H(4X - 3)$$

and

$$h_-(X) = 4H(4X - 1)$$

be two hat functions supported on $[0, 1/2]$ and $[1/2, 1]$ respectively. From Figure 6, we see that both functions are exactly zero outside their respective supports. Also, the intersection of $[0, 1/2]$ and $[1/2, 1]$ has an empty interior. This presents a specific example that corresponds to the general scenarios illustrated in Proposition 3.1.

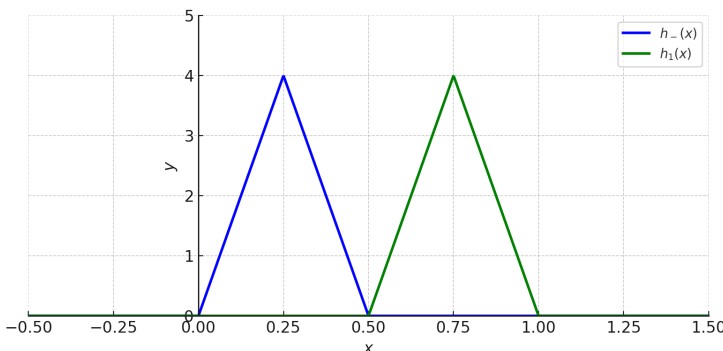

Figure 6: The plot showcases the density functions $h_-$ and $h_1$ given in Example D.1, which are two hat functions supported on $[0, 1/2]$ and $[1/2, 1]$, respectively.

When $\tilde{s} = 0$ without synthetic data (i.e., $h_2(X) = h_-(X)$), the regression function is given by

$$f_P(X) = \begin{cases} -1, & \text{if } X \in [0, 1/2], \\ 1, & \text{if } X \in (1/2, 1], \end{cases}$$

which is a function discontinuous on $[0, 1]$.

On the other hand, if we include synthstic anomalies in training and let $h_2(X) = \tilde{s}h_-(X) + (1 - \tilde{s})$ for some $\tilde{s} > 0$, the regression function becomes

$$f_P(X) = \begin{cases} -1, & \text{if } X \in [0, 1/2], \\ \frac{16s(X-1/2)-(1-s)\tilde{s}}{16s(X-1/2)+(1-s)\tilde{s}}, & \text{if } X \in (1/2, 3/4], \\ \frac{16s(1-X)-(1-s)\tilde{s}}{16s(1-X)+(1-s)\tilde{s}}, & \text{if } X \in (3/4, 1], \end{cases}$$

which is continuous. In particular, observe that for $X \in (1/2, 3/4)$,

$$f_P'(X) = \frac{32s(1-s)\tilde{s}}{\left(16s(X-1/2) + (1-s)\tilde{s}\right)^2}$$

and the Lipschitz constant ($C^1$ seminorm) of $f_P$ equals $\frac{32s}{(1-s)\tilde{s}}$. We can see that the regularity of $f_P$ gets better as $\tilde{s}$ becomes bigger. Figure 7 provides a visual illustration of the resulting regression function for various choices of $\tilde{s} > 0$. We can see that the larger the $\tilde{s}$, the smoother the $f_P$.

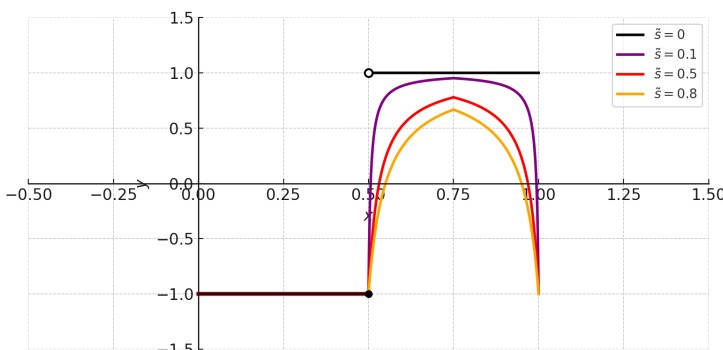

Figure 7: A comparison of the regression function $f_P(X)$ for varying parameter values $\tilde{s} = 0, \tilde{s} = 0.1, \tilde{s} = 0.5, \tilde{s} = 0.8$. The case $\tilde{s} = 0$ corresponds to the absence of synthetic anomalies and results in a discontinuous function with a jump from $-1$ to $1$. Increasing the parameter $\tilde{s}$ (i.e., adding more synthetic anomalies) results in progressively smoother functions. Here, we take $s = 0.5$.

The above example considers one-dimensional density functions. We now proceed to present an example involving $d$-dimensional density functions, demonstrating how the inclusion of synthetic anomalies can improve the smoothness of the resulting regression function. The example is given below.

*Example* D.2. Consider the univariate hat function defined on the interval $[-1/2, 1/2]$ given by

$$\max\{2 - 4|x|, 0\}, \tag{19}$$

where it peaks at $x = 0$ and linearly decreases to 0 at $x = -1/2$ and $x = 1/2$. Now consider a $d$-dimensional function where each dimension is a hat function, that is,

$$H(X) = H(X_1, \cdots, X_d) = \prod_{i=1}^{d} H(X_i) = \prod_{i=1}^{d} \max\{2 - 4|X_i|, 0\}. \tag{20}$$

This function is non-zero only within the region $X_i \in [-1/2, 1/2]$ for all $i$, and smoothly decays to zero as any $|X_i|$ approaches $1/2$. We can easily verify that this function is Lipschitz continuous (i.e., $C^1$ continuous).

Suppose the normal density function $h_1$ is given by

$$h_1(X) = H(X + (1/2, 0, 0, \cdots, 0)), \tag{21}$$

and suppose the real anomaly density function $h_-$ is given by

$$h_-(X) = H(X - (1/2, 0, 0, \cdots, 0)). \tag{22}$$

If $h_2 = h_-$ without synthetic data, the regression function is

$$
\begin{aligned}
f_P(X) &= \frac{s \cdot h_1(X) - (1-s) \cdot h_-(X)}{s \cdot h_1(X) + (1-s) \cdot h_-(X)} \\
&= \begin{cases} \frac{s \cdot h_1(X)}{s \cdot h_1(X)} = 1, & \text{if } X \in [-1/2, 0) \times [-1/2, 1/2]^{d-1}, \\ -\frac{(1-s) \cdot h_-(X)}{(1-s) \cdot h_-(X)} = -1, & \text{if } X \in (0, 1/2] \times [-1/2, 1/2]^{d-1}. \end{cases}
\end{aligned}
$$

We can see that $f_P$ is not continuous on the line segment $X_1 = 0$. With the choice $h_2 = h_-$, the regression function has lost its Lipschitz continuity. Learning a discontinuous function by neural network is generally difficult in theory. However, if we include synthetic anomalies and set $h_2 = \tilde{s} h_- + (1 - \tilde{s})$ for $\tilde{s} > 0$, then the regression function becomes

$$
f_P(X) = \frac{s \cdot h_1(X) - (1-s) \cdot h_-(X) - (1-s)(1-\tilde{s})}{s \cdot h_1(X) + (1-s) \cdot h_-(X) + (1-s)(1-\tilde{s})}
$$

We can verify that this regression function is Lipschitz continuous. In fact, because we have the positive constant term $(1-s)(1-\tilde{s})$ in the denominator, it helps in maintaining the smoothness of $f_P$.

## E    PROOFS OF PROPOSITION AND EQUATION

We present the proofs of Proposition and Equation in this section.

### E.1    PROOF OF EQUATION (4)

We first present the following lemma that showcases how the Tsybakov noise condition relates to our data distribution on the domain $\mathcal{X}$. This lemma will be instrumental in proving the statement of Equation (4).

**Lemma E.1.** *Let $\mu$ be a known probability measure on $\mathcal{X}$. Let $Q$ and $W$ be distributions on $\mathcal{X}$ such that $Q$ has a density $h_1$ with respect to $\mu$, and $W$ has a density $h_2$ with respect to $\mu$. For $s \in (0, 1)$, the Tsybakov noise condition (3) with noise exponent $q \in [0, \infty)$ is satisfied if and only if*

$$
\int_{\left\{ X \in \mathcal{X}: \left| \frac{h_1(X)}{h_2(X)} - \frac{1-s}{s} \right| \le \left( \frac{1-s}{s} \right) t \right\}} h_2(X) d\mu = \mathcal{O}(t^q). \tag{23}
$$

*In other words, $\frac{h_1}{h_2}$ has $\frac{1-s}{s}$-exponent $q$.*

*Proof of Lemma E.1.* In our problem setting, we observe that the regression function depends on both densities $h_1$ and $h_2$. For $X$ with $h_2(X) > 0$, the regression function can be expressed as

$$
f_P(X) = \frac{s \cdot \frac{h_1(X)}{h_2(X)} - (1-s)}{s \cdot \frac{h_1(X)}{h_2(X)} + (1-s)}.
$$

Note that the case $h_2(X) = 0$ means $f_P(X) = 1$ and the point $X$ is outside the domain stated in (3). It follows that for $t \in (0, 1/4)$, $|f_P(X)| \le t$ if and only if

$$
\frac{1-s}{s} \cdot \frac{1-t}{1+t} \le \frac{h_1(X)}{h_2(X)} \le \frac{1-s}{s} \cdot \frac{1+t}{1-t}, \tag{24}
$$

which is the same as

$$
-\frac{1-s}{s} \cdot \frac{2t}{1+t} \le \frac{h_1(X)}{h_2(X)} - \frac{1-s}{s} \le \frac{1-s}{s} \cdot \frac{2t}{1-t}. \tag{25}
$$

Thus, for $t \in (0, 1/4)$, we have

$$
\begin{aligned}
P_X(\{X \in \mathcal{X} : |f_\rho(X)| \le t\}) &= \int_{\{X \in \mathcal{X}: |f_\rho(X)| \le t\}} dP_X \\
&= \int_{X_t} [s \cdot h_1(X) + (1-s) \cdot h_2(X)] d\mu \\
&= \int_{X_t} \left( s \cdot \frac{h_1(X)}{h_2(X)} + (1-s) \right) h_2(X) d\mu,
\end{aligned}
$$

where $X_t := \left\{ X \in \mathcal{X} : -\frac{1-s}{s} \cdot \frac{2t}{1+t} \leq \frac{h_1(X)}{h_2(X)} - \frac{1-s}{s} \leq \frac{1-s}{s} \cdot \frac{2t}{1-t} \right\}$.

Then, we can get an upper bound for the measure by observing

$$s \cdot \frac{h_1(X)}{h_2(X)} \leq (1-s)\frac{1+t}{1-t} \leq (1-s)\frac{1+1/4}{1-1/4} = \frac{5(1-s)}{3} \leq 2(1-s);$$

on $X_t$ and enlarging the domain $X_t$ as

$$
\begin{aligned}
\mathrm{P}_X(\{X \in \mathcal{X} : |f_\rho(X)| \leq t\}) &= \int_{X_t} \left( s \cdot \frac{h_1(X)}{h_2(X)} + (1-s) \right) h_2(X)d\mu \\
&\leq 3(1-s) \int_{\left\{ X \in \mathcal{X} : \left| \frac{h_1(X)}{h_2(X)} - \frac{1-s}{s} \right| \leq \frac{1-s}{s} 3t \right\}} h_2(X)d\mu.
\end{aligned}
$$

A lower bound can be derived by observing

$$s \cdot \frac{h_1(X)}{h_2(X)} \geq (1-s)\frac{1-t}{1+t} \geq (1-s)(1-1/4) = \frac{3(1-s)}{4}$$

on $X_t$ and reducing the domain $X_t$ as

$$
\begin{aligned}
\mathrm{P}_X(\{X \in \mathcal{X} : |f_\rho(X)| \leq t\}) &= \int_{X_t} \left( s \cdot \frac{h_1(X)}{h_2(X)} + (1-s) \right) h_2(X)d\mu \\
&\geq \frac{3(1-s)}{2} \int_{\left\{ X \in \mathcal{X} : \left| \frac{h_1(X)}{h_2(X)} - \frac{1-s}{s} \right| \leq \frac{1-s}{s} 2t \right\}} h_2(X)d\mu.
\end{aligned}
$$

The proof of Lemma E.1 is complete. $\qquad\square$

With $C_q = \frac{c_0^{1/q}}{2q}(q+1)^{1+1/q}$, Equation (4) follows after Lemma E.1 and Proposition 1 from (Tsybakov, 2004).

## E.2    PROOF OF PROPOSITION 3.1

*Proof of Proposition 3.1.* Recall that in Proposition 3.1, we consider the domain $\mathcal{X}$ to be a union of two closed subdomains $\mathcal{X}_1$ and $\mathcal{X}_-$ such that the intersection of $\mathcal{X}_1$ and $\mathcal{X}_-$ is nonempty and has an empty interior. Suppose $h_1$ vanishes on $\mathcal{X}_-$ but its support equals $\mathcal{X}_1$ (i.e., $h_1 > 0$ on interior($\mathcal{X}_1$)), and $h_-$ vanishes on $\mathcal{X}_1$ but its support equals $\mathcal{X}_-$ (i.e., $h_- > 0$ on interior($\mathcal{X}_-$)).

If no synthetic anomalies are generated, the resulting regression function $f_P$ is given as

$$
\begin{aligned}
f_P(X) &= \frac{s \cdot h_1(X) - (1-s) \cdot h_-(X)}{s \cdot h_1(X) + (1-s) \cdot h_-(X)} \\
&= \begin{cases} \frac{h_1(X)}{h_1(X)} = 1, & \text{if } X \in \text{interior}(\mathcal{X}_1), \\ -\frac{h_-(X)}{h_-(X)} = -1, & \text{if } X \in \text{interior}(\mathcal{X}_-). \end{cases}
\end{aligned}
$$

In particular, we notice that $f_P$ is discontinuous at the intersection of $\mathcal{X}_1$ and $\mathcal{X}_-$ (there is a jump from 1 to $-1$). Take an arbitrary point $X^*$ in this intersection.

For any continuous function $f : \mathcal{X} \to \mathbb{R}$, the function $f$ takes its value $f(X^*)$ at $X^*$. If $f(X^*) \geq 0$, we take a sequence $\{X^{(n)}\}_{n \in \mathbb{N}}$ in interior($\mathcal{X}_1$) tending to $X^*$, then

$$\lim_{n \to \infty} \left( f\left(X^{(n)}\right) - f_P\left(X^{(n)}\right) \right) = f(X^*) - (-1) \geq 1.$$

In the same way, if $f(X^*) < 0$, we have another sequence $\{X^{(n)}\}_{n \in \mathbb{N}}$ approaching $X^*$ such that

$$\lim_{n \to \infty} \left( f\left(X^{(n)}\right) - f_P\left(X^{(n)}\right) \right) = f(X^*) - 1 < -1.$$

Then $\|f - f_P\|_{L^\infty[0,1]^d} \geq 1$. This concludes the proof. $\qquad\square$

## F    PROOF OF THEOREM 3.3

We present the proof of Theorem 3.3 in this section.

*Proof of Theorem 3.3.* The $k$-order forward difference of a function $g$ is defined as

$$\Delta_\tau^k g(x) = \sum_{\ell=0}^k (-1)^{k-\ell} \binom{k}{\ell} g(x + \ell\tau),$$

which can be written as an integral sum

$$\Delta_\tau^k g(x) = \int_0^\tau \cdots \int_0^\tau g^{(k)}(x + t_1 + \cdots + t_k) dt_1 \cdots dt_k$$

Recall the ReLU$^k$ function given by $\sigma^k(x) = (\max\{0, x\})^k$. Also recall the function $\sigma_\tau^k$ defined for some $0 < \tau \le 1$ as

$$\sigma_\tau^k(x) \;\; = \;\; \frac{1}{k!\tau^k} \sum_{\ell=0}^k (-1)^\ell \binom{k}{\ell} \sigma^k(x - \ell\tau) - \frac{1}{k!\tau^k} \sum_{\ell=0}^k (-1)^\ell \binom{k}{\ell} \sigma^k(-x - \ell\tau).$$

Note that the $k$-th derivative of the ReLU$^k$ function is

$$(\sigma^k)^{(k)}(x) = \begin{cases} k!, & \text{if } x > 0, \\ 0, & \text{if } x < 0. \end{cases}$$

If $x \ge 0$, we know $\sigma^k(-x - \ell\tau) = 0$ for every $\ell \in \mathbb{N}$ and $0 < \tau \le 1$. Thus, for $x \ge 0$, we have

$$\begin{aligned}
\sigma_\tau^k(x) \;\; &= \;\; \frac{1}{k!\tau^k} \sum_{\ell=0}^k (-1)^\ell \binom{k}{\ell} \sigma^k(x - \ell\tau) \\
&= \;\; \frac{1}{k!\tau^k} \sum_{\ell=0}^k (-1)^{k-\ell} \binom{k}{k-\ell} \sigma^k(x - k\tau + \ell\tau) \\
&= \;\; \frac{1}{k!\tau^k} \Delta_\tau^k (\sigma^k)(x - k\tau) \\
&= \;\; \frac{1}{k!\tau^k} \int_0^\tau \cdots \int_0^\tau (\sigma^k)^{(k)}(x - k\tau + t_1 + \cdots + t_k) dt_1 \cdots dt_k \\
&= \;\; \frac{1}{\tau^k} \int_0^\tau \cdots \int_0^\tau 1_{\{t_1 + \cdots + t_k > k\tau - x\}} dt_1 \cdots dt_k.
\end{aligned}$$

We can see from the above expression that $\sigma_\tau^k(0) = 0$, $\sigma_\tau^k(x) = 1$ for $x \ge k\tau$, and $\sigma_\tau^k(x)$ is strictly increasing for $x \in [0, k\tau]$. Similarly, for $x < 0$, we know that $\sigma_\tau^k(x) = -1$ for $x \le -k\tau$, and $\sigma_\tau^k(x)$ is strictly increasing for $x \in [-k\tau, 0]$. Then,

$$\sigma_\tau^k(x) = \begin{cases} 1, & \text{if } x \ge k\tau, \\ -1, & \text{if } x \le -k\tau \end{cases}$$

and $|\sigma_\tau^k(x)| \le 1$ for $x \in (-k\tau, k\tau)$. Moreover, $\text{sign}(\sigma_\tau^k(x)) = \text{sign}(x)$.

Notice that the convex Hinge loss $\phi(x) = \max\{0, 1-x\}$ is Lipschitz continuous on $\mathbb{R}$ with Lipschitz constant 1 because $|\phi(x_1) - \phi(x_2)| \le |x_1 - x_2|$ for all $x_1, x_2 \in \mathbb{R}$.

For any function $g$ such that $|g(X)| \le 1$, we have $\phi(Yg(X)) = 1 - Yg(X)$, and the excess generalization error between such a $g$ and the Bayes classifier $f_c$ is

$$
\begin{aligned}
\varepsilon(g) - \varepsilon(f_c) &= \int_{\mathcal{X}} \int_{\mathcal{Y}} 1 - Yg(X) - (1 - Yg_c(X)) d\mathrm{P}(Y|X) dP_{\mathcal{X}} \\
&= \int_{\mathcal{X}} \int_{\mathcal{Y}} -Y(g(X) - f_c(X)) d\mathrm{P}(Y|X) dP_{\mathcal{X}} \\
&= \int_{\mathcal{X}} (g(X) - f_c(X)) \int_{\mathcal{Y}} -Y d\mathrm{P}(Y|X) dP_{\mathcal{X}} \\
&= \int_{\mathcal{X}} (f_c(X) - g(X))(1 \cdot \mathrm{P}(Y = 1|X) - 1 \cdot \mathrm{P}(Y = -1|X)) dP_{\mathcal{X}} \\
&= \int_{\mathcal{X}} (f_c(X) - g(X))(2\eta(X) - 1) dP_{\mathcal{X}}.
\end{aligned}
$$

Notice that $2\eta(X) - 1 > 0$ if and only if $f_c(X) = \mathrm{sign}(f_P(X)) = \mathrm{sign}(2\eta(X) - 1)$, so we have

$$
\begin{aligned}
\varepsilon(g) - \varepsilon(f_c) &= \int_{\mathcal{X}} |g(X) - f_c(X)||2\eta(X) - 1| dP_{\mathcal{X}} \\
&= \int_{\mathcal{X}} |g(X) - f_c(X)||f_P(X)| dP_{\mathcal{X}}.
\end{aligned}
$$

Now take $g = \sigma_\tau^k(f)$. Let $\omega := \|f - f_P\|_{L^\infty[0,1]^d}$. We consider two cases separately: (i) $|f_P(X)| \le \omega$ and (ii) $|f_P(X)| > \omega$. Note that

$$
|\sigma_\tau^k(f(X)) - f_c(X)| \le |\sigma_\tau^k(f(X))| + |f_c(X)| \le 1 + 1 = 2, \qquad \forall X \in \mathcal{X}. \tag{26}
$$

For the case $|f_P(X)| \le \omega$, it follows from the Tsybakov noise condition (17) (with $q \in [0, \infty)$ and constant $c_0 > 0$) that

$$
\begin{aligned}
&\int_{\{x \in \mathcal{X} : |f_P(X)| \le \omega\}} |\sigma_\tau^k(f(X)) - f_c(X)||f_P(X)| dP_{\mathcal{X}} \\
&\overset{(26)}{\le} 2\omega \int_{\{x \in \mathcal{X} : |f_P(X)| \le \omega\}} dP_{\mathcal{X}} \\
&= 2\omega \cdot \mathrm{P}_X(\{X \in \mathcal{X} : |f_P(X)| \le \omega\}) \\
&\overset{(17)}{\le} 2\omega \cdot c_0 \omega^q \\
&= 2c_0 \omega^{q+1}.
\end{aligned}
$$

Next, for the other case $|f_P(X)| > \omega$, since $|f(X) - f_P(X)| \le \omega$ for all $X \in \mathcal{X}$, we know

$$
\mathrm{sign}(\sigma_\tau^k(f(X))) = \mathrm{sign}(f(X)) = \mathrm{sign}(f_P(X)) = f_c(X).
$$

If we have $\sigma_\tau^k(f(X)) \in \{-1, 1\}$, then

$$
|\sigma_\tau^k(f(X)) - f_c(X)| = 0 \tag{27}
$$

must holds. Also, if $|\sigma_\tau^k(f(X))| < 1$, then

$$
|f(X)| < k\tau \tag{28}
$$

and

$$
|f_P(X)| = |f_P(X) - f(X) + f(X)| \le \omega + |f(X)| \le \omega + k\tau. \tag{29}
$$

Thus, we have

$$\int_{\{x \in \mathcal{X} : |f_P(X)| > \omega\}} \left| \sigma_\tau^k \left( f(X) \right) - f_c(X) \right| |f_P(X)| dP_{\mathcal{X}}$$

$$\stackrel{(27)}{=} \int_{\{x \in \mathcal{X} : |f_P(X)| > \omega, |\sigma_\tau^k(f(X))| < 1\}} \left| \sigma_\tau^k \left( f(X) \right) - f_c(X) \right| |f_P(X)| dP_{\mathcal{X}}$$

$$\stackrel{(26),(29)}{\leq} 2(k\tau + \omega) \mathrm{P}_X \left( \left\{ x \in \mathcal{X} : |f_P(X)| > \omega, |\sigma_\tau^k \left( f(X) \right)| < 1 \right\} \right)$$

$$\leq 2(k\tau + \omega) \mathrm{P}_X \left( \left\{ x \in \mathcal{X} : |\sigma_\tau^k \left( f(X) \right)| < 1 \right\} \right)$$

$$\stackrel{(28)}{=} 2(k\tau + \omega) \mathrm{P}_X \left( \left\{ x \in \mathcal{X} : |f(X)| < k\tau \right\} \right)$$

$$\leq 2(k\tau + \omega) \mathrm{P}_X \left( \left\{ x \in \mathcal{X} : |f_P(X)| < k\tau + \omega \right\} \right)$$

$$\stackrel{(17)}{\leq} 2(k\tau + \omega) \cdot c_0 (k\tau + \omega)^q$$

$$= 2c_0 (k\tau + \omega)^{q+1}.$$

Now combining the above error bounds, we get

$$\varepsilon \left( \sigma_\tau^k \left( f(X) \right) \right) - \varepsilon(f_c)$$

$$= \int_{\mathcal{X}} \left| \sigma_\tau^k \left( f(X) \right) - f_c(X) \right| |f_P(X)| dP_{\mathcal{X}}$$

$$= \int_{\{x \in \mathcal{X} : |f_P(X)| \leq \omega\}} \left| \sigma_\tau^k \left( f(X) \right) - f_c(X) \right| |f_P(X)| dP_{\mathcal{X}}$$

$$+ \int_{\{x \in \mathcal{X} : |f_P(X)| > \omega\}} \left| \sigma_\tau^k \left( f(X) \right) - f_c(X) \right| |f_P(X)| dP_{\mathcal{X}}$$

$$\leq 2c_0 \omega^{q+1} + 2c_0 (k\tau + \omega)^{q+1}$$

$$\leq 4c_0 (k\tau + \omega)^{q+1}$$

$$= 4c_0 \left( k\tau + \|f - f_P\|_{L^\infty [0,1]^d} \right)^{q+1}.$$

The proof follows from the well-known comparison theorem in classification (Zhang, 2004) asserts that for the Hinge loss $\phi$ and any measurable function $f : \mathcal{X} \to \mathbb{R}$, the following inequality holds:

$$\underbrace{R(f) - R(f_c)}_{\text{excess risk}} \leq \underbrace{\varepsilon(f) - \varepsilon(f_c)}_{\text{excess generalization error}} . \tag{30}$$

$\square$

# G    PROOF OF THEOREM 4.5

Let $T = \{(X_i, 1)\}_{i=1}^n$ be the set of real normal data given, where each $X_i$ are drawn i.i.d. from an unknown distribution $Q$. Let $T^- = \{(X_i^-, -1)\}_{i=1}^{n^-}$ be the set of real anomalies given, where each $X_i^-$ are drawn i.i.d. from another unknown distribution $W$. Additionally, let $T' = \{(X_i', -1)\}_{i=1}^{n'}$ be the set of synthetic anomalies generated from $\mu = \text{Uniform}(\mathcal{X})$. We merge these data together $T \cup T^- \cup T' = \{(X_i, 1)\}_{i=1}^n \cup \{(X_i^-, -1)\}_{i=1}^{n^-} \cup \{(X_i', -1)\}_{i=1}^{n'}$ to train the ReLU network classifier. Theorem 4.5 proves that as the number of training data $(n, n^-, n')$ increases, the ReLU network classifier achieves a theoretically grounded accuracy in anomaly detection.

## G.1    EXPLICIT EXCESS RISK BOUND

Before presenting the proof, we restate the theorem with an explicit excess risk bound, along with the corresponding parameters $L^*, w^*, v^*, K^*$.

Recall our hypothesis space $\mathcal{H}_\tau$ defined in Definition 4.3 with some $0 < \tau \le 1$. Also recall the empirical risk minimizer w.r.t. Hinge loss $\phi$ trained on $T \cup T^- \cup T'$:

$$
\begin{aligned}
&f_{\text{ERM}} \\
=\ &\arg\min_{f \in \mathcal{H}_\tau} \varepsilon_{T,T^-,T'}(f) \\
=\ &\arg\min_{f \in \mathcal{H}_\tau} \left[ \frac{s}{n} \sum_{i=1}^n \phi\left(1 \cdot f(X_i)\right) + \frac{(1-s)\tilde{s}}{n^-} \sum_{i=1}^{n^-} \phi(-1 \cdot f(X_i^-)) + \frac{(1-s)(1-\tilde{s})}{n'} \sum_{i=1}^{n'} \phi(-1 \cdot f(X_i')) \right].
\end{aligned}
$$

**Theorem G.1** (Restatement of Theorem 4.5). *Let $n, n^-, n' \ge 3, n_{min} = \min\{n, n^-, n'\}, d \in \mathbb{N}, \alpha > 0$. Let $m \ge 1$ be an integer. Assume the Tsybakov noise condition holds for some noise exponent $q \in [0, \infty)$ and constant $c_0 > 0$. Consider the hypothesis space $\mathcal{H}_\tau$ with $N = \left\lceil \left( \frac{n_{min}}{(\log(n_{min})^4} \right)^{\frac{d}{d+\alpha(q+2)}} \right\rceil$, $\tau = N^{-\frac{\alpha}{d}}, K^* = 1, L^* = 8 + (m+5)(1 + \lceil \log_2(\max\{d, \alpha\}) \rceil), w^* = 6(d + \lceil \alpha \rceil)N, v^* = 141(d + \alpha + 1)^{3+d}N(m+6)$.*

*For any $0 < \delta < 1$, with probability $1 - \delta$, there holds,*

$$
R(sign(f_{\text{ERM}})) - R(f_c) \le C \left( \log\left( \frac{6}{\delta} \max\{n, n^-, n'\} \right) \right) \left( \frac{(\log n_{min})^4}{n_{min}} \right)^{\frac{\alpha(q+1)}{d+\alpha(q+2)}},
$$

*where $C$ is a positive constant independent of $n$ or $\delta$.*

## G.2 Error Decomposition

The well-known Comparison Theorem in classification (Zhang, 2004) asserts that for the Hinge loss $\phi$ and any measurable function $f : \mathcal{X} \to \mathbb{R}$, the following inequality holds:

$$
\underbrace{R(f) - R(f_c)}_{\text{excess risk}} \le \underbrace{\varepsilon(f) - \varepsilon(f_c)}_{\text{excess generalization error}}. \tag{31}
$$

This result implies that, to establish an upper bound on the excess risk of a classifier $f$, it suffices to bound its excess generalization error, $\varepsilon(f) - \varepsilon(f_c)$.

The proof of Theorem 4.5 begins with a standard error decomposition of $\varepsilon(f_{\text{ERM}}) - \varepsilon(f_c)$.

**Lemma G.2** (Error Decomposition of $\varepsilon(f_{\text{ERM}}) - \varepsilon(f_c)$). *Let $f_\mathcal{H}$ be any function in our hypothesis space $\mathcal{H}_\tau$. There holds*

$$
\varepsilon(f_{\text{ERM}}) - \varepsilon(f_c) \le \{\varepsilon(f_{\text{ERM}}) - \varepsilon_{T,T^-,T'}(f_{\text{ERM}})\} + \{\varepsilon_{T,T^-,T'}(f_\mathcal{H}) - \varepsilon(f_\mathcal{H})\} + \{\varepsilon(f_\mathcal{H}) - \varepsilon(f_c)\}. \tag{32}
$$

The above lemma decomposes the excess generalization error $\varepsilon(f_{\text{ERM}}) - \varepsilon(f_c)$ into three components. The first two components — $\{\varepsilon(f_{\text{ERM}}) - \varepsilon_{T,T^-,T'}(f_{\text{ERM}})\}$ and $\{\varepsilon_{T,T^-,T'}(f_\mathcal{H}) - \varepsilon(f_\mathcal{H})\}$ — are commonly considered the estimation errors, whereas the last component $\{\varepsilon(f_\mathcal{H}) - \varepsilon(f_c)\}$ approximation error does not depend on the training data.

Moving forward, we will derive upper bounds for these three error terms, starting with the approximation error.

*Proof of Lemma G.2.* We express $\varepsilon(f_{\text{ERM}}) - \varepsilon(f_c)$ by inserting empirical risks as follows

$$
\begin{aligned}
&\varepsilon(f_{\text{ERM}}) - \varepsilon(f_c) \\
=\ &\{\varepsilon(f_{\text{ERM}}) - \varepsilon_{T,T^-,T'}(f_{\text{ERM}})\} + \{\varepsilon_{T,T^-,T'}(f_{\text{ERM}}) - \varepsilon_{T,T^-,T'}(f_\mathcal{H})\} \\
&+ \{\varepsilon_{T,T^-,T'}(f_\mathcal{H}) - \varepsilon(f_\mathcal{H})\} + \{\varepsilon(f_\mathcal{H}) - \varepsilon(f_c)\}.
\end{aligned}
$$

We see that both $f_{\text{ERM}}$ and $f_\mathcal{H}$ lie on $\mathcal{H}_\tau$. By the definition of $f_{\text{ERM}}$, we know that $f_{\text{ERM}}$ minimizes the empirical risk over $\mathcal{H}_\tau$. Thus, we have $\varepsilon_{T,T^-,T'}(f_{\text{ERM}}) - \varepsilon_{T,T^-,T'}(f_\mathcal{H}) \le 0$ for all $f_\mathcal{H} \in \mathcal{H}_\tau$. This yields the expression (32). $\square$

### G.3 BOUNDING THE APPROXIMATION ERROR

In this subsection, we focus on deriving an upper bound for the approximation error. To achieve this, we leverage a result from (Schmidt-Hieber, 2020), which demonstrates that ReLU networks are capable of universally approximating any Hölder continuous function. For clarity, we first provide a formal definition of Hölder continuity.

We denote by $C^m(\mathcal{X})$ with $m \in \mathbb{N}$, the space of $m$-times differentiable functions on $\mathcal{X}$. Throughout this work, we consider the domain $\mathcal{X} = [0,1]^d$; however, this can be extended to any compact subset of $\mathbb{R}^d$. For any positive value $\alpha > 0$, let $[\alpha]^- = \lfloor \alpha - 1 \rfloor \in \mathbb{N} \cup \{0\}$. Let $\boldsymbol{\beta} = (\beta_1, \ldots, \beta_d) \in \mathbb{N}_0^d$ be an index vector, where $\mathbb{N}_0 = \mathbb{N} \cup \{0\}$. We define $|\boldsymbol{\beta}| = \beta_1 + \ldots + \beta_d$ and $x^{\boldsymbol{\beta}} = x_1^{\beta_1} \cdots x_d^{\beta_d}$ for an index vector $\boldsymbol{\beta}$. For a function $f : \mathcal{X} \to \mathbb{R}$ and a index vector $\boldsymbol{\beta} \in \mathbb{N}_0^d$, let the partial derivative of $f$ with $\boldsymbol{\beta}$ be

$$\partial^{\boldsymbol{\beta}} f = \frac{\partial^{|\boldsymbol{\beta}|} f}{\partial x^{\boldsymbol{\beta}}} = \frac{\partial^{|\boldsymbol{\beta}|} f}{\partial x_1^{\beta_1} \cdots \partial x_d^{\beta_d}}.$$

The result from (Schmidt-Hieber, 2020) specifically consider function

$$f \in \mathcal{H}^{\alpha,r}\left([0,1]^d\right) := \{f \in C^{[\alpha]^-}\left([0,1]^d\right) : \|f\|_{\mathcal{H}^{\alpha}([0,1]^d)} \le r\}.$$

Here, $\mathcal{H}^{\alpha,r}\left([0,1]^d\right)$ is a closed ball of radius $r > 0$ in the Hölder space of order $\alpha > 0$ w.r.t. the Hölder norm $\|\cdot\|_{\mathcal{H}^{\alpha}([0,1]^d)}$ given by

$$\|f\|_{\mathcal{H}^{\alpha}([0,1]^d)} = \sum_{|\boldsymbol{\beta}|_1 \le [\alpha]^-} \left\{ \|\partial^{\boldsymbol{\beta}} f\|_{C([0,1]^d)} + \sup_{x \ne y \in [0,1]^d} \frac{|\partial^{\boldsymbol{\beta}} f(x) - \partial^{\boldsymbol{\beta}} f(y)|}{|x-y|^{\alpha-[\alpha]^-}} \right\}.$$

**Lemma G.3** (Theorem 5 in (Schmidt-Hieber, 2020)). *Let $\alpha, r > 0$. For any Hölder continuous function $f \in \mathcal{H}^{\alpha,r}([0,1]^d)$ and for any integers $m \ge 1$ and $N \ge \max\left\{(\alpha+1)^d, (r+1)e^d\right\}$, there exists a ReLU neural network*

$$\widehat{f} \in \mathcal{F}(L^*, w^*, v^*, K^*)$$

*with depth*

$$L^* = 8 + (m+5)(1 + \lceil \log_2(\max\{d, \alpha\}) \rceil),$$

*maximum number of nodes*

$$w^* = 6(d + \lceil \alpha \rceil)N,$$

*number of nonzero parameters*

$$v^* = 141(d + \alpha + 1)^{3+d} N(m+6),$$

*and all parameters (absolute value) are bounded by $K^* = 1$ such that*

$$\left\|\widehat{f} - f\right\|_{L^{\infty}([0,1]^d)} \le (2r+1)(1 + d^2 + \alpha^2) 6^d N 2^{-m} + r 3^{\alpha} N^{-\frac{\alpha}{d}}. \tag{33}$$

We define our hypothesis space $\mathcal{H}_\tau$ based on the construction in Lemma G.3, with the goal of ensuring that its functions are well-suited to approximate the regression function $f_P$, assumed to be $\alpha$-Hölder continuous. Recall the "approx-sign" function $\sigma_\tau : \mathbb{R} \to [0,1]$, defined previously in equation (7) with a bandwidth parameter $\tau > 0$. $\mathcal{H}_\tau$ is defined as

$$\mathcal{H}_\tau := \mathrm{span}\left\{\sigma_\tau \circ f : f \in \mathcal{F}(L^*, w^*, v^*, K^*)\right\}$$

with parameters $L^*, w^*, v^*, K^*$ given in Lemma G.3, and $\tau$ to be determined later.

To estimate the term $\varepsilon(f_{\mathcal{H}}) - \varepsilon(f_c)$, we apply the upper bound provided in Theorem 3.3 with $k = 1$. For any measurable function $f : \mathcal{X} \to \mathbb{R}$, there holds

$$\varepsilon\left(\sigma_\tau\left(f\right)\right) - \varepsilon(f_c) \le 4c_0 \left(\tau + \|f - f_P\|_{L^{\infty}[0,1]^d}\right)^{q+1}.$$

Assuming $f_P \in \mathcal{H}^{\alpha,r}([0,1]^d)$ (i.e., $f_P$ is a $\alpha$-Hölder continuous), we choose $f \in \mathcal{F}(L^*, w^*, v^*, K^*)$ such that

$$\|f - f_P\|_{L^{\infty}([0,1]^d)} \le (2r+1)(1 + d^2 + \alpha^2) 6^d N 2^{-m} + r 3^{\alpha} N^{-\frac{\alpha}{d}}.$$

We see that $\sigma_\tau(f) = f_{\mathcal{H}} \in \mathcal{H}_\tau$ and we have

$$\varepsilon(f_{\mathcal{H}}) - \varepsilon(f_c) \le 4c_0 \left(\tau + (2r+1)(1+d^2+\alpha^2)6^d N 2^{-m} + r3^\alpha N^{-\frac{\alpha}{d}}\right)^{q+1}.$$

We restrict $\tau \le \min\{N2^{-m} + N^{-\frac{\alpha}{d}}, 1\}$. We choose $c_{r,\alpha,d} = 4c_0(1 + (2r+1)(1+d^2+\alpha^2)(6^d + 3^\alpha))^{q+1}$. Then, we have

$$\varepsilon(f_{\mathcal{H}}) - \varepsilon(f_c) \le c_{r,\alpha,d} \left(N2^{-m} + N^{-\frac{\alpha}{d}}\right)^{q+1}. \tag{34}$$

### G.4 BOUNDING THE ESTIMATION ERROR

In this subsection, we focus on deriving an upper bound of the estimation error term $\{\varepsilon(f_{\text{ERM}}) - \varepsilon_{T,T^-,T'}(f_{\text{ERM}})\} + \{\varepsilon_{T,T^-,T'}(f_{\mathcal{H}}) - \varepsilon(f_{\mathcal{H}})\}$.

We first rewrite this error term by inserting $\varepsilon(f_c)$ and $\varepsilon_{T,T^-,T'}(f_c)$ as

$$\varepsilon(f_{\text{ERM}}) - \varepsilon_{T,T^-,T'}(f_{\text{ERM}}) + \varepsilon_{T,T^-,T'}(f_{\mathcal{H}}) - \varepsilon(f_{\mathcal{H}})$$
$$= \varepsilon(f_{\text{ERM}}) - \varepsilon(f_c) - \{\varepsilon_{T,T^-,T'}(f_{\text{ERM}}) - \varepsilon_{T,T^-,T'}(f_c)\} \tag{35}$$
$$+ \varepsilon_{T,T^-,T'}(f_{\mathcal{H}}) - \varepsilon_{T,T^-,T'}(f_c) - \{\varepsilon(f_{\mathcal{H}}) - \varepsilon(f_c)\}. \tag{36}$$

Next, we will bound the two terms (35) and (36) respectively. We will first handle the error term (36).

**Step 1: Bound the term** $\varepsilon_{T,T^-,T'}(f_{\mathcal{H}}) - \varepsilon_{T,T^-,T'}(f_c) - \{\varepsilon(f_{\mathcal{H}}) - \varepsilon(f_c)\}$.

Here, we define three random variables given by

$$A(X) := \phi(f_{\mathcal{H}}(X)) - \phi(f_c(X)) \qquad \text{over } (\mathcal{X}, Q), \tag{37}$$

and

$$B(X) := \phi(-f_{\mathcal{H}}(X)) - \phi(-f_c(X)) \qquad \text{over } (\mathcal{X}, W), \tag{38}$$

and

$$C(X) := \phi(-f_{\mathcal{H}}(X)) - \phi(-f_c(X)) \qquad \text{over } (\mathcal{X}, \mu). \tag{39}$$

Then, we can write $\varepsilon_{T,T^-,T'}(f_{\mathcal{H}}) - \varepsilon_{T,T^-,T'}(f_c) - \{\varepsilon(f_{\mathcal{H}}) - \varepsilon(f_c)\}$ as a function of these three random variables $A$, $B$, and $C$ as follows:

$$\varepsilon_{T,T^-,T'}(f_{\mathcal{H}}) - \varepsilon_{T,T^-,T'}(f_c) - \{\varepsilon(f_{\mathcal{H}}) - \varepsilon(f_c)\}$$

$$= \frac{s}{n}\sum_{i=1}^{n}\phi(1 \cdot f_{\mathcal{H}}(X_i)) + \frac{(1-s)\tilde{s}}{n^-}\sum_{i=1}^{n^-}\phi(-1 \cdot f_{\mathcal{H}}(X_i^-)) + \frac{(1-s)(1-\tilde{s})}{n'}\sum_{i=1}^{n'}\phi(-1 \cdot f_{\mathcal{H}}(X_i'))$$

$$- \left(\frac{s}{n}\sum_{i=1}^{n}\phi(1 \cdot f_c(X_i)) + \frac{(1-s)\tilde{s}}{n^-}\sum_{i=1}^{n^-}\phi(-1 \cdot f_c(X_i^-)) + \frac{(1-s)(1-\tilde{s})}{n'}\sum_{i=1}^{n'}\phi(-1 \cdot f_c(X_i'))\right)$$

$$- \left(s\int_{\mathcal{X}}\phi(f_{\mathcal{H}}(X))dQ + (1-s)\tilde{s}\int_{\mathcal{X}}\phi(-f_{\mathcal{H}}(X))dW + (1-s)(1-\tilde{s})\int_{\mathcal{X}}\phi(-f_{\mathcal{H}}(X))d\mu\right)$$

$$+ \left(s\int_{\mathcal{X}}\phi(f_c(X))dQ + (1-s)\tilde{s}\int_{\mathcal{X}}\phi(-f_c(X))dW + (1-s)(1-\tilde{s})\int_{\mathcal{X}}\phi(-f_c(X))d\mu\right)$$

$$= s\left(\frac{1}{n}\sum_{i=1}^{n}A(X_i) - \mathrm{E}_Q[A(X)]\right) + (1-s)\tilde{s}\left(\frac{1}{n'}\sum_{i=1}^{n'}B(X_i^-) - \mathrm{E}_W[B(X)]\right)$$

$$+ (1-s)(1-\tilde{s})\left(\frac{1}{n'}\sum_{i=1}^{n'}C(X_i') - \mathrm{E}_\mu[C(X)]\right).$$

Moving on, we will apply the classic one-sided Bernstein's inequality to estimate (i): $\frac{1}{n}\sum_{i=1}^{n}A(X_i) - \mathrm{E}_Q[A(X)]$, (ii): $\frac{1}{n'}\sum_{i=1}^{n'}B(X_i^-) - \mathrm{E}_W[B(X)]$, and (iii): $\frac{1}{n'}\sum_{i=1}^{n'}C(X_i') - \mathrm{E}_\mu[C(X)]$, respectively, and obtain a high probability upper bound of $\varepsilon_{T,T^-,T'}(f_{\mathcal{H}}) - \varepsilon_{T,T^-,T'}(f_c) - \{\varepsilon(f_{\mathcal{H}}) - \varepsilon(f_c)\}$. The result is given in the Lemma below:

**Lemma G.4.** *Suppose the Tsybakov's noise condition holds for some $q \in [0, \infty]$ and constant $c_0 > 0$. For any $0 < \delta < 1$, with probability $1 - \frac{\delta}{2}$, we have*

$$\varepsilon_{T,T^-,T'}(f_{\mathcal{H}}) - \varepsilon_{T,T^-,T'}(f_c) - \{\varepsilon(f_{\mathcal{H}}) - \varepsilon(f_c)\}$$

$$\leq \quad C_q \left( \log \left( \frac{4}{\delta} \right) \right) \left( s^{\frac{1}{q+2}} \left( \frac{1}{n} \right)^{\frac{q+1}{q+2}} + ((1-s) \cdot \tilde{s})^{\frac{1}{q+2}} \left( \frac{1}{n^-} \right)^{\frac{q+1}{q+2}} + ((1-s) \cdot (1-\tilde{s}))^{\frac{1}{q+2}} \left( \frac{1}{n'} \right)^{\frac{q+1}{q+2}} \right) \tag{40}$$

$$+ \frac{(\varepsilon(f_{\mathcal{H}}) - \varepsilon(f_c))}{2}, \tag{41}$$

*where $C_q$ is a positive constant depending only on $q$ and $c_0$.*

*Proof of Lemma G.4.* The classic one-sided Bernstein's inequality states that, for a random variable $A$ with mean $\mathrm{E}[A]$ and variance $\mathrm{Var}[A] = \sigma^2$ satisfying $|A - \mathrm{E}[A]| \leq M$ for some $M > 0$ almost surely, the following holds for a random i.i.d. sample $\{X_i\}_{i=1}^n$ and any $t > 0$,

$$\mathrm{P}\left( \mathrm{E}[A] - \frac{1}{n} \sum_{i=1}^n A(X_i) > t \right) \leq \exp\left\{ -\frac{nt^2}{2(\sigma^2 + \frac{1}{3}Mt)} \right\}. \tag{42}$$

To apply Bernstein's inequality to random variables $A$ (37), $B$ (38), and $C$ (39), we need to first derive upper bounds for their variances. A previous result in Lemma C.3 in (Zhou et al., 2024) states that for any function $f : \mathcal{X} \to [-1, 1]$ and some $y \in \{-1, 1\}$, if the Tsybakov's noise condition holds, we have

$$\mathrm{E}_{P_{\mathcal{X}}^y}\left[ \{\phi(yf(X)) - \phi(yf_c(X))\}^2 \right] \leq \frac{5}{s_y} (c_0)^{\frac{1}{q+1}} (\varepsilon(f) - \varepsilon(f_c))^{\frac{q}{q+1}}, \tag{43}$$

where

$$s_y = s \quad \text{for } y = 1 \text{ and } s_y = 1 - s \quad \text{for } y = -1, \tag{44}$$

and

$$dP_{\mathcal{X}}^y = h_1 d\mu \quad \text{for } y = 1 \text{ and } dP_{\mathcal{X}}^y = h_2 d\mu \quad \text{for } y = -1.$$

We can see that (43) presented an upper bound for the second moment of $\phi(yf(X)) - \phi(yf_c(X))$. Then, we know the variance of $\phi(yf(X)) - \phi(yf_c(X))$ is

$$\mathrm{Var}[\phi(yf(X)) - \phi(yf_c(X))] \leq \frac{5}{s_y} (c_0)^{\frac{1}{q+1}} (\varepsilon(f) - \varepsilon(f_c))^{\frac{q}{q+1}}.$$

Then the variance $\sigma^2$ of the random variable $A$ (for $y = 1$) and $B, C$ (for $y = -1$) is bounded by

$$\sigma^2 \leq \frac{5}{s_y} (c_0)^{\frac{1}{q+1}} (\varepsilon(f) - \varepsilon(f_c))^{\frac{q}{q+1}}.$$

Now we are in a position to apply the one-sided Bernstein's inequality (42). For any $t > 0$, there holds, with probability at least $1 - \exp\left( -\frac{nt^2}{2(\sigma^2 + 2t/3)} \right)$,

$$\left( \frac{1}{n} \sum_{i=1}^n A(X_i) - \mathrm{E}_Q[A(X)] \right) \leq t.$$

For $0 < \delta < 1$, we set

$$1 - \exp\left( -\frac{nt^2}{2(\sigma^2 + 2t/3)} \right) = 1 - \frac{\delta}{6},$$

and we solve $t$ for the quadratic equation:

$$\log\left( \frac{6}{\delta} \right) = \frac{nt^2}{2(\sigma^2 + 2t/3)}.$$

The positive solution to this quadratic equation of $t$ is given by

$$
\begin{aligned}
t^* &= \frac{\frac{4}{3} \log\left(\frac{6}{\delta}\right) + \sqrt{\frac{16}{9} \log^2\left(\frac{6}{\delta}\right) + 8n\sigma^2 \log\left(\frac{6}{\delta}\right)}}{2n} \\
&\leq \frac{2}{3n} \log\left(\frac{6}{\delta}\right) + \frac{2}{3n} \log\left(\frac{6}{\delta}\right) + \frac{\sqrt{2\sigma^2 \log\left(\frac{6}{\delta}\right)}}{\sqrt{n}} \\
&\leq \frac{4}{3n} \log\left(\frac{6}{\delta}\right) + 4 \frac{\sqrt{\log\left(\frac{6}{\delta}\right)}}{\sqrt{ns}} (c_0)^{\frac{1}{2(q+1)}} \left(\varepsilon(f_{\mathcal{H}}) - \varepsilon(f_c)\right)^{\frac{q}{2(q+1)}}.
\end{aligned}
$$

We further apply the Young's inequality for product (Young, 1912) to the above estimate of $t^*$, and we get

$$
t^* \leq \frac{4}{3n} \log\left(\frac{6}{\delta}\right) + \left(\frac{q+2}{2(q+1)}\right) \left(4 \frac{\sqrt{\log\left(\frac{6}{\delta}\right)}}{\sqrt{ns}} (c_0)^{\frac{1}{2(q+1)}}\right)^{\frac{2(q+1)}{q+2}} + \frac{\varepsilon(f_{\mathcal{H}}) - \varepsilon(f_c)}{\frac{2(q+1)}{q}}.
$$

In other words, we know for probability at least $1 - \frac{\delta}{6}$, there holds

$$
\left(\frac{1}{n} \sum_{i=1}^{n} A(X_i) - \mathrm{E}_Q[A(X)]\right)
$$
$$
\leq \frac{4}{3n} \log\left(\frac{6}{\delta}\right) + \left(\frac{q+2}{2(q+1)}\right) \left(4 \frac{\sqrt{\log\left(\frac{6}{\delta}\right)}}{\sqrt{ns}} (c_0)^{\frac{1}{2(q+1)}}\right)^{\frac{2(q+1)}{q+2}} + \frac{\varepsilon(f_{\mathcal{H}}) - \varepsilon(f_c)}{\frac{2(q+1)}{q}}.
$$

We adopt the same approach and we obtain that, for probability at least at least $1 - \frac{\delta}{6}$, there holds

$$
\left(\frac{1}{n^-} \sum_{i=1}^{n^-} B(X_i^-) - \mathrm{E}_W[B(X^-)]\right)
$$
$$
\leq \frac{4}{3n^-} \log\left(\frac{6}{\delta}\right) + \left(\frac{q+2}{2(q+1)}\right) \left(4 \frac{\sqrt{\log\left(\frac{6}{\delta}\right)}}{\sqrt{n^-(1-s))}} (c_0)^{\frac{1}{2(q+1)}}\right)^{\frac{2(q+1)}{q+2}} + \frac{\varepsilon(f_{\mathcal{H}}) - \varepsilon(f_c)}{\frac{2(q+1)}{q}}
$$

and

$$
\left(\frac{1}{n'} \sum_{i=1}^{n'} C(X_i') - \mathrm{E}_\mu[C(X')]\right)
$$
$$
\leq \frac{4}{3n'} \log\left(\frac{6}{\delta}\right) + \left(\frac{q+2}{2(q+1)}\right) \left(4 \frac{\sqrt{\log\left(\frac{6}{\delta}\right)}}{\sqrt{n'(1-s))}} (c_0)^{\frac{1}{2(q+1)}}\right)^{\frac{2(q+1)}{q+2}} + \frac{\varepsilon(f_{\mathcal{H}}) - \varepsilon(f_c)}{\frac{2(q+1)}{q}}.
$$

Combining the above estimates, we know, for probability at least $1 - \frac{\delta}{2}$,

$$\varepsilon_{T,T^-,T'}(f_{\mathcal{H}}) - \varepsilon_{T,T^-,T'}(f_c) - \{\varepsilon(f_{\mathcal{H}}) - \varepsilon(f_c)\}$$

$$= s\left(\frac{1}{n}\sum_{i=1}^{n} A(X_i) - \mathrm{E}_Q[A(X)]\right) + (1-s)\tilde{s}\left(\frac{1}{n'}\sum_{i=1}^{n'} B(X_i^-) - \mathrm{E}_W[B(X)]\right)$$

$$+ (1-s)(1-\tilde{s})\left(\frac{1}{n'}\sum_{i=1}^{n'} C(X_i') - \mathrm{E}_\mu[C(X)]\right)$$

$$\leq \frac{4}{3}\left(\frac{s}{n} + \frac{(1-s)\tilde{s}}{n^-} + \frac{(1-s)(1-\tilde{s})}{n'}\right)\log\left(\frac{6}{\delta}\right)$$

$$+ 4^{\frac{2(q+1)}{q+2}}(c_0)^{\frac{1}{q+2}}\left(\log\left(\frac{6}{\delta}\right)\right)^{\frac{q+1}{q+2}}$$

$$\cdot\left(s^{\frac{1}{q+2}}\left(\frac{1}{n}\right)^{\frac{q+1}{q+2}} + (1-s)^{\frac{1}{q+2}}\tilde{s}\left(\frac{1}{n^-}\right)^{\frac{q+1}{q+2}} + (1-s)^{\frac{1}{q+2}}(1-\tilde{s})\left(\frac{1}{n'}\right)^{\frac{q+1}{q+2}}\right)$$

$$+ \left(\frac{q}{2(q+1)}\right)(\varepsilon(f_{\mathcal{H}}) - \varepsilon(f_c))$$

$$\leq C_q\left(\log\left(\frac{6}{\delta}\right)\right)$$

$$\cdot\left(s^{\frac{1}{q+2}}\left(\frac{1}{n}\right)^{\frac{q+1}{q+2}} + ((1-s)\cdot\tilde{s})^{\frac{1}{q+2}}\left(\frac{1}{n^-}\right)^{\frac{q+1}{q+2}} + ((1-s)\cdot(1-\tilde{s}))^{\frac{1}{q+2}}\left(\frac{1}{n'}\right)^{\frac{q+1}{q+2}}\right)$$

$$+ \frac{(\varepsilon(f_{\mathcal{H}}) - \varepsilon(f_c))}{2},$$

where $C_q$ is a positive constant depending only on $q$ and $c_0$. Note that in the last inequality, we have used the facts that $s \leq s^{\frac{1}{q+2}}, 1 - s \leq (1-s)^{\frac{1}{q+2}}, \tilde{s} \leq \tilde{s}^{\frac{1}{q+2}}, 1 - \tilde{s} \leq (1-\tilde{s})^{\frac{1}{q+2}}$ for all $s, \tilde{s} \in (0,1)$ and $q > 0$. We have also used the fact that $\frac{1}{n} \leq \left(\frac{1}{n}\right)^{\frac{q+1}{q+2}}, \frac{1}{n^-} \leq \left(\frac{1}{n^-}\right)^{\frac{q+1}{q+2}}, \frac{1}{n'} \leq \left(\frac{1}{n'}\right)^{\frac{q+1}{q+2}}$ for all $n, n^-, n' \in \mathbb{N}$ and $q > 0$. $\qquad\square$

**Step 2: Bound the term** $\varepsilon(f_{\mathrm{ERM}}) - \varepsilon(f_c) - \{\varepsilon_{T,T^-,T'}(f_{\mathrm{ERM}}) - \varepsilon_{T,T^-,T'}(f_c)\}$**.**

Here, we will proceed to estimate the error term $\varepsilon(f_{\mathrm{ERM}}) - \varepsilon(f_c) - \{\varepsilon_{T,T^-,T'}(f_{\mathrm{ERM}}) - \varepsilon_{T,T^-,T'}(f_c)\}$, which is given in (35). We will derive an upper bound for this error term using a concentration inequality in terms of covering numbers.

For $\epsilon > 0$, denote by $\mathcal{N}(\epsilon, \mathcal{H}) := \mathcal{N}(\epsilon, \mathcal{H}, \|\cdot\|_\infty)$ the $\epsilon$-covering number of a set of functions $\mathcal{H}$ with respect to $\|\cdot\|_\infty := \mathrm{ess\,sup}_{x \in \mathcal{X}}|f(x)|$. More specifically, $\mathcal{N}(\epsilon, \mathcal{H})$ is the minimal $M \in \mathbb{N}$ such that there exists functions $\{f_1, \ldots, f_M\} \in \mathcal{H}$ satisfying

$$\min_{1 \leq i \leq M}\|f - f_i\|_\infty \leq \epsilon, \qquad \forall f \in \mathcal{H}. \tag{45}$$

In particular, we seek to estimate the covering number of our hypothesis space $\mathcal{H}_\tau$ defined in Definition 4.3. $\mathcal{H}_\tau$ is the class of ReLU neural networks we use to learn the Bayes classifier.

**Lemma G.5** (Corollary C.6 from (Zhou et al., 2024)). *Consider the hypothesis space $\mathcal{H}_\tau$ defined in Definition 4.3 with hyperparameter $0 < \tau \leq 1$, and integers $m \geq 1$ and $N \geq \max\left\{(\alpha+1)^d, (r+1)e^d\right\}$. For any $0 < \epsilon \leq 1$, the $\epsilon$-covering number of $\mathcal{H}_\tau$ satisfies*

$$\log \mathcal{N}(\epsilon, \mathcal{H}_\tau) \leq c_{\alpha,d}m^2 N \log((\tau\epsilon)^{-1}mN), \tag{46}$$

*where $c_{\alpha,d}$ is a positive constant independent of $r, m, N, \tau$ or $\epsilon$.*

Next, by utilizing this estimate of $\epsilon$-covering number of $\mathcal{H}_\tau$, we derive the following upper bound for the error term $\varepsilon(f_{\mathrm{ERM}}) - \varepsilon(f_c) - \{\varepsilon_{T,T^-,T'}(f_{\mathrm{ERM}}) - \varepsilon_{T,T^-,T'}(f_c)\}$.

**Lemma G.6.** *Let $\alpha, r > 0$, and integers $m \geq 1$ and $N \geq \max\left\{(\alpha+1)^d, (r+1)e^d\right\}$. Suppose the Tsybakov's noise condition (3) holds for some $0 \leq q \leq \infty$ and constant $c_0 > 0$. Also let $\tau \geq \max\left\{\frac{1}{n}, \frac{1}{n^-}, \frac{1}{n'}\right\}$. For any $0 < \delta < 1$ and $n, n^-, n' \geq 3$, with probability $1 - \frac{\delta}{2}$, we have*

$$\varepsilon(f_{ERM}) - \varepsilon(f_c) - \left(\varepsilon_{T,T^-,T'}(f_{ERM}) - \varepsilon_{T,T^-,T'}(f_c)\right)$$

$$\leq \quad C_{q,\alpha,d}\left(\log\left(\frac{6}{\delta}\right) + 3m^2 N \log\left(\max\{n, n^-, n'\}mN\right)\right)^{\frac{q+2}{q+1}}$$

$$\cdot \left(\max\left\{\frac{s}{n}\log\left(\frac{n}{s}\right), \frac{(1-s)\tilde{s}}{n^-}\log\left(\frac{n^-}{(1-s)\tilde{s}}\right), \frac{(1-s)(1-\tilde{s})}{n'}\log\left(\frac{n'}{(1-s)(1-\tilde{s})}\right)\right\}\right)^{\frac{q+2}{q+1}}$$

$$+ \frac{\varepsilon(f_{ERM}) - \varepsilon(f_c)}{2},$$

*where $C_{q,\alpha,d}$ is a positive constant depending only on $q, c_0, \alpha$, and $d$.*

*Proof of Lemma G.6.* Consider a fixed function $f : \mathcal{X} \to [-1, 1]$. For $s, \tilde{s} \in (0, 1)$, we apply the one-sided Berstein's inequality (42) to following three random variables:

$$a(X) = s(\phi(f_{\mathcal{H}}(X)) - \phi(f_c(X))) \qquad \text{over } (\mathcal{X}, Q),$$

and

$$b(X) = (1-s)\tilde{s}(\phi(-f_{\mathcal{H}}(X)) - \phi(-f_c(X))) \qquad \text{over } (\mathcal{X}, W),$$

and

$$c(X) = (1-s)(1-\tilde{s})(\phi(-f_{\mathcal{H}}(X)) - \phi(-f_c(X))) \qquad \text{over } (\mathcal{X}, \mu).$$

We can see that, almost surely, $|a| \leq 2s$ and thereby $|a - \mathrm{E}[a]| \leq 4s$, $|b| \leq 2(1-s)\tilde{s}$ and thereby $|b - \mathrm{E}[b]| \leq 4(1-s)\tilde{s}$, and $|c| \leq 2(1-s)(1-\tilde{s})$ and thereby $|c - \mathrm{E}[c]| \leq 4(1-s)(1-\tilde{s})$. Moreover, by applying a similar argument in the proof of Lemma G.4, we know when the noise condition for some $0 \leq q \leq \infty$ holds, the variances of these random variables can be bounded by

$$\mathrm{Var}[a] \leq \frac{5s^2}{s}(c_0)^{\frac{1}{q+1}}(\varepsilon(f) - \varepsilon(f_c))^{\frac{q}{q+1}} = 5s(c_0)^{\frac{1}{q+1}}(\varepsilon(f) - \varepsilon(f_c))^{\frac{q}{q+1}}$$

and

$$\mathrm{Var}[b] \leq 5(1-s)\tilde{s}^2(c_0)^{\frac{1}{q+1}}(\varepsilon(f) - \varepsilon(f_c))^{\frac{q}{q+1}}$$

and

$$\mathrm{Var}[c] \leq 5(1-s)(1-\tilde{s})^2(c_0)^{\frac{1}{q+1}}(\varepsilon(f) - \varepsilon(f_c))^{\frac{q}{q+1}}.$$

We then apply the one-sided Bernstein's inequality to each of these random variables. We combine the estimates together and get

$$\frac{\varepsilon(f) - \varepsilon(f_c) - (\varepsilon_{T,T'}(f) - \varepsilon_{T,T'}(f_c))}{\left((\varepsilon(f) - \varepsilon(f_c))^{\frac{q}{q+1}} + \epsilon^{\frac{q}{q+1}}\right)^{1/2}} \leq 2\epsilon^{1 - \frac{q}{2(q+1)}}, \qquad \forall \epsilon > 0$$

with probability at least

$$1 - \exp\left\{-\frac{n\epsilon^{2 - \frac{q}{q+1}}}{s\left(10(c_0)^{\frac{1}{q+1}} + 2\epsilon^{1 - \frac{q}{q+1}}\right)}\right\} - \exp\left\{-\frac{n^-\epsilon^{2 - \frac{q}{q+1}}}{(1-s)\tilde{s}\left(10(c_0)^{\frac{1}{q+1}} + 2\epsilon^{1 - \frac{q}{q+1}}\right)}\right\}$$

$$- \exp\left\{-\frac{n'\epsilon^{2 - \frac{q}{q+1}}}{(1-s)(1-\tilde{s})\left(10(c_0)^{\frac{1}{q+1}} + 2\epsilon^{1 - \frac{q}{q+1}}\right)}\right\}.$$

Specifically, we consider our hypothesis space $\mathcal{H}_\tau$ and its covering number estimate $\mathcal{N}(\epsilon, \mathcal{H}_\tau)$, as given in (46) for any $\epsilon > 0$. Building on a similar result from Lemma C.8 in (Zhou et al., 2024), if the noise condition holds for some $0 \leq q \leq \infty$, we have for any $\epsilon > 0$,

$$\varepsilon(f) - \varepsilon(f_c) - (\varepsilon_{T,T'}(f) - \varepsilon_{T,T'}(f_c)) \leq 5\epsilon^{1 - \frac{q}{2(q+1)}}\left((\varepsilon(f) - \varepsilon(f_c))^{\frac{q}{q+1}} + \epsilon^{\frac{q}{q+1}}\right)^{\frac{1}{2}}, \qquad \forall f \in \mathcal{H}_\tau$$

with probability at least

$$
1 - \mathcal{N}\left(\epsilon, \mathcal{H}_\tau\right) \left\{ \exp\left\{ -\frac{n\epsilon^{2-\frac{q}{q+1}}}{s\left(10(c_0)^{\frac{1}{q+1}} + 2\epsilon^{1-\frac{q}{q+1}}\right)} \right\} \right.
$$

$$
+ \exp\left\{ -\frac{n^-\epsilon^{2-\frac{q}{q+1}}}{(1-s)\tilde{s}\left(10(c_0)^{\frac{1}{q+1}} + 2\epsilon^{1-\frac{q}{q+1}}\right)} \right\}
$$

$$
\left. + \exp\left\{ -\frac{n'\epsilon^{2-\frac{q}{q+1}}}{(1-s)(1-\tilde{s})\left(10(c_0)^{\frac{1}{q+1}} + 2\epsilon^{1-\frac{q}{q+1}}\right)} \right\} \right\}
$$

$$
\geq 1 - \exp\{c_{\alpha,d}m^2 N \log((\tau\epsilon)^{-1}mN)\}
$$

$$
\left\{ \exp\left\{ -\frac{n\epsilon^{2-\frac{q}{q+1}}}{s\left(10(c_0)^{\frac{1}{q+1}} + 2\right)} \right\} + \exp\left\{ -\frac{n^-\epsilon^{2-\frac{q}{q+1}}}{(1-s)\tilde{s}\left(10(c_0)^{\frac{1}{q+1}} + 2\right)} \right\} \right.
$$

$$
\left. + \exp\left\{ -\frac{n'\epsilon^{2-\frac{q}{q+1}}}{(1-s)(1-\tilde{s})\left(10(c_0)^{\frac{1}{q+1}} + 2\right)} \right\} \right\}.
$$

We choose $\tau$ satisfying $\tau \geq \max\{\frac{1}{n}, \frac{1}{n^-}, \frac{1}{n'}\}$. Then we know $\tau^{-1} \leq n$, $\tau^{-1} \leq n^-$, and $\tau^{-1} \leq n'$. We set the above confidence bound to be at least $1 - \delta/2$. Then, we find $\epsilon > 0$ that satisfies the following three inequalities:

$$
\exp\left\{ c_{\alpha,d}m^2 N \log(\epsilon^{-1}nmN) - \frac{n\epsilon^{2-\frac{q}{q+1}}}{s\left(10(c_0)^{\frac{1}{q+1}} + 2\right)} \right\} \leq \frac{\delta}{6}
$$

and

$$
\exp\left\{ c_{\alpha,d}m^2 N \log(\epsilon^{-1}n^-mN) - \frac{n^-\epsilon^{2-\frac{q}{q+1}}}{(1-s)\tilde{s}\left(10(c_0)^{\frac{1}{q+1}} + 2\right)} \right\} \leq \frac{\delta}{6}
$$

and

$$
\exp\left\{ c_{\alpha,d}m^2 N \log(\epsilon^{-1}n'mN) - \frac{n'\epsilon^{2-\frac{q}{q+1}}}{(1-s)(1-\tilde{s})\left(10(c_0)^{\frac{1}{q+1}} + 2\right)} \right\} \leq \frac{\delta}{6}.
$$

Following the proof of Lemma C.8 in (Zhou et al., 2024), we know $\epsilon$ satisfying all the above three inequalities can be given by

$$
\epsilon = \left( \max\left\{ \frac{Bs\log\left(\frac{n}{s}\right)}{n}, \frac{B^-(1-s)\tilde{s}}{n^-}\log\left(\frac{n'}{(1-s)\tilde{s}}\right), \frac{B'(1-s)(1-\tilde{s})}{n'}\log\left(\frac{n'}{(1-s)(1-\tilde{s})}\right) \right\} \right)^{\frac{q+1}{q+2}},
$$

$$
\tag{47}
$$

where $c_q = 10(c_0)^{\frac{1}{q+1}} + 2$, $B := c_q\left(\frac{2c_{\alpha,d}}{2-\frac{q}{q+1}}m^2 N + \log\left(\frac{6}{\delta}\right) + c_{\alpha,d}m^2 N\log(nmN)\right)$, $B^- := c_q\left(\frac{2c_{\alpha,d}}{2-\frac{q}{q+1}}m^2 N + \log\left(\frac{6}{\delta}\right) + c_{\alpha,d}m^2 N\log(n^-mN)\right)$, and $B' := c_q\left(\frac{2c_{\alpha,d}}{2-\frac{q}{q+1}}m^2 N + \log\left(\frac{6}{\delta}\right) + c_{\alpha,d}m^2 N\log(n'mN)\right)$. Note that

$$
B \leq c_{q,\alpha,d}\left(\log\left(\frac{6}{\delta}\right) + m^2 N(1 + \log(nmN))\right),
\tag{48}
$$

where $c_{q,\alpha,d} = c_q\left(\frac{2c_{\alpha,d}}{2-\frac{q}{q+1}} + c_{\alpha,d}\right) \geq 1$ is a positive constant depending only on $q, c_0, \alpha,$ and $d$. We also note that

$$
B^- \leq c_{q,\alpha,d}\left(\log\left(\frac{6}{\delta}\right) + m^2 N(1 + \log(n^-mN))\right)
\tag{49}
$$

and

$$B' \leq c_{q,\alpha,d}\left(\log\left(\frac{6}{\delta}\right) + m^2 N(1 + \log(n'mN))\right). \tag{50}$$

Next, we take $f = f_{\text{ERM}} \in \mathcal{H}_\tau$. We take $\epsilon$ chosen in (47). We apply Young's inequality for products (Young, 1912), and with probability at least $1 - \frac{\delta}{2}$,

$$\varepsilon(f_{\text{ERM}}) - \varepsilon(f_c) - \left(\varepsilon_{T,T^-,T'}(f_{\text{ERM}}) - \varepsilon_{T,T^-,T'}(f_c)\right)$$

$$\leq 5\epsilon^{1-\frac{q}{2(q+1)}}\left(\left(\varepsilon(f_{\text{ERM}}) - \varepsilon(f_c)\right)^{\frac{q}{q+1}} + \epsilon^{\frac{q}{q+1}}\right)^{\frac{1}{2}}$$

$$\leq 5^{\frac{2(q+1)}{q+2}}$$

$$\cdot \left(\max\left\{\frac{Bs\log\left(\frac{n}{s}\right)}{n}, \frac{B^-(1-s)\tilde{s}}{n^-}\log\left(\frac{n^-}{(1-s)\tilde{s}}\right), \frac{B'(1-s)(1-\tilde{s})}{n'}\log\left(\frac{n'}{(1-s)(1-\tilde{s})}\right)\right\}\right)^{\frac{q+1}{q+2}}$$

$$+ \frac{\varepsilon(f_{\text{ERM}}) - \varepsilon(f_c)}{2}$$

$$\leq 5^{\frac{2(q+1)}{q+2}}\left(\max\{B, B^-, B'\}\right)^{\frac{q+1}{q+2}}$$

$$\cdot \left(\max\left\{\frac{s\log\left(\frac{n}{s}\right)}{n}, \frac{(1-s)\tilde{s}}{n^-}\log\left(\frac{n^-}{(1-s)\tilde{s}}\right), \frac{(1-s)(1-\tilde{s})}{n'}\log\left(\frac{n'}{(1-s)(1-\tilde{s})}\right)\right\}\right)^{\frac{q+1}{q+2}}$$

$$+ \frac{\varepsilon(f_{\text{ERM}}) - \varepsilon(f_c)}{2}.$$

We then apply the upper bounds of $B$ (48), $B^-$ (49), and $B'$ (50) and get, with probability at least $1 - \frac{\delta}{2}$,

$$\varepsilon(f_{\text{ERM}}) - \varepsilon(f_c) - \left(\varepsilon_{T,T^-,T'}(f_{\text{ERM}}) - \varepsilon_{T,T^-,T'}(f_c)\right)$$

$$\leq C_{q,\alpha,d}\left(\log\left(\frac{6}{\delta}\right) + m^2 N(\log(nmN) + \log(n^-mN) + \log(n'mN))\right)^{\frac{q+1}{q+2}}$$

$$\cdot \left(\max\left\{\frac{s\log\left(\frac{n}{s}\right)}{n}, \frac{(1-s)\tilde{s}}{n^-}\log\left(\frac{n^-}{(1-s)\tilde{s}}\right), \frac{(1-s)(1-\tilde{s})}{n'}\log\left(\frac{n'}{(1-s)(1-\tilde{s})}\right)\right\}\right)^{\frac{q+1}{q+2}}$$

$$+ \frac{\varepsilon(f_{\text{ERM}}) - \varepsilon(f_c)}{2}$$

$$\leq C_{q,\alpha,d}\left(\log\left(\frac{6}{\delta}\right) + 3m^2 N\log(\max\{n, n^-, n'\}mN)\right)^{\frac{q+1}{q+2}}$$

$$\cdot \left(\max\left\{\frac{s\log\left(\frac{n}{s}\right)}{n}, \frac{(1-s)\tilde{s}}{n^-}\log\left(\frac{n^-}{(1-s)\tilde{s}}\right), \frac{(1-s)(1-\tilde{s})}{n'}\log\left(\frac{n'}{(1-s)(1-\tilde{s})}\right)\right\}\right)^{\frac{q+1}{q+2}}$$

$$+ \frac{\varepsilon(f_{\text{ERM}}) - \varepsilon(f_c)}{2},$$

where $C_{q,\alpha,d}$ is a positive constant depending only on $q, c_0, \alpha$, and $d$. $\qquad\square$

### G.5 COMBINING ERROR BOUNDS TOGETHER

Assume $n, n^-, n' \geq 3$. Let $\tau \geq \max\left\{\frac{1}{n}, \frac{1}{n^-}, \frac{1}{n'}\right\}$ and $\tau \leq \min\{N2^{-m} + N^{-\frac{\alpha}{d}}, 1\}$. Suppose Tsybakov's noise condition holds for some $q \in [0, \infty]$ and constant $c_0 > 0$. For any $0 < \delta < 1$, with

probability at least $1 - \delta$, we have

$$
\varepsilon(f_{\text{ERM}}) - \varepsilon(f_c)
$$
$$
\leq \quad \{\varepsilon(f_{\text{ERM}}) - \varepsilon_{T,T^-,T'}(f_{\text{ERM}})\} + \{\varepsilon_{T,T^-,T'}(f_{\mathcal{H}}) - \varepsilon(f_{\mathcal{H}})\} + \{\varepsilon(f_{\mathcal{H}}) - \varepsilon(f_c)\}
$$
$$
\leq \quad C_{q,\alpha,d} \left( \log\left(\frac{6}{\delta}\right) + 3m^2 N \log(\max\{n, n^-, n'\}mN) \right)^{\frac{q+1}{q+2}}
$$
$$
\cdot \left( \max\left\{ \frac{s \log\left(\frac{n}{s}\right)}{n}, \frac{(1-s)\tilde{s}}{n^-} \log\left(\frac{n^-}{(1-s)\tilde{s}}\right), \frac{(1-s)(1-\tilde{s})}{n'} \log\left(\frac{n'}{(1-s)(1-\tilde{s})}\right) \right\} \right)^{\frac{q+1}{q+2}}
$$
$$
+ \frac{\varepsilon(f_{\text{ERM}}) - \varepsilon(f_c)}{2}
$$
$$
+ C_q \left( \log\left(\frac{6}{\delta}\right) \right)
$$
$$
\cdot \left( s^{\frac{1}{q+2}} \left(\frac{1}{n}\right)^{\frac{q+1}{q+2}} + ((1-s)\cdot\tilde{s})^{\frac{1}{q+2}} \left(\frac{1}{n^-}\right)^{\frac{q+1}{q+2}} + ((1-s)\cdot(1-\tilde{s}))^{\frac{1}{q+2}} \left(\frac{1}{n'}\right)^{\frac{q+1}{q+2}} \right)
$$
$$
+ \frac{(\varepsilon(f_{\mathcal{H}}) - \varepsilon(f_c))}{2} + (\varepsilon(f_{\mathcal{H}}) - \varepsilon(f_c)),
$$

where we recall that $C_{q,\alpha,d}$ is a positive constant depending only on $q, c_0, \alpha$, and $d$; and $C_q$ is a positive constant depending only on $q$ and $c_0$.

We multiply both sides of the inequality by 2 and substitute the estimate of the final term, $\varepsilon(f_{\mathcal{H}}) - \varepsilon(f_c)$ (i.e., the approximation error term), from (34). This yields

$$
\varepsilon(f_{\text{ERM}}) - \varepsilon(f_c)
$$
$$
\leq \quad 2C_{q,\alpha,d} \left( \log\left(\frac{6}{\delta}\right) + 3m^2 N \log(\max\{n, n^-, n'\}mN) \right)^{\frac{q+1}{q+2}}
$$
$$
\cdot \left( \max\left\{ \frac{s \log\left(\frac{n}{s}\right)}{n}, \frac{(1-s)\tilde{s}}{n^-} \log\left(\frac{n^-}{(1-s)\tilde{s}}\right), \frac{(1-s)(1-\tilde{s})}{n'} \log\left(\frac{n'}{(1-s)(1-\tilde{s})}\right) \right\} \right)^{\frac{q+1}{q+2}}
$$
$$
+ 2C_q \left( \log\left(\frac{6}{\delta}\right) \right)
$$
$$
\cdot \left( s^{\frac{1}{q+2}} \left(\frac{1}{n}\right)^{\frac{q+1}{q+2}} + ((1-s)\cdot\tilde{s})^{\frac{1}{q+2}} \left(\frac{1}{n^-}\right)^{\frac{q+1}{q+2}} + ((1-s)\cdot(1-\tilde{s}))^{\frac{1}{q+2}} \left(\frac{1}{n'}\right)^{\frac{q+1}{q+2}} \right)
$$
$$
+ 3c_{r,\alpha,d} \left( N2^{-m} + N^{-\frac{\alpha}{d}} \right)^{q+1}.
$$

We select the smallest $m \in \mathbb{N}$ such that $N2^{-m} \leq N^{-\frac{\alpha}{d}}$. This condition translates to:

$$
m = \left\lceil \left(1 + \frac{\alpha}{d}\right) \frac{\log N}{\log 2} \right\rceil.
$$

With this choice of $m$, it follows that:

$$
N2^{-m} + N^{-\frac{\alpha}{d}} \leq 2N^{-\frac{\alpha}{d}}.
$$

We also know that

$$
\frac{s}{n} \log\left(\frac{n}{s}\right) \leq \frac{\log n}{n}
$$

and

$$
\frac{(1-s)\tilde{s}}{n^-} \log\left(\frac{n^-}{1-s)\tilde{s}}\right) \leq \frac{\log n^-}{n^-}
$$

and

$$
\frac{(1-s)(1-\tilde{s})}{n'} \log\left(\frac{n'}{(1-s)(1-\tilde{s})}\right) \leq \frac{\log n'}{n'}.
$$

Then, we have

$$\max\left\{\frac{s\log\left(\frac{n}{s}\right)}{n}, \frac{(1-s)\tilde{s}}{n^-}\log\left(\frac{n^-}{(1-s)\tilde{s}}\right), \frac{(1-s)(1-\tilde{s})}{n'}\log\left(\frac{n'}{(1-s)(1-\tilde{s})}\right)\right\}$$

$$\leq \max\left\{\frac{\log n}{n}, \frac{\log n^-}{n^-}, \frac{\log n'}{n'}\right\}$$

$$\leq \frac{\log(\min\{n, n^-, n'\})}{\min\{n, n^-, n'\}}.$$

For brevity, we let $n_{\min} := \min\{n, n^-, n'\}$. It follows that with probability at least $1 - \delta$,

$$\varepsilon(f_{\text{ERM}}) - \varepsilon(f_c)$$

$$\leq 2C_{q,\alpha,d}\left(\log\left(\frac{6}{\delta}\right) + 3m^2 N\log(\max\{n, n^-, n'\}mN) + 6C_q\log\left(\frac{6}{\delta}\right)\right)^{\frac{q+1}{q+2}}\left(\frac{\log(n_{\min})}{n_{\min}}\right)^{\frac{q+1}{q+2}}$$

$$+3c_{r,\alpha,d}\,2^{q+1}N^{-\frac{\alpha(q+1)}{d}}.$$

We choose $N \in \mathbb{N}$ to be the smallest integer satisfying

$$\left(N\frac{(\log(n_{\min}))^4}{n_{\min}}\right) \geq N^{-\frac{\alpha}{d}}, \tag{51}$$

that is,

$$N = \left\lceil\left(\frac{n_{\min}}{(\log(n_{\min}))^4}\right)^{\frac{d}{d+\alpha(q+2)}}\right\rceil. \tag{52}$$

Then, for some $C'_{q,\alpha,d} > 0$, when $n_{\min} \geq C'_{q,\alpha,d}$, we have $m = \left(1 + \frac{\alpha}{d}\right)\frac{\log N}{\log 2} \leq N$ and thereby

$$m^2 N\log(mN) \leq \left(2\left(1 + \frac{\alpha}{d}\right)\frac{\log N}{\log 2}\right)^2 N\log N^2$$

$$\leq \frac{8}{\log 2}\left(1 + \frac{\alpha}{d}\right)^2 N(\log N)^3$$

$$\leq \frac{8}{\log 2}\left(1 + \frac{\alpha}{d}\right)^2 N(\log n_{\min})^3.$$

Thus, with probability at least $1 - \delta$,

$$\varepsilon(f_{\text{ERM}}) - \varepsilon(f_c)$$

$$\leq 2C''_{q,\alpha,d}\left(\log\left(\frac{6}{\delta}\right) + (\log(\max\{n, n^-, n'\}))^{\frac{q+1}{q+2}}\right)\left(N\frac{(\log n_{\min})^3}{n_{\min}}\right)^{\frac{q+1}{q+2}}$$

$$+3c_{r,\alpha,d}\,2^{q+1}N^{-\frac{\alpha(q+1)}{d}},$$

where $C''_{q,\alpha,d}$ is a constant depending only on $q, \alpha, d$.

By the choice of $N$ at (52), we know

$$\left((N-1)\frac{(\log n_{\min})^4}{n_{\min}}\right)^{\frac{1}{q+2}} \leq (N-1)^{-\frac{\alpha}{N}}.$$

So, from $N \leq 2(N-1)$, the above estimate yields

$$\varepsilon(f_{\text{ERM}}) - \varepsilon(f_c)$$

$$\leq C''_{q,\alpha,d}\left(\log\left(\frac{6}{\delta}\right) + (\log(\max\{n, n^-, n'\}))^{\frac{q+1}{q+2}}\right)2^{\frac{q+1}{q+2}}(N-1)^{-\frac{\alpha(q+1)}{d}} + 3c_{r,\alpha,d}\,2^{q+1}N^{-\frac{\alpha(q+1)}{d}}$$

$$\leq \left(C''_{q,\alpha,d}\left(\log\left(\frac{6}{\delta}\right) + (\log(\max\{n, n^-, n'\}))^{\frac{q+1}{q+2}}\right)2^{\frac{q+1}{q+2}} \cdot 2^{\frac{\alpha(q+1)}{d}} + 3c_{r,\alpha,d}\,2^{q+1}\right)N^{-\frac{\alpha(q+1)}{d}}$$

$$\leq \widehat{C}_{q,\alpha,d,r}\left(\log\left(\frac{6}{\delta}\right) + \log(\max\{n, n^-, n'\})\right)\left(\frac{n_{\min}}{(\log n_{\min})^4}\right)^{-\frac{\alpha(q+1)}{d+\alpha(q+2)}},$$

Table 3: **AUPR** and **AUROC** results for our VC model with and without synthetic anomalies (-SA suffix) across five different datasets.

| Dataset | Type | Anom. | Random | AUPR | | AUROC | |
|---|---|---|---|---|---|---|---|
| | | | | VC | VC-SA (Ours) | VC | VC-SA (Ours) |
| NSL-KDD | Unk. | DoS | 0.431 | 0.345±0.036 | **0.793±0.055** | 0.331±0.131 | **0.807±0.057** |
| | | Probe | 0.197 | 0.180±0.010 | **0.649±0.078** | 0.388±0.096 | **0.868±0.034** |
| | | RA | 0.218 | 0.543±0.019 | **0.609±0.050** | 0.719±0.026 | **0.810±0.074** |
| | Known | PE | 0.007 | **0.609±0.001** | 0.486±0.146 | 0.986±0.006 | **0.988±0.001** |
| Thyroid | Unk. | Hyper. | 0.023 | 0.565±0.465 | **0.817±0.039** | 0.822±0.201 | **0.964±0.011** |
| | Known | Sub. | 0.053 | 0.512±0.380 | **0.751±0.020** | 0.856±0.192 | **0.985±0.003** |
| Arrhyth. | Unk. | All | 0.751 | **0.854±0.030** | 0.846±0.003 | **0.630±0.067** | 0.627±0.010 |
| MVTec (Image) | Unk. | Bottle | 0.683 | 0.996±0.001 | **0.997±0.000** | 0.991±0.002 | **0.994±0.001** |
| | | Cable | 0.577 | 0.795±0.013 | **0.868±0.005** | 0.733±0.019 | **0.807±0.007** |
| | | Capsule | 0.789 | 0.908±0.005 | **0.947±0.002** | 0.753±0.012 | **0.839±0.003** |
| | | Carpet | 0.714 | 0.998±0.000 | **0.999±0.000** | 0.996±0.001 | **0.998±0.000** |
| | | Grid | 0.682 | 0.999±0.000 | **1.000±0.000** | 0.998±0.000 | **0.999±0.000** |
| | | Hazelnut | 0.565 | **0.954±0.004** | 0.943±0.009 | 0.919±0.008 | **0.920±0.013** |
| | | Leather | 0.695 | 1.000±0.000 | 1.000±0.000 | 1.000±0.000 | 1.000±0.000 |
| | | Metal Nut | 0.756 | **0.982±0.001** | 0.975±0.006 | **0.938±0.004** | 0.916±0.017 |
| | | Pill | 0.817 | 0.925±0.002 | **0.950±0.009** | 0.713±0.001 | **0.801±0.025** |
| | | Screw | 0.699 | 0.839±0.044 | **0.844±0.033** | 0.696±0.064 | **0.708±0.051** |
| | | Tile | 0.670 | 0.980±0.007 | **0.997±0.000** | 0.970±0.010 | **0.994±0.001** |
| | | Transistor | 0.333 | 0.786±0.026 | **0.872±0.010** | 0.826±0.007 | **0.875±0.009** |
| | | Wood | 0.732 | 0.984±0.008 | **0.991±0.008** | 0.957±0.015 | **0.974±0.024** |
| | | Zipper | 0.758 | 0.995±0.002 | **0.998±0.001** | 0.985±0.005 | **0.993±0.004** |
| Adv-Bench (Text) | Unk. | satnews | 0.082 | **0.798±0.028** | 0.232±0.030 | **0.977±0.004** | 0.793±0.029 |
| | | CGFake | 0.130 | **0.097±0.004** | 0.080±0.002 | **0.390±0.027** | 0.242±0.019 |
| | | jigsaw | 0.130 | 0.185±0.040 | **0.340±0.011** | 0.637±0.044 | **0.697±0.008** |
| | | EDENCE | 0.113 | 0.102±0.008 | **0.721±0.082** | 0.473±0.029 | **0.941±0.017** |
| | | FAS | 0.140 | 0.087±0.002 | **0.126±0.007** | 0.247±0.015 | **0.494±0.032** |
| | Known | LUN | 0.074 | **0.762±0.032** | 0.532±0.027 | **0.972±0.004** | 0.918±0.010 |
| | | amazon_lb | 0.107 | 0.123±0.017 | **0.824±0.036** | 0.612±0.052 | **0.968±0.006** |
| | | HSOL | 0.030 | 0.042±0.009 | **0.731±0.015** | 0.652±0.044 | **0.961±0.003** |
| | | assassin | 0.022 | 0.048±0.003 | **0.533±0.062** | 0.761±0.016 | **0.931±0.002** |
| | | enron | 0.080 | 0.211±0.037 | **0.361±0.015** | 0.765±0.010 | **0.820±0.013** |

where $\widehat{C}_{q,\alpha,d,r}$ is a constant depending on $q, \alpha, d, r$.

Finally, by taking $\tau = N^{-\frac{\alpha}{d}}$, the restrictions $\tau \geq \max\left\{\frac{1}{n}, \frac{1}{n^-}, \frac{1}{n'}\right\}$ and $\tau \leq \min\{N2^{-m}+N^{-\frac{\alpha}{d}}, 1\}$ are both satisfied. This is because $\max\left\{\frac{1}{n}, \frac{1}{n^-}, \frac{1}{n'}\right\} = \frac{1}{n_{\min}}$ and

$$\tau = N^{-\frac{\alpha}{d}} \geq 2^{-\frac{\alpha}{d}}\left(\frac{n_{\min}}{(\log n_{\min})^4}\right)^{-\frac{\alpha}{d+\alpha(q+2)}} \geq 2^{-\frac{\alpha}{d}}(n_{\min})^{-\frac{\alpha}{d+\alpha(q+2)}} \geq 2^{-\frac{\alpha}{d}}\frac{1}{\sqrt{n_{\min}}}.$$

This conclude the proof.

# H EXPERIMENTS

We run experiments thrice and report the mean and standard deviation in all tables. Randomness is based on model initialization and data draws (for train/validation splits and synthetic anomalies).

We proceed to provide details on other experimental results for reproducibility and further discussion.

## H.1 ADDITIONAL EXPRIMENTAL RESULTS ON AUROC

Here, we further evaluate the performance of the vanilla classification (VC) model with and without synthetic anomalies using the area under the receiver operating characteristic curve (AUROC) across all five datasets: NSL-KDD, Thyroid, Arrhythmia, MVTec, and AdvBench. The results are

Table 4: Dataset details. Dimension is after one-hot encoding or feature extraction. Number of training data is rounded off to the nearest significant figure. The anomaly ratio represents the proportion of anomalies relative to the total number of training samples. We evaluated a range of anomaly ratios across datasets, including approximately 0.0008 (KDD), 0.05 (Thyroid), 0.23 (Arrhythmia), 0.09 (MVTec), and 0.59 (AdvBench).

| Dataset | Data Type | Domain | Dimension | Num. Training Data | Anomaly Ratio |
|---------|-----------|--------|-----------|--------------------|--------------| 
| NSL-KDD | Tabular | Cybersecurity | 119 | 70,000 | 0.0008 |
| Thyroid | Tabular | Medical | 21 | 4,000 | 0.05 |
| Arrhythmia | Tabular | Medical | 279 | 300 | 0.23 |
| MVTec | Image | Manufacturing | 1024 | 200-400 | 0.09 |
| AdvBench | Language | Harmful Text | 384 | 100,000 | 0.59 |

summarized in Table 3, which also includes the previously reported AUPR evaluations from Table 1a to facilitate direct comparison with the AUROC metrics. Overall, the AUROC results generally align with the AUPR findings. On the NSL-KDD dataset, for known anomalies, VC-SA performs slightly worse than VC, while for unknown anomalies, VC-SA shows substantial improvements, especially as VC falls below random performance on DoS and probe attacks. On the Thyroid dataset, VC-SA outperforms VC for both known and unknown anomalies, whereas on the Arrhythmia dataset, VC-SA achieves roughly the same performance as VC for unknown anomalies.

For the MVTec dataset, the results show that the VC-SA consistently outperforms the VC model in terms of both AUPR and AUROC across all anomaly types. The AUPR values for VC-SA are extremely close to 1.0 in nearly every category (e.g., Bottle: 0.997 vs. 0.996, Grid: 1.000 vs. 0.999), and AUROC shows a similar trend, with VC-SA achieving near-perfect accuracy (e.g., Leather: 0.994 vs. 0.991, Zipper: 0.993 vs. 0.985). This demonstrates that adding synthetic anomalies (SA) substantially boosts the model's ability to detect true anomalies accurately.

For the AdvBench dataset, the AUPR values show a varied performance between VC and VC-SA across anomaly types, but the trend in AUROC mostly mirrors the AUPR results. For example, in the case of satnews, both AUPR (0.232 vs. 0.798) and AUROC (0.793 vs. 0.977) follow the same pattern where VC outperforms VC-SA. Similarly, for LUN, the VC model has higher AUPR (0.762 vs. 0.532) and AUROC (0.918 vs. 0.964). Overall, while performance differences exist, AUROC trends generally align with AUPR outcomes in AdvBench.

## H.2 Datasets

Here, we present more details on the datasets we evaluate on. All numerical variables are normalized to be within 0 and 1 during training. For categorical variables, we one-hot encode them and sample from this discrete space (rather than in continuous space). During testing, if a datum contains a variable that is out-of-domain (i.e., not within 0 and 1 after normalization for numerical variables, or an unseen category for categorical variables), we can automatically assign the datum as anomalous. We do not encounter this trivial case during our experiments, so do not discuss this further. Details of each dataset are summarized in Table 4.

### H.2.1 Tabular Data

NSL-KDD[2], our cybersecurity benchmark dataset, has benign (normal) network traffic and 4 types of attacks (anomalies) during training and testing: Denial of Service (DoS), probe, remote access (RA), and privilege escalation (PE). To simulate semi-supervised AD, we use RA as known anomalies and the other 3 as unknown anomalies. We choose RA because it has the fewest training anomalies, allowing us to stress test our method.

Thyroid[3] is a medical dataset with patient vitals. Besides normal data, there are also hyperfunction and subnormal anomalies. We select subnormal thyroid patient vitals to be known anomalies, while hyperfunction thyroid patient vitals are left as the unknown anomalies.

---

[2]Licensed to "redistribute, republish, and mirror" with reference to Bergmann et al. (2019).
[3]CC-BY 4.0 license.

Arryhthmia[4] is a medical dataset with patient vitals. There are patients with normal and various kinds of anomalous heartbeats. We chose left and right BBB as our known anomalies, and the rest as unknown anomalies. We split the data into train and test datasets with a 0.2 test split. This dataset is small (see Table 4), so we make a few workarounds. First, for missing data, instead of dropping data, we use mean imputation, a common imputation method (see Xiao & Fan (2024)). Second, comparing unknown anomaly categories with few anomalies can have much noise and may not be meaningful, so we group all unknown anomaly categories into 1 "unknown anomaly" category during evaluation. Third, we use all known anomalies during training, so we do not evaluate on known anomalies for testing.

### H.2.2 IMAGE DATA

MVTec[5] is an industrial manufacturing image dataset. It contains images of manufacturing items. These images include mostly properly manufactured products (i.e., normal images) and products with defects (i.e., anomalous images) that are grouped by the type of defect. To simulate semi-supervised AD, we include a type of defect along with the original working product for 14 different items. For standardization, we pick the first defect type according to alphabetical order. Due to the small dataset size (see Table 4), once again, we use all known anomalies during training and exclude known anomaly evaluation during testing. To convert image data to tabular form, we use 1024-dimensional DINOv2[6] (Oquab et al., 2023) (frozen) embeddings, like in Lau et al. (2024b). We use these frozen embeddings rather than fine-tuning the feature extractor to avoid overfitting.

### H.2.3 LANGUAGE DATA

AdvBench[7] is a union of 10 smaller text datasets. Each data sample in the dataset corresponds to a sentence, and with it, a corresponding binary label of harmless (i.e., "normal") or harmful (i.e., "anomalous"). There are 5 types of harms (anomalies): misinformation, disinformation, toxic, spam and sensitive, each of which describe 2 of the datasets (for a total of 10 datasets). To convert text into tabular form, we use 384-dimensional BERT[8] (Reimers & Gurevych, 2019) (frozen) sentence embeddings. Comments on using language models zero-shot for AD can be found in Yang et al. (2025). As a remark, this language dataset is by far the most different from the original tabular setting we described. Language comprises discrete tokens that are strung together sequentially, so representing it in tabular form is not a trivial task (let alone, a representation that is compact). Furthermore, defining what is harmful (i.e., anomalous) is highly contextual. For instance, a piece of misinformation may not be toxic, and a toxic text may not be misinformation. Due to this under-constrained nature of detecting harm, we use normal data from all datasets and harmful texts from 1 dataset from each category during training.

### H.3 COMPUTE RESOURCES

To enhance reproducibility, we run experiments that a standard consumer-grade workstation can run. We use 128 GB of memory. We run neural networks on a single NVIDIA GeForce RTX 4090 GPU. Each experimental run takes at most one day to run, but usually within one hour. To run all experiments (including those without reported results), fewer than 10 days of active compute is estimated to be needed.

### H.4 COMPOSITE METHODS

Composite methods first do unsupervised AD to identify data that belong to the training classes, and then binary classifiers differentiate normal from known anomalous data given that the data are known. Concretely, we obtain a score from unsupervised AD on $P(\mathbf{x}$ belongs to training class$)$ and a score from binary classifiers on $P(\mathbf{x}$ normal$|\mathbf{x}$ belongs to training class$)$. Then, we can obtain a score of

---

[4]CC-BY 4.0 license.

[5]CC BY-NC-SA 4.0 license.

[6]Apache License.

[7]Dataset has no license, provided by a Google drive link on `https://github.com/thunlp/Advbench`. Paper is published in EMNLP, which has a CC-BY 4.0 license.

[8]Apache License.

$P(\mathbf{x} \text{ normal}) = \mathbb{P}(\mathbf{x} \text{ from training class})P(\mathbf{x} \text{ normal}|\mathbf{x} \text{ from training class})$. We test kernel-based, tree-based and neural network methods: for unsupervised AD, we use OCSVM (Schölkopf et al., 1999), Isolation Forest (Liu et al., 2008) and DeepSVDD (Ruff et al., 2018); for binary classifiers, we use SVM (Cortes & Vapnik, 1995), Random Forests[9] (Ho, 1995; Breiman, 2001) and neural networks. We multiply the score (shifted to be positive) from each unsupervised AD method with each binary classifier for a composite model (e.g., OCSVM with Random Forest), which produces $3 \times 3 = 9$ types of composite models. One limitation is that some models (e.g., SVM models, Isolation Forest) predict scores which are not probabilities. Nevertheless, we can still use their score to preserve the ranking of anomaly scores. The area under the precision-recall curve (AUPR) metric we use is a threshold-agnostic metric that allows us to evaluate the separability of anomalies from normal data in the output space. Although this two-stage approach is logical, it leads to an undesirable model: normal and known anomalies are modeled as similar, while known and unknown anomalies are modeled as different by unsupervised AD. Ideally, we would like to model the problem as how the problem arises: the normal class is different from all the anomalies (Ruff et al., 2020), while anomalies (known or unknown) may or may not be similar (Ruff et al., 2021).

**Composite Models are as poor as Random**   For brevity, we omit the results of composite models because they all have random performance for both known and unknown anomalies in both NSL-KDD and AdvBench datasets, which we used for preliminary experiments. Although they were designed to inherit the benefits of both unsupervised AD and binary classifiers, they instead inherited errors from both.

In addition, to evaluate the reasonability of our method of composing scores, we ablated against different ways of composing scores: (1) normalizing scores to within 0 and 1 (which produces a score that resembles a probability score) and (2) normalizing scores by their sample variance. However, these other approaches to compose the output of models still produce random results. Hence, the straightforward approach of composing scores from unsupervised AD and binary classifiers does not seem like the the right step forward. It is not unreasonable to believe that the composite models perform badly because the anomaly scores of different models are fundamentally different. The difference could lead to an inability to combine the scores, or that the scores are not cohesive due to a lack of integrated optimization.

## H.5   OUR METHOD

The full details can be found in our code implementation, which uses PyTorch for neural network implementation. Section 4 outlines our training process, with a visualization on the right of Figure 2. To provide greater clarity, we proceed to be more explicit in how we implement our method.

We set up a binary classification task between the normal and anomaly class. For training, the normal class has normal data, while the anomaly class has known and synthetic anomalies. Before training starts, synthetic anomalies are sampled uniformly from the data support $\mathcal{X}$. Specifically, we normalize all features to be between 0 and 1 and sample synthetic anomalies from $[0, 1]^d$ (note that for one-hot encoded variables, we sample from the discrete support instead). The base model is a ReLU network that predicts a scalar from 0 to $1^{10}$ that describes how anomalous each datum is. Model weights are optimized via gradient descent on the loss.

During testing, the trained network predicts a scalar from 0 to 1 for normal data, known and unknown anomalies.

In the following two subsections, we outline 2 kinds of hyperparameter choices: how we represented some theoretical details that are impractical, and other key hyperparameters which may have differed from the theoretical description.

---

[9]We train SVM and Random Forest with balanced class weights.

[10]In our theoretical model, the output is from -1 to 1, but the output can always be scaled according to match user preferences.

### H.5.1 IMPLEMENTING THEORY

Our theoretical model in Definition 4.3 has sparsity constraints and have all absolute values bounded by 1. These constraints are difficult to model during optimization. Hence, we use weight decay during gradient descent to model the preference towards sparser and smaller solutions.

Without knowledge of parameters like $\alpha$, we do not have knowledge on what to set depth $L^*$ and width $w^*$ to. In particular, for width, our theoretical model seeks a width of $6(d + \alpha)N$. However, $N$ can be $\mathcal{O}(e^d)$, which makes implementation computationally infeasible (e.g., $d = 119$ for NSL-KDD, so $N > 10^{51}$ and the width will be too large for training). Instead, in our experiments, we used the dimension of the input $d$ and the number of data samples $M$ to guide our choice of the width — higher dimension and more data should have wider networks for expressivity. We chose widths of 678 for NSL-KDD ($d = 119$, $M \approx 70,000$), 200 for Thyroid ($d = 21$, $M \approx 4,000$) and 500 for Arrhythmia ($d = 278$, $M \approx 300$), 6000 for MVTec ($d = 1024$, $M \approx 300$) and 6000 for AdvBench ($d = 384$, $M \approx 100,000$). We comment on depth in the next subsection.

### H.5.2 SYNTHETIC ANOMALY COUNTS $n'$ AND OTHER HYPERPARAMETERS

For most datasets, the number of synthetic anomalies we use is $n' = n + n^-$, which equals to the number of real training data points (i.e., normal data and known anomalies). The choice $n' = n + n^-$ follows from our main theorem (Theorem 4.5), which shows the convergence rate of the excess risk depending on $n_{\min} = \min\{n, n^-, n'\}$. This means that increasing $n'$ further beyond $n$ or $n^-$ does not further improve the convergence rate. However, to ensure that $n_{\min}$ is not limited by $n'$, we set $n' = n + n^-$ in our experiments so that $n_{\min} = \min\{n, n^-\}$. Also, we would like to avoid adding too many to avoid diluting the known anomaly supervision signal and contaminating the normal data during training.

Our ablations in Table 2 show that the choice of $n'$ is not that sensitive for unknown anomaly performance, achieving a similar AUPR for $n' = 0.001r, r, 5r$ where $r = n + n^-$. Such results show that this parameter is generally not too sensitive. For AdvBench, we notice that validation performance is not perfect, denoting that some anomalies are probably difficult. To avoid overfitting to known anomalies, we further increase the number of synthetic anomalies to $n' = 3 \cdot (n + n^-)$.

In our initial experiments, we observe that the network does not converge during training and remains at high loss. We identify this as a symptom of vanishing (or zero) gradient. To alleviate this symptom, we make the following 3 changes. First, we make our network shallower, similar to the idea of structural risk minimization; our theoretical model seeks a depth of $8 + (m + 5)(1 + \log_2(\max(d, \alpha))) \geq 8 + 5 \cdot 2 = 18$, but we instead choose a depth of 3. The neural network depth is the main hyperparameter we tune. Second, we change out ReLU activations with leaky ReLU activations to avoid dead neuron problems. Third, instead of hinge loss, we use logistic loss to avoid zero loss on some samples. Models are then trained with early stopping on the validation loss.

These changes help with learning. We believe that the first change is the most significant, but keep the second and third change as well. This flexibility of switching out hyperparameters shows that our theory is not bound by specific hyperparameter choices and can be applied in general to standard binary classifiers.

## H.6 ARE SYNTHETIC ANOMALIES REPRESENTATIVE OF UNKNOWN ANOMALIES?

Distributionally, the answer is no — our synthetic anomalies are drawn from a uniform distribution, while unknown anomalies can be drawn from any arbitrary distribution (which is likely more concentrated than the uniform distribution). In other words, good performance on synthetic anomalies does not necessitate good performance on unknown anomalies. The extreme case is when unknown anomalies are adversarial samples deliberately crafted to evade detection. In general, though, we observe that the model does not perform as well on unknown anomalies as it does on synthetic anomalies, even if these unknown anomalies are not deliberately crafted to be adversarial. Our validation AUPR with normal data, known and synthetic anomalies is always high (usually perfect) but AUPR on unknown anomalies can be low (e.g., models generally struggle on AdvBench dataset).

Nevertheless, our goal is not to create synthetic anomalies that resemble unknown anomalies — if unknown anomalies were known, we could directly do binary classification. Rather, we build on

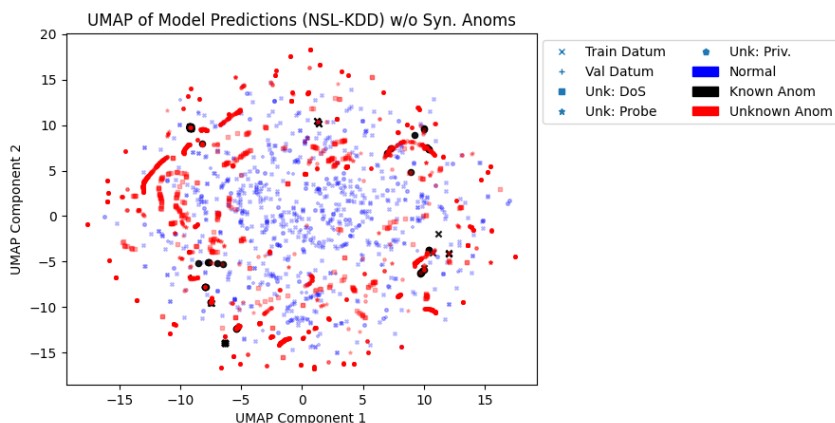

Figure 8: **Data visualization** of the model trained without synthetic anomalies. Unknown anomalies are more scattered than the model trained with synthetic anomalies.

density level set estimation to detect unknown anomalies (Section 3). This is why we use validation loss and not validation performance for early stopping. Geometrically, we hope that synthetic anomalies can label the unknown regions / "open space" (i.e., regions without data) as anomalous, so models will be trained to classify those regions as anomalous. Then, when unknown anomalies appear in these regions, the model will be trained to identify these unknown anomalies as anomalous. The trade-off is that, apriori, we are unaware of which regions these unknown anomalies will appear in, so we will need to sample many synthetic anomalies to fill up more of these unknown regions (especially in higher dimensions), and hence will also require more model expressivity. To illustrate, in our data visualization in Figure 3, synthetic anomalies are scattered everywhere, some of them in regions (left of image) where real data are absent. Only a portion of synthetic anomalies are around the region of unknown anomalies (right of image).

Nonetheless, we observe that unknown anomalies cluster (e.g., bottom left, top of Figure 3), some quite closely to known anomalies. This clustering is not found in the model trained without synthetic anomalies (Figure 8). Here, our model has learnt discriminative features that are similar across real (known and unknown) anomalies.

