# OpenReview forum: "Bridging Unsupervised and Semi-Supervised Anomaly Detection: A Provable and Practical Framework with Synthetic Anomalies"
_ICLR.cc/2026/Conference — Submitted to ICLR 2026_

### Official Review · Reviewer_DuyK · 2025-10-30

**Soundness:** 3
**Presentation:** 3
**Contribution:** 3
**Rating:** 6
**Confidence:** 3

**Summary:**

This paper demonstrates that using synthetic anomalies improves the performance of semi-supervised anomaly detection.
It first formulates anomaly detection as a binary classification problem,
then shows why training a model using only normal data and known anomalies is difficult.
The proposed method theoretically resolves this issue by incorporating synthetic anomalies generated from a uniform distribution in addition to known anomalies.
Experiments on tabular, image, and text data show that adding synthetic anomalies enhances the performance of semi-supervised anomaly detection.

**Strengths:**

- By defining a unified anomaly detection framework based on binary classification, this paper can handle both unsupervised and semi-supervised settings. Based on this, this paper also theoretically justify the use of synthetic anomalies.
- The experimental results are very strong. Across a variety of methods and datasets, incorporating synthetic anomalies leads to improved performance.

**Weaknesses:**

Please see the Questions section.

**Questions:**

- In the proposed method, noise drawn from a uniform distribution is used as synthetic anomalies. However, since normal data also is a subset of a uniform distribution, would not the synthetic anomalies contain normal data as well? If so, I would expect the detection performance for normal data to drop. Why is the proposed method able to avoid this? (For example, DROCC also uses synthetic anomalies, but it includes mechanisms to avoid overlapping with normal data. That approach feels more natural to me; yet for images and text, adding uniform noise to DROCC actually improves performance.)
- This paper uses autoencoders (AEs) in the experiments, but how about trying DeepSVDD? For tabular data, AEs may be better, but for image data I expect DeepSVDD to yield stronger results. I am interested in how the proposed method would perform within DeepSVDD-based variants such as DROCC, ABC, and DeepSAD.

---

### Official Review · Reviewer_SFnp · 2025-11-01

**Soundness:** 2
**Presentation:** 1
**Contribution:** 2
**Rating:** 4
**Confidence:** 4

**Summary:**

This paper proposes to add synthetic anomalies to diversify the collected anomalies under the semi-supervised setting. Authors connect anomaly detection with binary classification and introduces synthetic anomalies to mitigate two issues in semi-supervised AD: false negative modeling and insufficient regularity of learning. Some theoretical analyses are provided to justify the effectiveness of incorporating synthetic anomalies. Experiments across tabular, image, and text datasets  demonstrate the applicability of the proposed framework.

**Strengths:**

1.  Some theoretical analyses are conducted on the effectiveness of introducing synthetic anomalies for semi-supervised anomaly detection.

2. The experiments span diverse modalities (tabular, image, text) and multiple AD methods, showing general applicability of the “synthetic anomaly” principle.

**Weaknesses:**

1. Formulating anomaly task as binary classification is fundamentally inappropriate. Since the type of anomaly is uncountable, anomaly detection is usually formulated as one-class classification to model the distribution of normal data or to learn the pattern of them. Using binary classifier may learn a unreliable decision boundary.

2. The theoretical analysis of convergence is narrow to the network using ReLU as activation function. Extending the theoretical guarantees to broader architectures or activation functions would significantly strengthen the generality and impact of the results.

3. The novelty of this paper is weak. While the theoretical framing is elegant, the core idea is adding synthetic anomalies, which is not new. The main contribution lies in extending this idea to a semi-supervised setting, which feels incremental and does not substantially push the frontier of anomaly detection research.

4. Synthetic anomaly generation is overly simplistic. The use of uniformly random noise as synthetic anomalies is questionable, especially for complex or high-dimensional data. This weakens the practical significance of the framework and may not generalize to high-dimensional or structured data. There is no comparison with more informative or adaptive anomaly generation methods.

5. The presented ablation resembles a sensitivity analysis rather than a comprehensive investigation.

6. The writing of this work is terrible and should be significantly improvoed.

**Questions:**

1. How sensitive is the framework to the way synthetic anomalies are generated? Would a more structured generator improve performance?
2. It is possible to generalize the theoretical guarantees to broader architectures or activation functions?

---

> ### Author Response · Authors · 2025-11-25
> **Response to reviewer  SFnp**
>
> We sincerely thank the reviewer for their constructive comments. We respond to each point separately in the following.
>
> **Weakness 1**: *"Formulating anomaly task as binary classification is fundamentally inappropriate. Since the type of anomaly is uncountable, anomaly detection is usually formulated as one-class classification to model the distribution of normal data or to learn the pattern of them. Using binary classifier may learn a unreliable decision boundary."*
>
> **Response**: We fully understand that anomaly detection, unlike standard binary classification, focuses  on distinguishing normal from not normal. There may be uncountably many types of anomaly, but as long as the model correctly identifies the boundary of normal behavior, it can flag anomalies effectively.
>
>  We would like to clarify that our proposed method follows this principle. Our approach uses available normal data, real anomalies, and synthetic anomalies to train a classifier whose decision function represents the decision boundary surrounding the normal region.
>  Importantly, the synthetic anomalies are not designed to mimic real anomalies. Their purpose is to provide contrastive signals that help the model better distinguish what lies outside the normal data region.
>
>  Once this decision boundary is established, the model can detect a broad range of unseen anomalies. In this sense, our formulation remains aligned with the core principle of one-class anomaly detection (i.e., characterizing normality) while leveraging binary classification only as a practical mechanism for boundary learning.
>
> **Weakness 2 & Question 2**: *"The theoretical analysis of convergence is narrow to the network using ReLU as activation function. Extending the theoretical guarantees to broader architectures or activation functions would significantly strengthen the generality and impact of the results."*
>
> *"Is it possible to generalize the theoretical guarantees to broader architectures or activation functions?"*
>
> **Response**: Thank you for your valuable suggestions! You're right; our theoretical analysis focuses on networks using ReLU activation function, as their approximation properties allow us to derive clear, tractable convergence rate.
>
> We'd like to emphasize that, to the best of our knowledge, our work provides the first theoretical convergence analysis for semi-supervised anomaly detection using neural networks of any kind. Given the novelty and technical depth of this contribution, we aimed to present a rigorous foundation by concentrating on the ReLU case.
> That said, the theoretical framework is not fundamentally restricted to ReLU. In principle, it can be extended to other activation functions, including sigmoid-type functions such as sigmoid and tanh. However, these activations possess distinct approximation properties, meaning that establishing analogous convergence guarantees would require substantially different analysis.
>
> Given the scope of the current paper, we elected not to pursue these extensions here. Nevertheless, we fully recognize the value of generalizing the theory to broader architectures and activation functions, and we look forward to exploring these directions in future work.
>
> **Weakness 3**: *"The novelty of this paper is weak. While the theoretical framing is elegant, the core idea is adding synthetic anomalies, which is not new. The main contribution lies in extending this idea to a semi-supervised setting, which feels incremental and does not substantially push the frontier of anomaly detection research."*
>
> **Response**: We thank the reviewer for acknowledging that our theoretical framing is elegant. We also agree that the core idea of adding synthetic anomalies to support anomaly detection is not new. As discussed and cited in our manuscript, several previous works, including (Sipple, 2020) and others, have explored related strategies.
>  However, what has been largely missing in the literature is a theoretical understanding of why synthetic anomalies help and under what conditions they improve learning. This theoretical gap is precisely what our work aims to fill. We do not claim to have introduced the concept of sampling synthetic anomalies. Instead, our contribution is to provide, to the best of our knowledge, the first rigorous analysis demonstrating that adding synthetic anomalies during training can guarantee improved regularity of the regression function and, as a consequence, yield fast convergence rate for the excess risk. This theoretical guarantees represent the core novelty of our work, rather than the sampling mechanism itself.

---

> > ### Author Response · Authors · 2025-11-25
> > **Response to reviewer SFnp (part 2)**
> >
> > **Weakness 4**: *"Synthetic anomaly generation is overly simplistic. The use of uniformly random noise as synthetic anomalies is questionable, especially for complex or high-dimensional data. This weakens the practical significance of the framework and may not generalize to high-dimensional or structured data. There is no comparison with more informative or adaptive anomaly generation methods."*
> >
> > **Response**: Thank you for your thoughtful comment pointing out the simplicity of our proposed method. We'd like to clarify that the simplicity of uniform synthetic anomalies is intentional and central to our contribution.
> > A core message of our work is that even simple synthetic anomalies, such as uniformly sampled noise, can be theoretically and empirically effective. Our analysis shows that such uniformly generated anomalies are sufficient to help a ReLU network classifier learn a provably optimal decision boundary around the normal data.
> >
> > We'd emphasize once again that our method generates synthetic anomalies not with the intention of resembling
> > real anomalies, but to provide a contrastive signal to learn the decision boundary of normal
> > data.
> > In many real-world applications, the distribution of true anomalies is highly complex, high-dimensional (as you mentioned), and continuously evolving. It is often unrealistic to assume prior knowledge of the true
> > anomaly distribution. In particular in cybersecurity, new types of cyber attack emerge every
> > day, designed to bypass detection systems trained on past data.
> > The strength of our approach lies precisely in working without any prior knowledge of the anomaly distribution.
> > Thus, the use of uniformly random synthetic anomalies is not a limitation but rather a design choice that enhances generality and robustness.
> >
> > We understand the reviewer’s concern about whether such a simple synthetic-anomaly mechanism can perform well in high-dimensional data. We conducted experiments on the MVTec dataset, a popular benchmark for image anomaly detection. Across all categories, training our network with uniformly generated synthetic anomalies consistently improved detection accuracy compared to training without them. We note that, after embedding each image into a feature vector, the data dimension is 1024. This evidence supports the effectiveness of our approach even dealing with complex, high-dimensional data.
> >
> > **Weakness 5**: *"The presented ablation resembles a sensitivity analysis rather than a comprehensive investigation."*
> >
> > **Response**: We thank the reviewer for this comment. Our ablation study (as shown in Table 2) varies one factor of our method at a time: network width, network depth,  or the number of synthetic anomalies. This design is intentional, with the purpose to isolate the specific effect of each hyperparameter on performance. This is a standard practice in machine learning research.
> >
> >  We agree that a more exhaustive investigation, such as studying the joint interactions among all hyperparameters could provide additional insights. However, this is beyond the scope of the current paper, especially because the current ablations already capture the key behaviors relevant to our theoretical claims. For example, we can see that deeper networks improve expressivity but risk vanishing gradients,
> > width has relatively minor impact, and
> > even a small amount of synthetic anomalies consistently improves unknown anomaly detection.
> >
> >  **Weakness 6**: *"The writing of this work is terrible and should be significantly improved"*
> >
> >  **Response**: We appreciate the reviewer’s candid feedback. The space constraints of the submission format made parts of the presentation more compressed than ideal. If the reviewer could point to specific sections that were unclear, we would be grateful for the guidance and happy to refine them.

---

> > > ### Author Response · Authors · 2025-11-25
> > > **Response to reviewer SFnp (part 3)**
> > >
> > > **Question 1**: *"How sensitive is the framework to the way synthetic anomalies are generated? Would a more structured generator improve performance?"*
> > >
> > > **Response**: This question is related to the previous Weakness 5. We first recall that
> > > $n$ denotes the number of normal samples, $n^-$ denotes the number of real anomalies, and $n^\prime$ denotes the number of synthetic anomalies we choose to generate. We also let $r = n + n^-$.
> > > Our ablation study in Table 2 varies $n^\prime$ while keeping other factors fixed. When no synthetic anomalies are used, the AUPR for unknown attacks (DoS / Probe / RA) is only about
> > > 0.345/0.180/0.543.
> > >  Adding only a small amount of synthetic anomalies ($n^\prime = 0.001r$) already boosts the corresponding AUPR to
> > > 0.744/0.651/0.598, which is very close to the performance for $n=r$ (our default setting) and $n=5r$.
> > > Overall across
> > > $n^\prime\in (0.001r,r,5r)$, the detection accuracy stays within a relatively narrow range, indicating that our framework is not highly sensitive to the exact amount of synthetic anomalies as long as some are present. Only when $n^\prime$ becomes very large ($n^\prime=20r$) do we observe slight degradation.
> > >
> > >  Regarding whether a more structured generator could further improve performance, this is possible and is a meaningful direction for future work. However, our goal in this paper is to show that a very simple, distribution-agnostic generator (uniform noise) is already sufficient to obtain strong empirical performance and rigorous theoretical guarantees, without relying on any assumptions or prior knowledge about the true anomaly distribution.
> > >
> > > Reference:
> > > John Sipple. Interpretable, multidimensional, multimodal anomaly detection with negative sampling for detection of device failure. In International Conference on Machine Learning (ICML), pp. 9016–9025. PMLR, 2020.

---

> > ### Comment · Reviewer_SFnp · 2025-11-26
> >
> > I thank the authors' detailed responses to my questions. However, my main concerns  haven't been addressed, which including the following aspects: 1) the appropriateness of using binary classifier; 2) the usefulness of the theoretical analyses; 3) the overly simpleness of using uniformly random noise as synthetic anomalies; 4) The poor presentation of the paper. I still hold my original viewpoint and keep my rating unchanged.

---

> > > ### Author Response · Authors · 2025-11-30
> > > **Response by authors**
> > >
> > > We thank the reviewer for acknowledging our earlier rebuttal. We would like to respond to reviewer's new comments below:
> > >
> > > 1. Response to point (1): *"the appropriateness of using binary classifier"*
> > >
> > > We respectfully note that this concern has been addressed in our earlier response.
> > > Our method is not simply solving a standard binary classification problem. As we previously clarified, the classifier in our framework is used specifically to learn the decision boundary around the normal region, which is fully aligned with the fundamental principle of one-class classification. The reviewer’s own statement: *"anomaly detection is usually formulated as one-class classification to model the distribution of normal data or to learn the pattern of them"* precisely describes what our method achieves: the classifier’s decision function explicitly models the boundary of the normal data manifold. The synthetic anomalies are not meant to resemble real anomalies but to provide contrastive signals that help define this boundary. Thus, describing our approach as *"just a binary classifier"* does not reflect the actual purpose or mechanism of the method.
> > >
> > > 2. Response to point (2): *"the usefulness of the theoretical analyses"*
> > >
> > > Our theory directly supports the effectiveness of our proposed method: namely, training a ReLU network classifier using both synthetic and real anomalies. Without these theoretical results, we would lack the rigor and confidence to justify why such a simple method should work or under what conditions it succeeds. Our analysis provides exactly this foundation: it establishes that the resulting decision function achieves minimax optimal excess risk, thereby offering the first theoretical guarantee for semi-supervised anomaly detection with neural networks. This theoretical grounding is essential to understanding why the method is effective and is therefore a central contribution of the paper.
> > >
> > > 3. Response to point (3): *"the overly simpleness of using uniformly random noise as synthetic anomalies"*
> > >
> > > We respectfully point out that this issue was also addressed in our previous rebuttal. Our method is indeed simple, but we disagree that this simplicity is a limitation. The use of uniformly random synthetic anomalies is a deliberate design choice that enhances generality and robustness, and **this simplicity is in fact one of the key strengths of our work**. We have provided **extensive theoretical and empirical evidence supporting its effectiveness**. First, our theory shows that the excess risk of the resulting decision function achieves minimax optimal accuracy. Seoncd, our experiments confirm strong performance even on high-dimensional data such as the MVTec image anomaly detection benchmark. These results collectively demonstrate that uniform synthetic anomalies are not only sufficient, but also advantageous for practical and theoretical reasons.
> > >
> > > 4. Response to point (4): *"poor presentation of the paper"*
> > >
> > > We previously asked the reviewer to kindly indicate which section(s) were unclear so that we could refine them accordingly. We are grateful for any concrete pointers that the reviewer can provide, but we believe that the current presentation is adequate to convey the main ideas and results.

---

### Official Review · Reviewer_n4YP · 2025-11-01

**Soundness:** 1
**Presentation:** 1
**Contribution:** 1
**Rating:** 0
**Confidence:** 4

**Summary:**

The paper deals with semi-supervised anomaly detection. It states that anomaly detection and semi-supervised anomaly detection can be approached as binary classification.

It proposes as algorithmical contribution in section 4.1 to sample from a uniform distribution background to create artificial anomaly samples.

They cite several theoretical results on excess risk convergence over a function class for binary classification. They prove certain special cases.

They perform experiments using relu networks.

**Strengths:**

Negligible in the light of the below weaknesses.

**Weaknesses:**

The algorithmical proposal of this paper, sampling anomaly background ,  is long known (e.g. the paper by Sipple 2020, but it is known before), is trivial and has no novelty. There has been advantages over that, see eg SMOTE, Chawla et al from 2002 .

- The claim that Semi-supervised anomaly detection can be treated as classification is not novel. Steinwart 2005 has established this formally for anomaly detection in general (Corollary 3 and theorem 4 in Steinwart 2005).

Were it not for the old suggestion of sampling negatives and the many mistakes, this paper would feel like a recapitulation of Steinwart 2005, or one tries to confuse the readers over the simplicity of the algorithmical content by citing convergence bounds.

- The paper has a number of severe theoretical mistakes:

1. line 148-149 they state that if a non-negative lower bound converges to zero, then the term which it bounds from below must also converge to zero. Their statement is: $a >= b$, $b>=0$, $b \rightarrow 0$ implies $a \rightarrow 0$

for evidence see  "from [4], we see ..."

2. a similar wrong conclusion occurs in lines 240-244

"From Proposition 3.1 and Theorem 3.3, we can see that if the regression function is discontinuous, the approximation error is high (at least 1), which may lead to vacuous excess risk bounds (i.e., excess risk can be high and is not guaranteed to converge). Lacking theoretical guarantees, the Bayes classifier cannot be effectively learned."

If an upper bound diverges, it does not mean that the quantity bounded by it would have to diverge as well. Same kind of logical mistake as in 1., but now with an upper bound.

3.Proposition 4.2 is obviously wrong. They claim continuitity, however if $h_-(X)$ is discontinuous, then $f_P(X)$ can be discontinuous, too.
E.g. choose $s=0.5, \tilde{s}=0.5, h_1 =c$, then
$f_{P}(x) = \frac{0.5c -0.25 h_-(X) -0.25}{0.5c + 0.25h_-(X) + 0.25}$

4. eq (4) is proven in Steinwart (2005) as an upper bound, see Theorem 10 in Steinwart (2005).
Proving the exact same result a lower bound would be very surprising.
They use exactly the same argument as Steinwart 2005 in the proof of theorem 10, but arrive at the opposite direction of inequality.

If this is corrected to the correct  direction of inequality, their extension is straightforward. Steinwart 2005 assumes for the anomaly density to be $\mu$. They assume that it has density $h_2$ with respect to $\mu$ . There is no technical effort in doing this change.

btw, line 1101 makes a lower bound (it should use 3/5 as constant but this is minor) .


5. Proposition 3.1 is wrong because they do not ensure that $\mu(X_1) >0$ and $\mu(X_-) >0$ . One can choose closed sets such that  $\mu(X_1) =0$ and $\mu(X_-) =0$ Then one can get a zero $\ell_{\infty}$-norm to $f(x)=0$

but even if one would fix that, it would be of no consequence, see point 7

6. It could be that Theorem 4.5 has an unfavourable rate $O( (log n) ^4  / n ) ^{ (c+\alpha) / (c+d) }$

For $c = \alpha q$ is typically small compared to the input dimensionality $d$ if one wants smooth settings as they state it

for even moderate input dimensionalities d the bound is worse than the typical $O(n^{-1/2})$ results .

7. the "insufficient regularity of learning" problem as they state it is no problem for training a classifier:

assume $P[Y=1|X]$ makes a jump in direction orthogonal to the decision boundary, but the decision boundary is a standard hyperplane. This is trivially learnable with 1 layer.

- Ironically, Tsybakovs noise condition, which is repeatedly cited by the submitters of this paper, requires a steepness of $\eta(x) = P(Y=1|X=x )$ around 0.5 for faster convergence rates. They state that this steepness would a problem for learning. This is a direct contradiction to the results from Tsybakov and Steinwart.

- Overall, the paper has very poor readability.

**Questions:**

none

**Details Of Ethics Concerns:**

Gaslighting / obfuscating style of writing:
a very trivial proposition (sampling from the uniform density for anomalies), covered up by lots of math formalisms being actually trivial extensions from Steinwart 2005.

It is very close to Steinwart 2005 for the theoretical results when one removes or corrects the errors.

update: what I had in mind: if these authors submitted another paper, to check that they do not employ the same obfuscating style.

There is a striking contradiction between  the "insufficient regularity of learning" problem as they state it and the results from Tsybakov and Steinwart .

- Ironically, Tsybakovs noise condition, which is repeatedly cited by the submitters of this paper, requires a steepness of $\eta(x) = P(Y=1|X=x )$ for faster convergence rates. They state that this steepness is a problem for learning. This is a direct contradiction to the result from Tsybakov and Steinwart .

How can one dabble with the math of these papers and not get the semantic content of their conditions ? This is unexpected. If a human  is using the math, then the human should have some idea of what it is doing, no ?

This is a paper that makes one want to quit doing research.

---

> ### Author Response · Authors · 2025-11-19
> **Response to reviewer n4YP**
>
> We thank the reviewer for their comments and the time they devoted to evaluating our work. We respond to each point separately in the following.
>
> First of all, we agree that sampling synthetic anomalies to help anomaly detection is not a new idea. As discussed and cited in our manuscript, several previous works (including (Sipple, 2020) and others) have explored related strategies. However, what has been largely missing in the literature is a theoretical understanding of why synthetic anomalies help and under what conditions they improve detection accuracy. This theoretical gap is precisely what our work aims to fill. We did not claim to have introduced the concept of sampling synthetic anomalies. Instead, our contribution is to provide, to the best of our knowledge, the first rigorous analysis demonstrating that adding synthetic anomalies during training can guarantee improved regularity of the regression function and, as a consequence, yield fast convergence rate for the excess risk. This theoretical guarantees represent the core novelty of our work, rather than the sampling mechanism itself.
>
> We will clarify the novelty of our work more in detail in the following, especially in point 9.
>
> 1. Comment: *``line 148-149 they state that if a non-negative lower bound converges to zero, then the term which it bounds from below must also converge to zero. Their statement is: $a \geq b, b \geq 0, b \to 0$ implies $a\to 0$.
> For evidence see "from [4], we see ..."*
>
> Response: We thank the reviewer for pointing this out. The inequality in this sentence indeed contains a typo: the symbol $\geq$ should have been $\leq$. The intended statement in equation should have been:
>
> $0 \leq S_{\mu, h_1, h_2, \rho}(f) \leq C_q (R(f) - R^*)^{\frac{q}{q+1}}.$
>
> It follows that $ S_{\mu, h_1, h_2, \rho}(f) \to 0$
> whenever $R(f) - R^* \to 0$.
>
>
>
> 2. Comment: *``A similar wrong conclusion occurs in lines 240-244 "From Proposition 3.1 and Theorem 3.3, we can see that if the regression function is discontinuous, the approximation error is high (at least 1), which may lead to vacuous excess risk bounds (i.e., excess risk can be high and is not guaranteed to converge). Lacking theoretical guarantees, the Bayes classifier cannot be effectively learned. If an upper bound diverges, it does not mean that the quantity bounded by it would have to diverge as well. Same kind of logical mistake as in 1., but now with an upper bound."*
>
> Response: We thank the reviewer for this comment and would like to clarify our interpretation of Theorem 3.3. Roughly speaking, Theorem 3.3 shows that ``excess risk $\leq$ approximation error."
> We do not claim, and did not intend to suggest, that a large or non-vanishing upper bound forces the excess risk itself to diverge. This is why our wording is carefully qualified: we wrote that a large approximation error **may** lead to vacuous excess risk bounds and that the excess risk **can be high** and **is not guaranteed to** converge. Our point is precisely that, when the regression function is discontinuous and cannot be well approximated by the chosen hypothesis class, Theorem 3.3 alone cannot provide a meaningful convergence guarantee for the Bayes classifier.
>
> 3. Comment: *``Proposition 4.2 is obviously wrong. They claim continuity; however, if
> $h_{-}(X)$ is discontinuous, then $f_P(X)$
>  can be discontinuous, too."*
>
> Response: We'd like to clarify that Proposition 4.2 begins with the line “Suppose the condition stated in Proposition 3.1 holds.” Proposition 3.1 assumes that both density functions $h_1$ and $h_{-}$ are $C^r$-continuous on the domain $\mathcal{X}$.
> Therefore, the scenario raised by the reviewer  — where $h_{-}$ is discontinuous  — falls outside the conditions under which Proposition 4.2 is stated and proved.
>
> Moreover, the text right before Proposition 4.2 explains the intuition: although Proposition 3.1 shows that $f_P$ can be discontinuous even when $h_1$ and $h_{-}$ are $C^r$-continuous; Proposition 4.2 shows that, under the same regularity conditions of Proposition 3.1, sampling synthetic anomalies from a uniform distribution (i.e., setting $h_2 = \tilde{s} h_{-} + (1-\tilde{s}$))
> ensures that $f_P$ is continuous.

---

> > ### Author Response · Authors · 2025-11-20
> > **Response to reviewer n4YP (part 2)**
> >
> > 4. Comment: *``eq (4) is proven in Steinwart (2005) as an upper bound, see Theorem 10 in Steinwart (2005). Proving the exact same result a lower bound would be very surprising. They use exactly the same argument as Steinwart 2005 in the proof of theorem 10, but arrive at the opposite direction of inequality.
> > If this is corrected to the correct direction of inequality, their extension is straightforward. Steinwart 2005 assumes for the anomaly density to be $\mu$. They assume that it has density $h_2$
> >  with respect to $\mu$. There is no technical effort in doing this change.
> > btw, line 1101 makes a lower bound (it should use 3/5 as constant but this is minor) ."*
> >
> > Response: As mentioned earlier in our response to point 1, equation (4) in the manuscript contains a typo: the symbol $\geq$ should have been $\leq$. The intended statement in equation should have been:
> >
> > $0 \leq S_{\mu, h_1, h_2, \rho}(f) \leq C_q (R(f) - R^*)^{\frac{q}{q+1}}.$
> >
> > With the corrected inequality, our bound is fully consistent with the corresponding result in (Steinwart et al., 2005). We will fix this typo in the revised version.
> > We'd also like to emphasize that we were fully transparent about the relationship between our result and the work of (Steinwart et al.,2005). Immediately before equation (4), we explicitly state that we “extend Steinwart et al. (2005) (proven in Appendix E.1) to derive a bound on the AD error,” and we carefully explain how the original formulation is adapted to our anomaly detection setting.
> >
> > 5. Comment: *``Proposition 3.1 is wrong because they do not ensure that $\mu(X_1)>0$ and $\mu(X_{-})>0$. One can choose closed sets such that $\mu(X_1)=0$ and $\mu(X_{-})=0$ Then one can get a zero $\ell_\infty$-norm to $f(x)=0$. but even if one would fix that, it would be of no consequence, see point 7"*
> >
> > Response: We'd like to clarify that Proposition 3.1 explicitly assumes that the density function $h_1$ has support
> > $X_1$ and the density function $h_{-}$ has support $X_{-}$, where $X_1$ and $X_{-}$ are subdomains of $\mathcal{X}$. These assumptions already require that $\mu(X_1)>0$ and $\mu(X_{-})>0$; otherwise, it would not be possible to define valid density functions supported on $X_1$ and $X_{-}$. Thus, the case raised by the reviewer does not fall within the assumptions under which Proposition 3.1 is stated.
> >
> > 6. Comment: *``It could be that Theorem 4.5 has an unfavorable rate $\mathcal{O}\left(((\log n)^4/n)^{\frac{c+\alpha}{c+d}}\right)$.
> > For $c=\alpha q$ is typically small compared to the input dimensionality $d$
> >  if one wants smooth settings as they state it.
> > For even moderate input dimensionalities $d$ the bound is worse than the typical
> >  results $\mathcal{O}\left(n^{-\frac{1}{2}}\right)$."*
> >
> > Response: We respectfully disagree with the assertion that $c = \alpha q$ is ``typically small'' compared to the input dimensionality $d$ in smooth settings. As stated in Theorem 4.5, $\alpha > 0$ denotes the Hölder smoothness index of the regression function $f_P$. This parameter $\alpha$ increases as $f_P$ becomes smoother and, in principle, can be arbitrarily large. Moreover, $q > 0$ is the noise exponent in the Tsybakov's noise condition, which also increases with better class separation and can go to infinity in the case when data from two classes are separable. Consequently, the product $c = \alpha q$ need not be small relative to $d$; it can in fact be comparable to or larger than $d$.
> >
> > Nevertheless, we'd like to emphasize that the excess risk bound we obtain:
> >
> > $\mathcal{O}\left(n^{-\frac{\alpha(q+1)}{d+\alpha(q+2)}}\right)$
> >
> > has long been considered as the minimax benchmark for nonparametric binary classification problem with smooth regression and Tsybakov noise, as noted in the paragraph immediately following Theorem 4.5. More specifically, this rate matches the minimax lower bound established in the classical statistics literature (Audibert \& Tsybakov, 2007), implying that no classifier can achieve a faster rate under these assumptions.
> > In fact, many work in the ML and statistics literature has aimed to design classifiers that match this minimax convergence rate. Examples include adaptive tree-based methods (Scott \& Nowak, 2006),  bipartite ranking methods (Clemencon \& Robbiano, 2011), and neural network classifiers (e.g., Kim et al., 2021; Hu et al., 2022).
> >
> > It is indeed true that the exponent $\frac{\alpha(q+1)}{d+\alpha(q+2)}$ may be smaller than $\frac{1}{2}$, resulting in rates slower than $\mathcal{O}\left(n^{-\frac{1}{2}}\right)$. However, the purpose of expressing the rate in terms of $d$, $\alpha$, and $q$ is precisely to make explicit how each parameter influences the excess risk and to illustrate the  difficulty of the problem under different smoothness, noise level, and dimensionality.

---

> > > ### Author Response · Authors · 2025-11-20
> > > **Response to reviewer n4YP (part 3)**
> > >
> > > 7. Comment: *`` the "insufficient regularity of learning" problem as they state it is no problem for training a classifier:
> > > assume $P[Y=1|X]$ makes a jump in direction orthogonal to the decision boundary, but the decision boundary is a standard hyperplane. This is trivially learnable with 1 layer."*
> > >
> > > Response: The "insufficient regularity of learning" problem we refer to, as indicated in Proposition 3.1, concerns situations in which the regression function $f_P(X) = E[Y=1|X]$
> > > becomes a discontinuous step function taking only the values $1$ or $-1$, that is,
> > >
> > > $f_P(x)=1$ if $x\in X_1$, and $f_P(x)=-1$ if $x\in X_{-}$.
> > >
> > > Here, $X_1$ and $X_{-}$ are two subdomains of $\mathcal{X}$.
> > > This corresponds to the scenario the reviewer describes, in which $P[Y=1|X]$ exhibits a jump across a decision boundary.
> > >
> > > However, we must respectfully but firmly disagree with the reviewer’s comment that such a discontinuous function can be approximated by a one-layer ReLU network. A one-layer ReLU network of the form $\sum_{j=1}^m a_j \sigma(w_j^T x + b_j)$ is a globally continuous piecewise-linear function with $\sigma(u)=\max(0,u)$. Such networks cannot uniformly approximate a discontinuous function over any region in which the discontinuity occurs.
> > >
> > > Thus, when $f_P$ becomes a discontinuous step function, a one-layer ReLU network cannot approximate it to arbitrary accuracy, and training becomes fundamentally constrained by the lack of regularity. This is precisely why we emphasize the importance of ensuring sufficient smoothness in the regression function through our proposed approach.
> > >
> > > 8. Comment: *``Ironically, Tsybakov's noise condition, which is repeatedly cited by the submitters of this paper, requires a steepness of $\eta(x) = P(Y=1|X=x)$
> > >  around 0.5 for faster convergence rates. They state that this steepness would a problem for learning. This is a direct contradiction to the results from Tsybakov and Steinwart."*
> > >
> > > Response: Since our manuscript does not use the term "steepness,'' we are not entirely certain what the reviewer intends by this word. From the context, however, we infer that the "steepness'' of $\eta(x)=P(Y=1\mid X=x)$ is meant to describe how quickly the conditional probability function $\eta$ moves away from $1/2$ in a probabilistic sense, as characterized by Tsybakov's noise condition:
> > >
> > > $P\left(|\eta(x)-\tfrac{1}{2}|\le t\right) < c_0 t^{q}, \qquad \forall t>0.$
> > >
> > > This condition ensures that the probability of $\eta(x)$ remaining close to $1/2$ decays at a rate governed by the noise exponent $q>0$. Larger values of $q$  typically yield faster convergence rates for the excess risk in classification.
> > >
> > >  We'd like to emphasize that this steepness in Tsybakov’s condition is probabilistic, not geometric. It does not impose smoothness on $\eta$, nor does it prevent $\eta$ from having  discontinuity across the decision boundary. The "insufficient regularity of learning" issue we discuss concerns the smoothness of the regression function; specifically, situations where $f_P$ (or equivalently $\eta$) becomes a discontinuous step function. This discontinuity is not related to the noise level
> > > $q$ and is not ruled out by the Tsybakov's noise condition.
> > > For this reason, there is no contradiction between our discussion and the results of Tsybakov and Steinwart. The noise condition controls the measure of the near-boundary region $|\eta(x) -\frac{1}{2}| \leq t$, but it does not guarantee the regularity of $\eta$ or $f_P$.

---

> > > > ### Author Response · Authors · 2025-11-20
> > > > **Response to reviewer n4YP (part 4)**
> > > >
> > > > 9. Comment: *``Gaslighting / obfuscating style of writing: a very trivial proposition (sampling from the uniform density for anomalies), covered up by lots of math formalisms being actually trivial extensions from Steinwart 2005.
> > > > It is very close to Steinwart 2005 for the theoretical results when one removes or corrects the errors."*
> > > >
> > > > Response:
> > > > We firmly and respectfully disagree with the reviewer’s comment of our work as "gaslighting / obfuscating’" or as a "very trivial extension’" of (Steinwart et al., 2005). Our manuscript builds upon the classification-based formulation of anomaly detection introduced in (Steinwart et al., 2005), and we have been fully transparent about this connection throughout the paper. However, our theoretical contributions go well beyond the results in (Steinwart et al., 2005), both conceptually and technically. We highlight some key differences here.
> > > >
> > > > First of all,  Steinwart et al. (2005) and our paper consider fundamentally different function classes to tackle the anomaly detection problem. Steinwart et al. (2005) study support vector machines, a kernel-based method, and all theoretical results in that work are derived in the reproducing kernel Hilbert space (RKHS) framework. In contrast, our paper studies ReLU neural network classifiers, whose function class, approximation properties, and theoretical guarantees differ substantially from those of SVMs.
> > > >
> > > > Moreover, we establish a series of theoretical results that are entirely absent from (Steinwart et al., 2005). Proposition 3.1 shows that in the absence of synthetic anomalies, the regression function can become discontinuous even when both class-conditional densities are continuous. This phenomenon is not discussed or implied in (Steinwart et al., 2005). Theorem 3.3 establishes a direct relationship between the excess risk of a ReLU network classifier and the approximation error of the regression function. No similar result appears in (Steinwart et al., 2005). Proposition 4.2 proves that adding synthetic anomalies sampled from the uniform distribution provably increases the regularity of the regression function, with concrete examples given in the Appendix. Again, no similar result appears in (Steinwart et al., 2005). Our main theorem (Theorem 4.5) establishes that the excess risk of the ReLU classifier trained on normal data, known anomalies, and synthetic anomalies converges to zero at a minimax optimal rate. This level of statistical precision does not appear in (Steinwart et al., 2005), which proves only consistency of SVMs and does not provide convergence rates. We are able to derive explicit convergence rate because we overcome two fundamental theoretical obstacles
> > > > unique to the anomaly detection problem:
> > > >
> > > > **i) Non-i.i.d. nature of the training data**:
> > > >
> > > > In anomaly detection, the normal and anomalous samples are generated from two distinct distributions. This breaks the standard i.i.d. assumption used in most learning theory analyses, making many classical analysis tools inapplicable. We derived novel concentration inequalities in our analysis
> > > > (see, e.g., Lemma G.4 and Lemma G.6 in the Appendix) to accommodate the
> > > > non-i.i.d. nature of our data.
> > > >
> > > > **ii) Approximation error analysis for SVMs vs. neural networks**:
> > > >
> > > > A kernel $K$
> > > > induces an integral operator $L_K$. A common assumption in kernel methods is that the target function lies in the range of the $r$-power of the integral operator $L_K$, a condition
> > > > often referred to as the source condition.
> > > > For classification problems, the target function corresponds to the Bayes classifier,
> > > > which is typically discontinuous and therefore does not satisfy the source condition. For
> > > > example, when K is a Gaussian kernel, the source condition would requires the Bayes
> > > > classifier to be $C^\infty$-continuous, which is very unlikely to hold in practice.
> > > > In contrast, neural networks — particularly those equipped with ReLU activations — are universal approximators that can achieve tight approximation error bounds for a
> > > > broad class of smooth functions. This property is crucial in our analysis, as it enables
> > > > us to establish the minimax-optimal convergence rate of the excess risk.
> > > >
> > > >  10. Comment: *``How can one dabble with the math of these papers and not get the semantic content of their conditions ? This is unexpected. If a human is using the math, then the human should have some idea of what it is doing, no ?"*
> > > >
> > > >  Response: We are surprised that such a clarification is necessary, but we would like to firmly state that all mathematical analysis and proofs in the manuscript were carried out by the authors. The reviewer raised several questions regarding the interpretation of certain technical conditions, and we have addressed each of these points carefully in our previous responses. That said, we believe every scientific accusation should be grounded by evidence.

---

> > > > > ### Author Response · Authors · 2025-11-20
> > > > > **References**
> > > > >
> > > > > John Sipple. Interpretable, multidimensional, multimodal anomaly detection with negative sampling
> > > > > for detection of device failure. In *International Conference on Machine Learning (ICML)*, pp.
> > > > > 9016–9025. PMLR, 2020.
> > > > >
> > > > > Ingo Steinwart, Don Hush, and Clint Scovel. A classification framework for anomaly detection. *Journal of Machine Learning Research*, 6(8):211–232, 2005.
> > > > >
> > > > > Jean-Yves Audibert and Alexandre B Tsybakov. Fast learning rates for plug-in classifiers. *The Annals
> > > > > of Statistics*, pp. 608–633, 2007.
> > > > >
> > > > > Clayton Scott and Robert D. Nowak. Minimax-optimal classification with dyadic decision trees. *IEEE Transactions on Information Theory* 52, no. 4 (2006): 1335-1353.
> > > > >
> > > > > Stéphan Clémençon and Sylvain Robbiano. Minimax learning rates for bipartite ranking and plug-in rules. In *International Conference on Machine Learning (ICML)*, pp. 441-448. 2011.
> > > > >
> > > > > Yongdai Kim, Ilsang Ohn, and Dongha Kim. Fast convergence rates of deep neural networks for classification. *Neural Networks* 138 (2021): 179-197.
> > > > >
> > > > > Tianyang Hu, Ruiqi Liu, Zuofeng Shang, and Guang Cheng. Minimax optimal deep neural network classifiers under smooth decision boundary. *Journal of Machine Learning Research* 26, no. 136 (2025): 1-38.

---

> > > > > ### Comment · Reviewer_n4YP · 2025-11-25
> > > > > **reply**
> > > > >
> > > > > >> First of all, Steinwart et al. (2005) and our paper consider fundamentally different function classes to tackle the anomaly detection problem. Steinwart et al. (2005) study support vector machines, a kernel-based method, and all theoretical results in that work are derived in the reproducing kernel Hilbert space (RKHS) framework. In contrast, our paper studies ReLU neural network classifiers, whose function class, approximation properties, and theoretical guarantees differ substantially from those of SVMs.
> > > > >
> > > > > Here the reviewer also disagrees. The main result in Steinwart 2005, Theorem 10 does not depend on RKHS or SVM specifics. Theorem 10 in Steinwart 2005 covers already the case of any reasonable negative density if Tsybakovs noise conditions holds. That covers also the case when one would sample from the negative class.
> > > > >
> > > > > >> Response: We are surprised that such a clarification is necessary, but we would like to firmly state
> > > > >
> > > > > please see the reviewers comment on Massarts noise. What is stated in this paper about insufficient irregularity is a direct contradiction to existing results in learning theory.

---

> > > > ### Comment · Reviewer_n4YP · 2025-11-25
> > > > **reply**
> > > >
> > > > point 1
> > > > >> A one-layer ReLU network of the form
> > > > is a globally continuous piecewise-linear function with . Such networks cannot uniformly approximate a discontinuous function over any region in which the discontinuity occurs.
> > > >
> > > > That misses the point because for classification one uses thresholding. Then that is solved.
> > > >
> > > > >> The "insufficient regularity of learning" problem we refer to, as indicated in Proposition 3.1, concerns situations in which the regression function $f_P(X) = E[Y=1|X]$ becomes a discontinuous step function taking only the values $1$ or $-1$, that is $f_P(x)=1$ if $x\in X_1$, and $f_P(x)=-1$ if $x\in X_{-}$.
> > > >
> > > > Exactly this is where learning theory shows the opposite. To convince the other reviewers, see eg slides 7 and 8 here https://www.stats.ox.ac.uk/~rebeschi/teaching/AFoL/22/material/slides07.pdf
> > > >
> > > >
> > > >
> > > > consider Massarts noise condition shown there. This one is easier to interpret
> > > > There exists a $\gamma \in [0,1/2)$ such that
> > > > $$P(|\eta(x)-1/2|> \gamma)  = 1$$
> > > >
> > > > A clear discontinuity at the decision boundary. If the problem is that discontinuos, then one can get $O(n^{-1})$ fast rates of convergence and not the standard $O(n^{-1/2})$. The claim that the authors are making about "insufficient regularity of learning" is technically not right.
> > > >
> > > > One can also see Tsybakovs noise in   https://www.stats.ox.ac.uk/~rebeschi/teaching/AFoL/22/material/slides07.pdf  two slides later. One gets fast convergence if $\alpha \rightarrow 1$ in the notation of these slides.
> > > >
> > > > At this point a major claim of the submission appears to be wrong.

---

> > ### Comment · Reviewer_n4YP · 2025-11-25
> >
> > okay the reviewer agrees here, the reviewer was wrong about Proposition 4.2.

---

### Meta-Review · Area_Chair_GbhS · 2025-12-28

**Summary:**

The paper proposes theory and a theoretically grounded methodology for semi-supervised anomaly detection. The approach is to classify normalities versus anomalies, where the anomaly class is enriched by uniformly drawn auxiliary anomalies.

As the authors acknowledge, the method itself is not novel. The idea of incorporating synthetic anomalies, in particular uniformly drawn ones, was already proposed in Steinwart et al. (2005). The authors argue that their main contribution lies on the theoretical side and claim that their framework constitutes the first theoretical framework for semi-supervised anomaly detection (“Theoretically, we provide the first mathematical formulation of semi-supervised AD”, line 51). However, a theoretical framework for semi-supervised anomaly detection was already presented by Blanchard et al. (2010) "Semi-Supervised Novelty Detection" (JMLR 11 (2010) 2973-3009), which appeared more than 15 years ago and is not referenced in the present paper.

Moreover, the main theoretical result, presented as Equation (4), contains a major error in the submitted version. As acknowledged by the authors in the rebuttal, the inequality sign needs to be reversed. Overall, the theory appears to be an extension of Steinwart et al. (2005), but the paper does not sufficiently differentiate its proofs and results from the original work. This makes it difficult for the reviewer to assess which parts, if any, are substantially novel from a proof or methodological perspective, and which parts closely follow existing arguments.

As pointed out by Reviewer n4YP in a detailed review, the main results around Theorem 10 in Steinwart et al. (2005) apply to arbitrary measurable classifiers and are not restricted to kernel-based methods. Consequently, the paper’s repeated argument that it provides a fundamentally new neural network–specific analysis is not fully convincing.

In conclusion, the paper’s contributions on both the theoretical and methodological sides are minor, and the novelty is overstated. I therefore recommend rejecting the paper in its current form.

**Reviewer Concerns:**

The reviewers raised concerns about overstated novelty on both the methodological and theoretical sides. The empirical approach of augmenting anomaly detection with uniformly sampled synthetic anomalies is well known from prior work. The main theoretical claims are insufficiently positioned relative to existing frameworks for semi-supervised anomaly detection, and key results are closely related to earlier theory. Reviewers also questioned the central motivation based on “insufficient regularity of learning,” the clarity of the presentation, and the interpretation of empirical results.

**Reviewer Scores:**

The authors’ rebuttal addresses some technical details but does not sufficiently resolve the main concerns raised by Reviewers SFnp and n4YP. It is therefore unlikely that the reviewers’ scores would have been raised based on the rebuttal.

---

### Decision · Program_Chairs · 2026-01-26

Reject